# A Theoretical Framework for Target Propagation

**Alexander Meulemans[1], Francesco S. Carzaniga[1], Johan A.K. Suykens[2],**
**João Sacramento[1], Benjamin F. Grewe[1]**

[1]Institute of Neuroinformatics, University of Zürich and ETH Zürich
[2]ESAT-STADIUS, KU Leuven
ameulema@ethz.ch

## Abstract

The success of deep learning, a brain-inspired form of AI, has sparked interest in understanding how the brain could similarly learn across multiple layers of neurons. However, the majority of biologically-plausible learning algorithms have not yet reached the performance of backpropagation (BP), nor are they built on strong theoretical foundations. Here, we analyze target propagation (TP), a popular but not yet fully understood alternative to BP, from the standpoint of mathematical optimization. Our theory shows that TP is closely related to Gauss-Newton optimization and thus substantially differs from BP. Furthermore, our analysis reveals a fundamental limitation of difference target propagation (DTP), a well-known variant of TP, in the realistic scenario of non-invertible neural networks. We provide a first solution to this problem through a novel reconstruction loss that improves feedback weight training, while simultaneously introducing architectural flexibility by allowing for direct feedback connections from the output to each hidden layer. Our theory is corroborated by experimental results that show significant improvements in performance and in the alignment of forward weight updates with loss gradients, compared to DTP.

## 1 Introduction

Backpropagation (BP) (Rumelhart et al., 1986; Linnainmaa, 1970; Werbos, 1982) has emerged as the gold standard for training deep neural networks (LeCun et al., 2015) but long-standing criticism on whether it can be used to explain learning in the brain across multiple layers of neurons has prevailed (Crick, 1989). First, BP requires exact weight symmetry of forward and backward pathways, also known as the *weight transport problem*, which is not compatible with the current evidence from experimental neuroscience studies (Grossberg, 1987). Second, it requires the transmission of signed error signals (Lillicrap et al., 2020). This raises the question whether (i) weight transport and (ii) signed error transmission are necessary for training multilayered neural networks.

Recent work (Lillicrap et al., 2016; Nøkland, 2016) showed that random feedback connections are sufficient to propagate errors and that feedback does not need to adhere to the layer-wise structure of the forward pathway, thereby indicating that weight transport is not strictly necessary for training multilayered neural networks. However, follow-up work (Bartunov et al., 2018; Launay et al., 2019; Moskovitz et al., 2018; Crafton et al., 2019) indicated that random feedback weights are not sufficient for more complex problems and require adjustments to better approximate the symmetric layer-wise connectivity of BP (Akrout et al., 2019; Kunin et al., 2020; Liao et al., 2016; Xiao et al., 2018; Guerguiev et al., 2020), although encouraging recent results suggest that the symmetric connectivity constraint from BP might be surmountable (Lansdell et al., 2020).

*Target propagation* (TP) represents a fundamentally different stream of research into alternatives for BP, as it propagates target activations (not errors) to the hidden layers of the network and then

updates the weights of each layer to move closer to the target activation (Bengio, 2014; Lee et al., 2015; Bengio et al., 2015; Ororbia and Mali, 2019; Manchev and Spratling, 2020; LeCun, 1986). TP thereby alleviates the two main criticisms on the biological plausibility of BP. Another complementary line of research investigates how learning rules could be implemented in biological micro-circuits (Sacramento et al., 2018; Guerguiev et al., 2017; Lillicrap et al., 2020) and relies on the core principles of TP. While TP as presented in Bengio (2014) and Lee et al. (2015) is appealing for bridging the gap between deep learning and neuroscience, its optimization properties are not yet fully understood, neither does it scale to more complex problems (Bartunov et al., 2018).

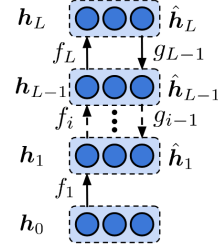

Here, we present a novel theoretical framework for TP and its well-known variant *difference target propagation* (DTP; Lee et al. (2015)) that describes its optimization characteristics and limitations. More specifically, we (i) identify TP in the setting of invertible networks as a hybrid method between approximate Gauss-Newton optimization and gradient descent. (ii) We show that, consistent with its limited performance on challenging problems, DTP suffers from inefficient parameter updates for non-invertible networks. (iii) To overcome this limitation, we propose a novel reconstruction loss for DTP, that restores the hybrid between Gauss-Newton optimization and gradient descent and (iv) we provide theoretical insights into the optimization characteristics of this hybrid method. (v) We introduce a DTP variant with direct feedback connections from the output to the hidden layers. (vi) Finally, we provide experimental results showing that our new DTP variants improve the ability to propagate useful feedback signals and thus learning performance.

Figure 1: Schematic illustration of the TP framework.

## 2 Background and Notation

We briefly review TP (Lee et al., 2015; Bengio, 2014) and Gauss-Newton optimization (GN). See Fig. 1 for a schematic of TP and the supplementary materials (SM) for more information on GN.

**Target propagation.** We consider a feedforward fully connected network with forward mappings:

$$\boldsymbol{h}_i = f_i(\boldsymbol{h}_{i-1}) = s_i(W_i \boldsymbol{h}_{i-1}) = s_i(\boldsymbol{a}_i), \quad i = 1, ..., L, \tag{1}$$

with $\boldsymbol{h}_i$ the vector with post-activation values of layer $i$, $\boldsymbol{a}_i$ the pre-activation values, $s_i$ a smooth nonlinear activation function, $W_i$ the layer weights, $f_i$ a shorthand notation and $\boldsymbol{h}_0$ the network input. Based on the output $\boldsymbol{h}_L$ of the network and the label $\boldsymbol{l}$ of the training sample, a loss $\mathcal{L}(\boldsymbol{l}, \boldsymbol{h}_L)$ is computed. While BP backpropagates the gradients of this loss function, TP computes an output target and backpropagates this target. The output target $\hat{\boldsymbol{h}}_L$ is defined as the output activation tweaked in the negative gradient direction:

$$\hat{\boldsymbol{h}}_L = \boldsymbol{h}_L - \hat{\eta} e_L \triangleq \boldsymbol{h}_L - \hat{\eta} \Big( \frac{\partial \mathcal{L}}{\partial \boldsymbol{h}_L} \Big)^T, \tag{2}$$

with $\hat{\eta}$ the output target stepsize. Note that with $\hat{\eta} = 0.5$ and an L2 loss, we have $\hat{\boldsymbol{h}}_L = \boldsymbol{l}$. $\hat{\boldsymbol{h}}_L$ is backpropagated to produce hidden layer targets $\hat{\boldsymbol{h}}_i$:

$$\hat{\boldsymbol{h}}_i = g_i(\hat{\boldsymbol{h}}_{i+1}) = t_i(Q_i \hat{\boldsymbol{h}}_{i+1}), \quad i = L - 1, ..., 1. \tag{3}$$

with $g_i$ an approximate inverse of $f_{i+1}$, $Q_i$ the feedback weights and $t_i$ a smooth nonlinear activation function. One can also choose other parameterizations for $g_i$. Based on $\hat{\boldsymbol{h}}_i$, local layer losses $\mathcal{L}_i(\hat{\boldsymbol{h}}_i, \boldsymbol{h}_i) = \|\hat{\boldsymbol{h}}_i - \boldsymbol{h}_i\|_2^2$ are defined. The forward weights $W_i$ are then updated by taking a gradient descent step on this local loss, assuming that $\hat{\boldsymbol{h}}_i$ stays constant. Finally, to train the feedback parameters $Q_i$, $f_{i+1}$ and $g_i$ are seen as a shallow auto-encoder pair. $Q_i$ can then be trained by a gradient step on a reconstruction loss:

$$\mathcal{L}_i^{\text{rec}}\Big( g_i \big( f_{i+1}(\boldsymbol{h}_i) \big), \boldsymbol{h}_i \Big) = \big\| g_i \big( f_{i+1}(\boldsymbol{h}_i) \big) - \boldsymbol{h}_i \big\|_2^2. \tag{4}$$

Lee et al. (2015) argue for injecting additive noise in $\boldsymbol{h}_i$ for the reconstruction loss, such that the backward mapping is also learned in a region around the training points.

**Gauss-Newton optimization.** The Gauss-Newton (GN) algorithm (Gauss, 1809) is an iterative approximate second-order optimization method that is used for non-linear least-squares regression

problems. The GN update for the model parameters $\boldsymbol{\beta}$ is given by:

$$\Delta\boldsymbol{\beta} = -\left(J^T J + \lambda I\right)^{-1} J^T \boldsymbol{e}_L = -J^T \left(J J^T + \lambda I\right)^{-1} \boldsymbol{e}_L, \tag{5}$$

with $J$ the Jacobian of the model outputs w.r.t $\boldsymbol{\beta}$, $\boldsymbol{e}_L$ the output errors (considering an $L_2$ loss) and $\lambda$ a Tikhonov damping constant (Tikhonov, 1943; Levenberg, 1944). For $\lambda \to 0$, the update simplifies to $\Delta\boldsymbol{\beta} = -J^\dagger \boldsymbol{e}$, with $J^\dagger$ the Moore-Penrose pseudo inverse of $J$ (Moore, 1920; Penrose, 1955).

## 3   Theoretical Results

To understand how TP can optimize a loss function, we need to know what kind of update the propagated targets represent. Here, we lay down a theoretical framework for TP, showing that the targets represent an update computed with an approximate Gauss-Newton optimization method for invertible networks and we improve DTP to propagate these *Gauss-Newton targets* in non-invertible networks. Furthermore, we show how these GN targets optimize a global loss function.

### 3.1   TP for invertible networks computes GN targets

Invertible networks represent the ideal case for TP, as TP relies on using approximate inverses to propagate targets through the network. We formalize invertible feed-forward networks as follows:

**Condition 1** (Invertible networks). *The feed-forward neural network has forward mappings $\boldsymbol{h}_i = f_i(\boldsymbol{h}_{i-1}) = s_i(W_i \boldsymbol{h}_{i-1})$, $i = 1, ..., L$ where $s_i$ can be any differentiable, monotonically increasing and invertible element-wise function with domain and image equal to $\mathbb{R}$ and where $W_i$ can be any invertible matrix. The feedback functions for propagating the targets are the inverse of the forward mapping functions: $g_i(\hat{\boldsymbol{h}}_{i+1}) = f_{i+1}^{-1}(\hat{\boldsymbol{h}}_{i+1}) = W_{i+1}^{-1} s_{i+1}^{-1}(\hat{\boldsymbol{h}}_{i+1})$.*

The target update $\Delta\boldsymbol{h}_i \triangleq \hat{\boldsymbol{h}}_i - \boldsymbol{h}_i$ represents how the hidden layer activations should be changed to decrease the output loss and plays a similar role to the backpropagation error $\boldsymbol{e}_i \triangleq \partial\mathcal{L}/\partial\boldsymbol{h}_i$. As shown by Lee et al. (2015), a first-order Taylor expansion of this target update reveals how the output error gets propagated to the hidden layers, which we restate in Lemma 1 (full proof in SM).

**Lemma 1.** *Assuming Condition 1 holds and the output target stepsize $\hat{\eta}$ is small compared to $\|\boldsymbol{h}_L\|_2/\|\boldsymbol{e}_L\|_2$, the target update $\Delta\boldsymbol{h}_i \triangleq \hat{\boldsymbol{h}}_i - \boldsymbol{h}_i$ can be approximated by*

$$\Delta\boldsymbol{h}_i \triangleq \hat{\boldsymbol{h}}_i - \boldsymbol{h}_i = -\hat{\eta}\left[\prod_{k=i}^{L-1} J_{f_{k+1}}^{-1}\right]\boldsymbol{e}_L + \mathcal{O}(\hat{\eta}^2) = -\hat{\eta} J_{\bar{f}_{i,L}}^{-1}\boldsymbol{e}_L + \mathcal{O}(\hat{\eta}^2), \tag{6}$$

*with $\boldsymbol{e}_L = (\partial\mathcal{L}/\partial\boldsymbol{h}_L)^T$ evaluated at $\boldsymbol{h}_L$, $J_{f_{k+1}} = \partial f_{k+1}(\boldsymbol{h}_k)/\partial\boldsymbol{h}_k$ evaluated at $\boldsymbol{h}_k$ and $J_{\bar{f}_{i,L}} = \partial f_L(..(f_{i+1}(\boldsymbol{h}_i)))/\partial\boldsymbol{h}_i$ evaluated at $\boldsymbol{h}_i$.*

If this target update is compared with the BP error $\boldsymbol{e}_i = \prod_{k=i}^{L-1} J_{f_{k+1}}^T \boldsymbol{e}_L$, we see that the transpose operation is replaced by an inverse operation. Gauss-Newton optimization (GN) uses a pseudo-inverse of the Jacobian of the output with respect to the parameters (eq. 5), which hints towards a relation between TP and GN. The following theorem makes this relationship explicit (full proof in SM).

**Theorem 2.** *Consider an invertible network specified in Condition 1. Further, assume a mini-batch size of 1 and an $L_2$ output loss function. Under these conditions and in the limit of $\hat{\eta} \to 0$, TP uses Gauss-Newton optimization with a block-diagonal approximation of the Gauss-Newton curvature matrix, with block sizes equal to the layer sizes, to compute the local layer targets $\hat{\boldsymbol{h}}_i$.*

Theorem 2 thus shows that TP can be interpreted as a hybrid method between Gauss-Newton optimization and gradient descent. First, an approximation of GN is used to compute the hidden layer targets, after which gradient descent on the local losses $\mathcal{L}_i = \|\hat{\boldsymbol{h}}_i - \boldsymbol{h}_i\|_2^2$ is used for updating the forward parameters.

### 3.2   DTP for non-invertible networks does not compute GN targets

In general, deep networks are not invertible due to varying layer sizes and non-invertible activation functions. For non-invertible networks, $g_i$ is not the exact inverse of $f_{i+1}$ but tries to approximate it

instead. This approximation causes reconstruction errors to interfere with the target updates $\Delta \boldsymbol{h}_i$, as can be seen in its first order Taylor approximation.

$$\Delta \boldsymbol{h}_i \triangleq \hat{\boldsymbol{h}}_i - \boldsymbol{h}_i = -J_{g_i} \Delta \boldsymbol{h}_{i+1} + g_i(\boldsymbol{h}_{i+1}) - \boldsymbol{h}_i + \mathcal{O}(\|\Delta \boldsymbol{h}_{i+1}\|_2^2) \tag{7}$$

The second and third term in the right-hand side represent the reconstruction error, as this error would be zero if $g_i$ is the perfect inverse of $f_{i+1}$. Due to the interference of these reconstruction errors with the target updates, vanilla TP fails to propagate useful learning signals backwards for non-invertible networks. The *Difference Target Propagation* (DTP) method (Lee et al., 2015) solves this issue by subtracting the reconstruction error from the propagated targets: $\hat{\boldsymbol{h}}_i = g_i(\hat{\boldsymbol{h}}_{i+1}) - \big(g_i(\boldsymbol{h}_{i+1}) - \boldsymbol{h}_i\big)$, known as the *difference correction*. Similar to Lemma 1, we obtain an approximation of the DTP target updates.

**Lemma 3.** *Assuming the output target step size $\hat{\eta}$ is small compared to $\|\boldsymbol{h}_L\|_2 / \|\boldsymbol{e}_L\|_2$, the target update $\Delta \boldsymbol{h}_i \triangleq \hat{\boldsymbol{h}}_i - \boldsymbol{h}_i$ of the DTP method can be approximated by*

$$\Delta \boldsymbol{h}_i \triangleq \hat{\boldsymbol{h}}_i - \boldsymbol{h}_i = -\hat{\eta} \left[ \prod_{k=i}^{L-1} J_{g_k} \right] \boldsymbol{e}_L + \mathcal{O}(\hat{\eta}^2), \tag{8}$$

*with $J_{g_k} = \partial g_k(\boldsymbol{h}_{k+1}) / \partial \boldsymbol{h}_{k+1}$ evaluated at $\boldsymbol{h}_{k+1}$.*

For propagating GN targets, $\prod_{k=i}^{L-1} J_{g_k}$ needs to be equal to $J_{\bar{f}_{i,L}}^\dagger$, with $\bar{f}_{i,L}$ the forward mapping from layer $i$ to the output. However, two issues prevent this from happening in DTP. First, the layer-wise reconstruction loss (eq. 4) for training the parameters of $g_i$, combined with the nonlinear parameterization of $g_i$ and $f_{i+1}$ (eq. 3 and 1), does not ensure that $J_{g_i} = J_{f_{i+1}}^\dagger$ (see SM). Second, even if $J_{g_i} = J_{f_{i+1}}^\dagger$ for all layers, it would still in general not be the case that $\prod_{k=i}^{L-1} J_{g_k} = J_{\bar{f}_{i,L}}^\dagger$, as $J_{\bar{f}_{i,L}}^\dagger$ cannot be factorized as $\prod_{k=i+1}^{L} J_{f_k}^\dagger$ in general (Campbell and Meyer, 2009). From these two issues, we see that DTP does not propagate real GN targets to its hidden layers by default and in section 3.4 we show that this leads to inefficient parameter updates. However, by introducing a new reconstruction loss used for training the feedback functions $g_i$, we can ensure that the DTP method propagates approximate GN targets.

### 3.3   Propagating Gauss-Newton targets in non-invertible networks

Here, we present a novel *difference reconstruction loss* (DRL) that trains the feedback parameters to propagate GN targets.

**Difference reconstruction loss.** First, we introduce shorthand notations for how targets get propagated through the network in DTP. The computation of $\hat{\boldsymbol{h}}_i$ based on the target in the next layer can be written as

$$\hat{\boldsymbol{h}}_i = g_i^{\text{diff}}(\hat{\boldsymbol{h}}_{i+1}, \boldsymbol{h}_{i+1}, \boldsymbol{h}_i) \triangleq g_i(\hat{\boldsymbol{h}}_{i+1}) + \big(\boldsymbol{h}_i - g_i\big(f_{i+1}(\boldsymbol{h}_i)\big)\big). \tag{9}$$

The sequence of computations for $\hat{\boldsymbol{h}}_i$ based on the output target can be defined recursively as $\hat{\boldsymbol{h}}_i = \bar{g}_{L,i}^{\text{diff}}(\hat{\boldsymbol{h}}_L, \boldsymbol{h}_L, \boldsymbol{h}_i) \triangleq g_i^{\text{diff}}\big(\bar{g}_{L,i+1}^{\text{diff}}(\hat{\boldsymbol{h}}_L, \boldsymbol{h}_L, \boldsymbol{h}_{i+1}), \boldsymbol{h}_{i+1}, \boldsymbol{h}_i\big)$, with $\bar{g}_{L,L-1}^{\text{diff}} = g_{L-1}^{\text{diff}}$. Further, $g_i(\hat{\boldsymbol{h}}_{i+1})$ can be defined as a function of the output target as $g_i(\hat{\boldsymbol{h}}_{i+1}) = \bar{g}_{L,i}(\hat{\boldsymbol{h}}_L) \triangleq g_i\big(\bar{g}_{L,i+1}^{\text{diff}}(\hat{\boldsymbol{h}}_L, \boldsymbol{h}_L, \boldsymbol{h}_{i+1})\big)$. Finally, consider $\bar{f}_{i,L}$ as the forward mapping from layer $i$ to the output. With this shorthand notation, DRL can be defined compactly as

$$\mathcal{L}_i^{\text{rec,diff}} = \frac{1}{\sigma^2} \sum_{b=1}^{B} \mathbb{E}_{\boldsymbol{\epsilon}_1 \sim \mathcal{N}(0,1)} \left[ \|\bar{g}_{L,i}^{\text{diff}}\big(\bar{f}_{i,L}(\boldsymbol{h}_i^{(b)} + \sigma \boldsymbol{\epsilon}_1), \boldsymbol{h}_L^{(b)}, \boldsymbol{h}_i^{(b)}\big) - (\boldsymbol{h}_i^{(b)} + \sigma \boldsymbol{\epsilon}_1)\|_2^2 \right]$$
$$+ \mathbb{E}_{\boldsymbol{\epsilon}_2 \sim \mathcal{N}(0,1)} \left[ \lambda \|\bar{g}_{L,i}^{\text{diff}}(\boldsymbol{h}_L^{(b)} + \sigma \boldsymbol{\epsilon}_2, \boldsymbol{h}_L^{(b)}, \boldsymbol{h}_i^{(b)}) - \boldsymbol{h}_i^{(b)}\|_2^2 \right], \tag{10}$$

with $B$ the minibatch size and $\sigma$ the noise standard deviation. See Figure S1 for a schematic of DRL. The parameters of $g_i$ are updated by gradient descent on $\mathcal{L}_i^{\text{rec,diff}}$. $\mathcal{L}_i^{\text{rec,diff}}$ is also dependent on other feedback mapping functions $g_{j>i}$, however, their parameters are not updated with $\mathcal{L}_i^{\text{rec,diff}}$, but with the corresponding $\mathcal{L}_j^{\text{rec,diff}}$ instead.

DRL is based on three intuitions. First, we send a noise-corrupted sample $\boldsymbol{h}_i^{(b)} + \sigma\boldsymbol{\epsilon}$ in a reconstruction loop through the output layer, instead of only through the next layer. This is needed because $J_{\bar{f}_{i,L}}^{\dagger}$, which needs to be approximated for propagating GN targets, is not factorizable over the layers. Second, we use the same difference correction as in DTP, to ensure that the expectation can be taken over white noise, while $J_{\bar{f}_{i,L}}$ is still evaluated at the real training representations $\boldsymbol{h}_i^{(b)}$. Third, we introduce a regularization term that plays a similar role to Tikhonov damping in GN (see eq. 5). DRL uses the same feedback path $\bar{g}_{L,i}^{\text{diff}}\big(\bar{f}_{i,L}(\boldsymbol{h}_i + \sigma\boldsymbol{\epsilon}_1), \boldsymbol{h}_L, \boldsymbol{h}_i\big)$ as for propagating the target signals $\bar{g}_{L,i}^{\text{diff}}(\hat{\boldsymbol{h}}_L, \boldsymbol{h}_L, \boldsymbol{h}_i)$ in the training phase for the forward weights, with a noise-corrupted sample $\bar{f}_{i,L}(\boldsymbol{h}_i + \sigma\boldsymbol{\epsilon}_1)$ instead of the output target $\hat{\boldsymbol{h}}_L$. In the following theorem, we show that our DRL trains the feedback mappings $g_i$ in a correct way for propagating approximate GN targets (full proof in SM).

**Theorem 4.** *The difference reconstruction loss for layer $i$, with $\sigma$ driven in limit to zero, is equal to*

$$\lim_{\sigma \to 0} \mathcal{L}_i^{rec,diff} = \sum_{b=1}^{B} \mathop{\mathbb{E}}_{\boldsymbol{\epsilon}_1 \sim \mathcal{N}(0,1)} \Big[ \| J_{\bar{g}_{L,i}}^{(b)} J_{\bar{f}_{i,L}}^{(b)} \boldsymbol{\epsilon}_1 - \boldsymbol{\epsilon}_1 \|_2^2 \Big] + \mathop{\mathbb{E}}_{\boldsymbol{\epsilon}_2 \sim \mathcal{N}(0,1)} \Big[ \lambda \| J_{\bar{g}_{L,i}}^{(b)} \boldsymbol{\epsilon}_2 \|_2^2 \Big], \qquad (11)$$

*with $J_{\bar{g}_{L,i}}^{(b)} = \prod_{k=i}^{L-1} J_{\bar{g}_k}^{(b)}$ the Jacobian of feedback mapping function $\bar{g}_{L,i}$ evaluated at $\boldsymbol{h}_k^{(b)}$, $k = i+1,..,L$, and $J_{\bar{f}_{i,L}}^{(b)}$ the Jacobian of the forward mapping function $\bar{f}_{i,L}$ evaluated at $\boldsymbol{h}_i^{(b)}$. The minimum of $\lim_{\sigma \to 0} \mathcal{L}_i^{rec,diff}$ is reached if for each batch sample holds that*

$$J_{\bar{g}_{L,i}}^{(b)} = \prod_{k=i}^{L-1} J_{\bar{g}_k}^{(b)} = J_{\bar{f}_{i,L}}^{(b)T} \big( J_{\bar{f}_{i,L}}^{(b)} J_{\bar{f}_{i,L}}^{(b)T} + \lambda I \big)^{-1}. \qquad (12)$$

*When the regularization parameter $\lambda$ is driven in limit to zero, this results in $J_{\bar{g}_{L,i}}^{(b)} = J_{\bar{f}_{i,L}}^{(b)\dagger}$.*

Theorem 4 shows that by minimizing the difference reconstruction loss for training the feedback mappings $g_i$, we get closer to propagating GN targets as $\hat{\boldsymbol{h}}_i$. In equation (12), we see that the regularization term introduces Tikhonov damping (see eq. 5). This damping interpolates between the pseudo-inverse and the transpose of $J_{\bar{f}_{i,L}}$, so for large $\lambda$, GN targets resemble gradient targets. For practical reasons, we approximate the expectations with a single sample during training and replace the regularization term by weight decay on the feedback parameters, as this has a similar effect on restricting the magnitude of $\| J_{\bar{g}_{L,i}}^{(b)} \|_F^2$ (see SM). In practice, the absolute minimum of the difference reconstruction loss will not be reached, as $J_{\bar{g}_{L,i}}^{(b)}$, for different samples $b$, will depend on the same limited amount of parameters of $g_i$. Hence, a parameter setting will be sought that brings $J_{\bar{g}_{L,i}}^{(b)}$ as close as possible to $J_{\bar{f}_{i,L}}^{(b)\dagger}$ for all batch samples $b$, but they will in general not be equal for each $b$.

**Direct difference target propagation.** The theory behind DRL motivates direct connections from the output towards the hidden layers for propagating targets. The idea for widening the reconstruction loop from layer $i$ to the output layer arose from the fact that the pseudo-inverse of $J_{\bar{f}_{i,L}}$ cannot be factorized over layer-wise pseudo-inverses of $J_{f_{k>i}}$. As the training of feedback paths does not benefit from adhering to the layer-wise structure, we can push this idea further by introducing *Direct Difference Target Propagation* (DDTP) as a new DTP variant. In DDTP, the network has direct feedback mapping functions $g_i(\hat{\boldsymbol{h}}_L)$ from the output to hidden layer $i$. Various parametrizations of $g_i$ are possible, as shown in Fig. 3. In the notation of the previous section, $\bar{g}_{L,i}^{\text{diff}}(\hat{\boldsymbol{h}}_L, \boldsymbol{h}_L, \boldsymbol{h}_i) = g_i^{\text{diff}}(\hat{\boldsymbol{h}}_L, \boldsymbol{h}_L, \boldsymbol{h}_i)$ and $\bar{g}_{L,i}(\hat{\boldsymbol{h}}_L) = g_i(\hat{\boldsymbol{h}}_L)$. With this notation, the difference reconstruction loss can be used out of the box to train the direct feedback mappings $g_i$.

### 3.4 Optimisation properties of Gauss-Newton targets

In the previous sections, we showed how DTP can train its feedback connections to propagate GN targets to the hidden layers. In this section, we investigate how the resulting hybrid method between Gauss-Newton and gradient descent is used to optimize the actual weight parameters of a neural network (all full proofs can be found in the SM). We consider the ideal case of perfect GN targets, called the *Gauss-Newton Target method* (GNT), as formalised by the condition below.

**Condition 2** (Gauss-Newton Target method). *The network is trained by GN targets: each hidden layer target is computed by*

$$\Delta \boldsymbol{h}_i^{(b)} \triangleq \hat{\boldsymbol{h}}_i^{(b)} - \boldsymbol{h}_i^{(b)} = -\hat{\eta} J_{\bar{f}_{i,L}}^{(b)\dagger} \boldsymbol{e}_L^{(b)}, \tag{13}$$

*after which the network parameters of each layer $i$ are updated by a gradient descent step on its corresponding local mini-batch loss $\mathcal{L}_i = \sum_b \|\Delta \boldsymbol{h}_i^{(b)}\|_2^2$, while considering $\hat{\boldsymbol{h}}_i$ fixed.*

We begin by investigating deep linear networks trained with GNT. As shown in Theorem 5, the GNT method has a characteristic behaviour for linear contracting networks. (i) Its parameter updates push the output activation along the negative gradient direction in the output space and (ii) its parameter updates are *minimum-norm* (i.e. the most efficient) in doing so.

**Theorem 5.** *Consider a contracting linear multilayer network ($n_1 \geq n_2 \geq .. \geq n_L$) trained by GNT according to Condition 2. For a mini-batch size of 1, the parameter updates $\Delta W_i$ are minimum-norm updates under the constraint that $\boldsymbol{h}_L^{(m+1)} = \boldsymbol{h}_L^{(m)} - c^{(m)} \boldsymbol{e}_L^{(m)}$ and when $\Delta W_i$ is considered independent from $\Delta W_{j \neq i}$, with $c^{(m)}$ a positive scalar and $m$ indicating the iteration.*

Two important corollaries follow from Theorem 5. First, we show that DTP on contracting linear networks also pushes the output activation along the negative gradient direction, however, its parameter updates are not minimum-norm, due to layer-wise training of the feedback weights. Hence, a substantial part of the DTP parameter updates does not have any effect on the network output, leading to inefficient parameter updates (see Fig. 2). This result helps to explain why DTP does not scale to more complex problems. Second, we show for linear networks, that GNT updates are aligned with the Gauss-Newton updates on the network parameters, for a minibatch size of 1. This follows from the minimum-norm properties of GN in an overparameterized setting (corollaries in SM). We stress that the alignment of the GNT updates with the GN parameter updates only holds for a minibatch size of 1. Averaging GNT parameter updates over a minibatch is not the same as computing the GN parameter updates on a minibatch (see SM). Usually, GN optimization for deep learning is done on large mini-batches (Botev et al., 2017; Martens and Grosse, 2015). However, recent theoretical results prove convergence of GN in an overparameterized setting with small minibatches (Cai et al., 2019; Zhang et al., 2019), resembling GNT on linear networks. In the following theorem, we show that a linear network trained with GNT indeed converges to the global minimum.

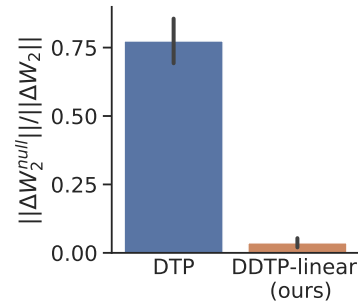

Figure 2: The average ratio between the norm of the components of $\Delta W_2$ that lie in the nullspace of $\partial \bar{f}_{0,L}(\boldsymbol{h}_0)/\partial \Delta W_2$ and the norm of $\Delta W_2$. The nullspace components cannot influence the output and are thus useless. A nonlinear 2-hidden-layer network was used on a synthetic regression dataset, trained with DTP (blue) and DDTP-linear (orange, see section 4). Error bars are the standard deviation.

**Theorem 6.** *Consider a linear multilayer network of arbitrary architecture, trained with GNT according to Condition 2 and with arbitrary batch size. The resulting parameter updates lie within 90 degrees of the corresponding loss gradients. Moreover, for an infinitesimal small learning rate, the network converges to the global minimum.*

For nonlinear networks, the minimum-norm interpretation of $\Delta W_i$ does not hold exactly. However, the target updates $\Delta \boldsymbol{h}_i$ of GNT are still minimum-norm for pushing the output along the negative gradient direction in the output space. Hence, the GN-part of the hybrid GNT method is minimum-norm, but the gradient descent part not anymore. In practice, the interpretation of Theorem 5 approximately holds for nonlinear networks (see SM). Further, GNT on nonlinear networks does not converge to a true local minimum of the loss function. However, our experimental results indicate that the DTP variants with approximate GN targets succeed in decreasing the output loss sufficiently, even for nonlinear networks (section 4). To help explain these results, we show that the GNT updates partially align with the loss gradient updates for networks with large hidden layers and a small output layer, as is the case in many network architectures for classification and regression tasks. Indeed, the properties of $J_{\bar{f}_{i,L}}$ are in general similar to those of a random matrix (Arora et al., 2015) and for zero-mean random matrices in $\mathbb{R}^{m \times n}$ with $n \gg m$, a scalar multiple of its transpose is a good

approximation of its pseudo-inverse. Intuitively, this is easy to understand, as $J^\dagger = J^T(JJ^T)^{-1}$ and $JJ^T$ is close to a scalar multiple of the identity matrix in this case (theorem in SM).

To conclude our theory, we showed that the layerwise training of the feedback parameters in DTP leads to inefficient forward parameter updates, and can be resolved by using the new DRL, which also allows for direct feedback connections. Further, we showed that TP and DTP, when combined with DRL, differ substantially from both BP and GN and can be best interpreted as a hybrid method between GN and gradient descent, which produces approximate minimum-norm parameter updates.

## 4  Experiments

We evaluate the new DTP variants on a set of standard image classification datasets: MNIST (LeCun, 1998), Fashion-MNIST (Xiao et al., 2017) and CIFAR10 (Krizhevsky et al., 2014).[1] We used fully connected networks with $\tanh$ nonlinearities, with a softmax output layer and cross-entropy loss, optimized by Adam (Kingma and Ba, 2014) (see SM for how our theory can be adapted to the cross-entropy loss). For the hyperparameter searches, we used a validation set of 5000 samples from the training set for all datasets. We report the test errors corresponding to the epoch with the best validation errors (experimental details in SM). We used targets to train all layers in DTP and its variants, in contrast to Lee et al. (2015), who trained the last hidden layer with BP.

We experimentally evaluate the following new DTP variants (algorithms available in SM). (i) For **DTP-DRL** (DTP with DRL) we use the same layerwise feedback architecture as in the original DTP method (see Fig. 1), but the feedback parameters are trained with DRL (eq. 10). (ii) We consider two DDTP variants, both trained with DRL. **DDTP-linear** has direct linear connections as shown in Fig. 3. In **DDTP-RHL** (DDTP with a Random Hidden Layer), the output target is projected by a random fixed matrix $R$ to a wide hidden feedback layer: $\hat{\boldsymbol{h}}_{fb} = \tanh(R\hat{\boldsymbol{h}}_L)$. From this hidden feedback layer, direct (trained) connections are made to the hidden layers of the network:

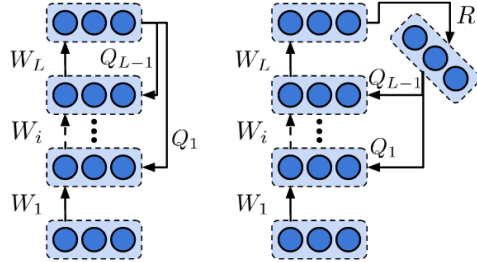

Figure 3: Structure of DDTP-linear (left) and DDTP-RHL (right).

$g_i(\hat{\boldsymbol{h}}_L) = \tanh(Q_i \hat{\boldsymbol{h}}_{fb})$ (see Fig. 3). (iii) To decouple the contributions of DRL with other factors, we did four controls. We compare our methods with **DTP** and with direct feedback alignment (**DFA**) (Nøkland, 2016) as a reference for methods with direct feedback connections. For **DDTP-control**, we train a DDTP-linear architecture with a reconstruction loss that incorporates a loop through the output layer, but does not use the difference correction that is present in DRL. In **DTP (pre-trained)**, we pre-train the feedback weights of DTP in the same manner as we do for our new DTP variants: 6 epochs before starting the training of the forward weights and one epoch of pure feedback training between each epoch of training both forward and feedback weights.

Table 1: Test errors corresponding to the epoch with the best validation error over a training of 100 epochs (5x256 fully connected (FC) hidden layers for MNIST and Fashion-MNIST, 3xFC1024 for CIFAR10). Mean $\pm$ SD for 10 random seeds. The best test errors (except BP) are displayed in bold.

|  | MNIST | Frozen-MNIST | Fashion-MNIST | CIFAR10 |
|---|---|---|---|---|
| BP | $1.98 \pm 0.14\%$ | $4.39 \pm 0.13\%$ | $10.74 \pm 0.16\%$ | $45.60 \pm 0.50\%$ |
| DDTP-linear | $\mathbf{2.04 \pm 0.08}\%$ | $6.42 \pm 0.17\%$ | $\mathbf{11.11 \pm 0.35}\%$ | $\mathbf{50.36 \pm 0.26}\%$ |
| DDTP-RHL | $2.10 \pm 0.14\%$ | $\mathbf{5.11 \pm 0.19}\%$ | $11.53 \pm 0.31\%$ | $51.94 \pm 0.49\%$ |
| DTPDRL | $2.21 \pm 0.09\%$ | $6.10 \pm 0.17\%$ | $11.22 \pm 0.20\%$ | $50.80 \pm 0.43\%$ |
| DDTP-control | $2.51 \pm 0.08\%$ | $9.70 \pm 0.31\%$ | $11.71 \pm 0.28\%$ | $51.75 \pm 0.43\%$ |
| DTP | $2.39 \pm 0.19\%$ | $10.64 \pm 0.53\%$ | $11.49 \pm 0.23\%$ | $51.74 \pm 0.30\%$ |
| DTP (pre-trained) | $2.26 \pm 0.18\%$ | $9.31 \pm 0.40\%$ | $11.52 \pm 0.31\%$ | $52.20 \pm 0.50\%$ |
| DFA | $2.17 \pm 0.14\%$ | / | $11.26 \pm 0.25\%$ | $51.28 \pm 0.41\%$ |

Table 1 displays the test errors for all experiments. DDTP-linear systematically outperforms both the original DTP method and the controls on all datasets. The better performance of DTPDRL compared to DTP shows that the DRL loss is indeed an improvement on the layer-wise reconstruction loss. Fig. 4 reveals a significant difference between the methods in the alignment of the updates with both the loss gradients and ideal GNT updates. Clearly, our methods are better capable of sending aligned teaching signals backwards through the network, despite that all performances lie close together. For further investigating whether the various methods are able to propagate useful learning signals, we designed the Frozen-MNIST experiment. In this experiment, all the forward parameters are frozen, except for those of the first hidden layer. All feedback parameters are still trained. To reach a good test performance, the network must be able to send useful teaching signals backwards deep into the network. Our results confirm that with DRL, the DTP variants are better capable of backpropagating useful teaching signals due to their better alignment with both the gradient and GNT updates. This experiment is not compatible with DFA, as this method relies on the alignment of all forward parameters with the fixed feedback parameters. We observed that DDTP-RHL, which has significantly more feedback parameters compared to DDTP-linear and DTPDRL, produces the best feedback signals in the frozen MNIST task, but is challenging to train on more complex tasks when the forward weights are not frozen.

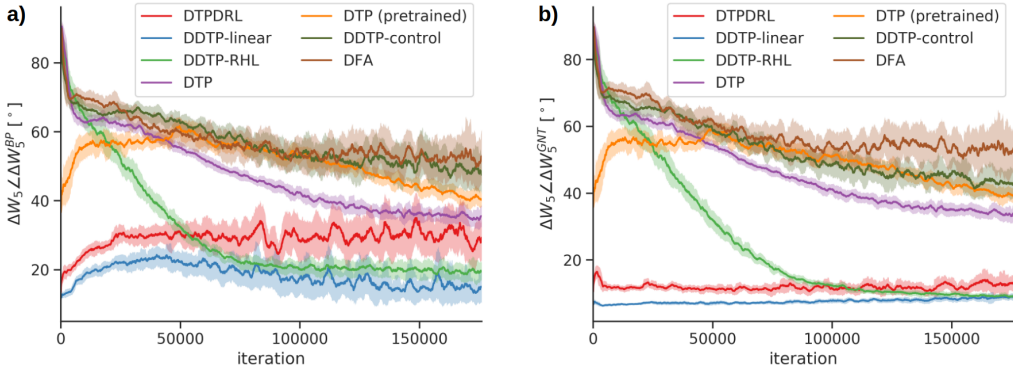

Figure 4: Angles between the weight updates $\Delta W_5$ of the fifth (last) hidden layer with (a) the loss gradient directions and (b) the GNT weight updates according to Condition 2 on Fashion-MNIST. A window-average of the angles is plotted, together with the window-standard-deviation.

For disentangling optimization capabilities and implicit regularization mechanisms that could both influence the test performance, we performed a second line of experiments focused on minimizing the training loss.[2] The results in Table 2 show that the DRL loss leads to significantly better optimization capabilities, compared to DTP and the controls. Interestingly, DDTP-linear achieves a lower training loss after 100 epochs compared to BP on MNIST and Fashion-MNIST, which might indicate that GNT can lead to acceleration on those simple datasets.

Table 2: Training loss after 100 epochs. Mean $\pm$ SD for 10 random seeds. The best training losses (except BP) are displayed in bold.

|  | MNIST | Frozen-MNIST | Fashion-MNIST | CIFAR10 |
|---|---|---|---|---|
| BP | $2.87^{\pm 4.11} \cdot 10^{-8}$ | $0.191^{\pm 0.019}$ | $6.46^{\pm 0.25} \cdot 10^{-5}$ | $1.01^{\pm 0.57} \cdot 10^{-7}$ |
| DDTP-linear | $\mathbf{1.99^{\pm 1.90} \cdot 10^{-9}}$ | $0.349^{\pm 0.029}$ | $\mathbf{1.03^{\pm 0.15} \cdot 10^{-5}}$ | $\mathbf{1.77^{\pm 0.06} \cdot 10^{-6}}$ |
| DDTP-RHL | $2.29^{\pm 0.56} \cdot 10^{-7}$ | $\mathbf{0.289^{\pm 0.014}}$ | $3.51^{\pm 0.80} \cdot 10^{-3}$ | $4.26^{\pm 3.61} \cdot 10^{-2}$ |
| DTPDRL | $3.43^{\pm 2.57} \cdot 10^{-9}$ | $0.317^{\pm 0.016}$ | $1.36^{\pm 0.48} \cdot 10^{-3}$ | $2.13^{\pm 0.83} \cdot 10^{-6}$ |
| DDTP-control | $4.74^{\pm 9.51} \cdot 10^{-6}$ | $0.995^{\pm 0.049}$ | $3.88^{\pm 2.63} \cdot 10^{-3}$ | $6.14^{\pm 0.92} \cdot 10^{-5}$ |
| DTP | $1.28^{\pm 1.04} \cdot 10^{-7}$ | $1.184^{\pm 0.072}$ | $4.07^{\pm 0.42} \cdot 10^{-2}$ | $9.81^{\pm 6.42} \cdot 10^{-5}$ |
| DTP (pre-trained) | $5.72^{\pm 5.91} \cdot 10^{-4}$ | $0.928^{\pm 0.002}$ | $2.73^{\pm 0.67} \cdot 10^{-2}$ | $6.77^{\pm 1.89} \cdot 10^{-5}$ |
| DFA | $1.57^{\pm 2.10} \cdot 10^{-6}$ | / | $1.98^{\pm 0.24} \cdot 10^{-2}$ | $2.05^{\pm 1.3} \cdot 10^{-5}$ |

As Theorem 4 applies to general feed-forward mappings, we can also use the GNT framework for convolutional neural networks (CNNs). As a proof-of-concept, we trained a small CNN on CIFAR10 with the methods that have direct feedback connections. Table 3 shows promising results, indicating that our theory can also be used in the more practical setting of CNNs, as DDTP-linear performs comparably to BP for this small CNN. For comparing with DTP and DTPDRL, careful design of the feedback pathways is needed, which is outside of the scope of this theoretical work.

Table 3: Test errors corresponding to the epoch with the best validation error over a training of 100 epochs on CIFAR10 with a small CNN with $\tanh$ nonlinearities (Conv5x5x32; Maxpool3x3; Conv5x5x64; Maxpool3x3; FC512; FC10). Mean $\pm$ SD for 10 seeds. The best test error (except BP) is displayed in bold.

| BP | DDTP-linear | DDTP-control | DFA |
|---|---|---|---|
| $24.38 \pm 0.29\%$ | $\mathbf{23.99 \pm 0.31}\%$ | $28.69 \pm 0.87\%$ | $30.00 \pm 0.74\%$ |

## 5 Discussion

In a seminal series of papers, LeCun introduced the TP framework and proposed a method to determine targets which is formally equivalent to BP (LeCun, 1986; LeCun et al., 1988). Thirty years later, Bengio (2014) suggested to propagate targets using shallow autoencoders, which could be trained without BP. Our theory establishes this form of TP, when extended to DTP and DRL, as a proper credit assignment algorithm, while uncovering fundamental differences to BP. In particular, we have shown that autoencoding TP is best seen as a hybrid between GN and gradient descent. Intriguingly, this suggests a potential link between TP and feedback alignment, a biologically plausible alternative to BP, which has also been related to GN under certain conditions (Lillicrap et al., 2016).

For the optimization of non-invertible neural networks, however, the connection to GN is lost in DTP. Therefore, we introduced a new reconstruction loss, DRL, which involves averaging over stochastic activity perturbations. This novel loss function not only reestablishes the link to GN, but also suggests a new family of algorithms which directly propagates targets from the network output to each hidden layer. Interestingly, numerous anatomical studies of the mammalian neocortex consistently reported such direct feedback connections in the brain (Ungerleider et al., 2008; Rockland and Van Hoesen, 1994). Our approach is similar in spirit to perturbative algorithms (Lansdell et al., 2020; Wayne, 2013; Le Cun et al., 1989), with the important difference that we recover GN targets instead of activation gradients.

In practice, the propagated targets for DRL methods are not exactly equal to GNT because of limited feedback parameterization capacity, limited training iterations for the feedback path and the approximation of $\lambda$ by weight decay. Figures 4 and S4-S6 demonstrate that our methods approximate GNT well. However, for upstream layers, future studies are required for further improvement, e.g. by investigating better feedback parameterizations or by using dynamical inversion (Podlaski and Machens, 2020). A better alignment between targets and GNT in upstream layers will likely improve the performance on more complex tasks. We note that while GN optimization enjoys desirable properties, it is presently unclear if GN targets can be more effective than gradients in neural network optimization. Future work is required to determine if DRL methods can close the gap to BP in large-scale problems, such as those considered by Bartunov et al. (2018).

DRL requires distinct phases to learn. In particular, it needs a separate noisy phase for each hidden layer, while the other layers are not corrupted by noise, similar to Lee et al. (2015), Akrout et al. (2019) and Kunin et al. (2020). Although there is mounting evidence for stochastic computation in cortex (e.g., London et al., 2010), coordinated alternations in noise levels are likely difficult to orchestrate in the brain. In this respect DRL is less biologically-plausible than standard DTP. Thus, DRL is best seen as a theoretical upper bound for training feedback weights to propagate GN targets, which can serve as a basis for future more biologically plausible feedback weight training methods.

To conclude, we have shown that it is possible to do credit assignment in a neural network – i.e., determine how each synaptic strength influences the output (Hinton et al., 1984) – with TP, using only information that is local to each neuron, in a way that is fundamentally different from conventional BP. Our new direct feedback learning algorithm reinforces the belief that it is possible to optimize neural circuits without requiring the precise, layerwise symmetric feedback structure imposed by BP.

## Broader impact

Since the nature of our work is mostly theoretical with no immediate practical applications, we do not anticipate any direct societal impact. However, on the long-term, our work can have an impact on related research communities such as neuroscience or deep learning, which can have both positive and negative societal impact, depending on how these fields develop. For example, we show that the TP framework, when using a new reconstruction loss, is a viable credit assignment method for feedforward networks that fundamentally differs from the standard training method known as BP. Furthermore, TP only uses information which is local to each neuron and mitigates the weight transport and signed error transmission problem, the two major criticisms of BP. This renders TP appealing for neuroscientists that investigate how credit assignment is organized in the brain (Lillicrap et al., 2020; Richards et al., 2019) and how neural circuits (dys)function in health and disease. From a machine learning perspective, the TP framework has inspired new training methods for recurrent neural networks (RNNs) (Manchev and Spratling, 2020; Ororbia et al., 2020; DePasquale et al., 2018; Abbott et al., 2016), which is beneficial because the conventional backpropagation-through-time method (Werbos, 1988; Robinson and Fallside, 1987; Mozer, 1995) for training RNNs still suffers from significant drawbacks, such as vanishing and exploding gradients (Hochreiter, 1991; Hochreiter and Schmidhuber, 1997). Here, our work provides a new angle for the field to investigate the theoretical underpinnings of credit assignment in RNNs based on TP.

## Acknowledgements

This work was supported by the Swiss National Science Foundation (B.F.G. CRSII5-173721 and 315230_189251), ETH project funding (B.F.G. ETH-20 19-01), the Human Frontiers Science Program (RGY0072/2019) and funding from the Swiss Data Science Center (B.F.G, C17-18, J. v. O. P18-03). Johan Suykens acknowledges support of ERC Advanced Grant E-DUALITY (787960), KU Leuven C14/18/068, FWO G0A4917N, Ford KU Leuven Alliance KUL0076, Flemish Government AI Research Program, Leuven.AI Institute. João Sacramento was supported by an Ambizione grant (PZ00P3_186027) from the Swiss National Science Foundation. The authors would like to thank Nik Dennler for his help in the implementation of the experiments and William Podlaski for insightful discussions on credit assignment through inversion. João Sacramento further thanks Greg Wayne for inspiring discussions on perturbation-based credit assignment algorithms.

## Footnotes

[1]PyTorch implementation of all methods is available on `github.com/meulemansalex/theoretical_framework_for_target_propagation`

[2]We used new hyperparameter settings for all methods, optimized for minimizing the training loss.

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
