[Supplementary Material]

# Supplementary Material:
# A Theoretical Framework for Target Propagation.

**Alexander Meulemans[1], Francesco S. Carzaniga[1], Johan A.K. Suykens[2],
João Sacramento[1], Benjamin F. Grewe[1]**

[1]Institute of Neuroinformatics, University of Zürich and ETH Zürich
[2]ESAT-STADIUS, KU Leuven
`ameulema@ethz.ch`

## Table of Contents

## A   Proofs and extra information on theoretical results

In this section, we show all the theorems and proofs, and provide extra information and discussion if appropriate.

### A.1   Proofs section 3.1

**Condition S1** (Invertible networks)**.** *The feed-forward neural network has forward mappings $\boldsymbol{h}_i = f_i(\boldsymbol{h}_{i-1}) = s_i(W_i\boldsymbol{h}_{i-1})$, $i = 1,...,L$ where $s_i$ can be any differentiable, monotonically increasing and invertible element-wise function with domain and image equal to $\mathbb{R}$ and where $W_i$ can be any invertible matrix. The feedback functions for propagating the targets are the inverse of the forward mapping functions: $g_i(\hat{\boldsymbol{h}}_{i+1}) = f_{i+1}^{-1}(\hat{\boldsymbol{h}}_{i+1}) = W_{i+1}^{-1}s_{i+1}^{-1}(\hat{\boldsymbol{h}}_{i+1})$.*

**Lemma S1** (Lemma 1 in main manuscript)**.** *Assuming Condition S1 holds and the output target stepsize $\hat{\eta}$ is small compared to $\|\boldsymbol{h}_L\|_2/\|\boldsymbol{e}_L\|_2$, the target update $\Delta \boldsymbol{h}_i \triangleq \hat{\boldsymbol{h}}_i - \boldsymbol{h}_i$ can be approximated by*

$$\Delta \boldsymbol{h}_i \triangleq \hat{\boldsymbol{h}}_i - \boldsymbol{h}_i = -\hat{\eta}\left[\prod_{k=i}^{L-1} J_{f_{k+1}}^{-1}\right]\boldsymbol{e}_L + \mathcal{O}(\hat{\eta}^2) = -\hat{\eta}J_{\bar{f}_{i,L}}^{-1}\boldsymbol{e}_L + \mathcal{O}(\hat{\eta}^2), \qquad (14)$$

*with $\boldsymbol{e}_L = (\partial \mathcal{L}/\partial \boldsymbol{h}_L)^T$ evaluated at $\boldsymbol{h}_L$, $J_{f_{k+1}} = \partial f_{k+1}(\boldsymbol{h}_k)/\partial \boldsymbol{h}_k$ evaluated at $\boldsymbol{h}_k$ and $J_{\bar{f}_{i,L}} = \partial f_L(..(f_{i+1}(\boldsymbol{h}_i)))/\partial \boldsymbol{h}_i$ evaluated at $\boldsymbol{h}_i$.*

*Proof.* (*Proof rephrased from Lee et al. (2015).*) We prove this lemma by a series of first-order Taylor expansions and using the inverse function theorem. We start with a Taylor approximation for $\hat{\boldsymbol{h}}_{L-1}$ around $\boldsymbol{h}_{L-1}$.

$$\hat{\boldsymbol{h}}_{L-1} = g_{L-1}(\hat{\boldsymbol{h}}_L) = g_{L-1}(\boldsymbol{h}_L - \hat{\eta}\boldsymbol{e}_L) \tag{15}$$

$$= g_{L-1}(\boldsymbol{h}_L) - \hat{\eta}J_{g_{L-1}}\boldsymbol{e}_L + \mathcal{O}(\hat{\eta}^2) \tag{16}$$

$$= \boldsymbol{h}_{L-1} - \hat{\eta}J_{f_L}^{-1}\boldsymbol{e}_L + \mathcal{O}(\hat{\eta}^2) \tag{17}$$

with $J_{g_i}$ the Jacobian of $g_i$ with respect to $\boldsymbol{h}_{i+1}$, evaluated at $\boldsymbol{h}_{i+1}$ and $J_{f_i}$ the Jacobian of $f_i$ with respect to $\boldsymbol{h}_{i-1}$, evaluated at $\boldsymbol{h}_{i-1}$. For the last step, we used that $g_i$ is the perfect inverse of $f_{i+1}$ and the inverse function theorem. If we continue doing Taylor expansions for each layer, we reach a general expression for $\hat{\boldsymbol{h}}_i$:

$$\hat{\boldsymbol{h}}_i = \boldsymbol{h}_i - \hat{\eta}\left[\prod_{k=i}^{L-1} J_{f_{k+1}}^{-1}\right]\boldsymbol{e}_L + \mathcal{O}(\hat{\eta}^2) \tag{18}$$

The Jacobian $J_{\bar{f}_{i,L}}$ of the total forward mapping $\bar{f}_{i,L}$ from layer $i$ to $L$ is given by:

$$J_{\bar{f}_{i,L}} = \prod_{k=L}^{i+1} J_{f_k}. \tag{19}$$

As all $J_{f_k}$ are square and invertible (following from Condition S1), its inverse is given by

$$J_{\bar{f}_{i,L}}^{-1} = \prod_{k=i}^{L-1} J_{f_{k+1}}^{-1}. \tag{20}$$

Using the definition $\Delta\boldsymbol{h}_i \triangleq \hat{\boldsymbol{h}}_i - \boldsymbol{h}_i$ concludes the proof. $\qquad\square$

For proving Theorem 2 of the paper, we need to introduce first a lemma.

**Lemma S2.** *Consider a feed-forward neural network with as forward mapping function $\boldsymbol{h}_i = f_i(\boldsymbol{h}_{i-1}) = s_i(W_i\boldsymbol{h}_{i-1})$, $i = 1, ..., L$ where $s_i$ can be any differentiable element-wise function. Furthermore, assume a mini-batch size of 1 and a $L_2$ output loss function. Under these conditions, the Gauss-Newton optimization step for the layer activations, with a block-diagonal approximation of the Gauss-Newton curvature matrix with blocks equal to the layer sizes, is given by:*

$$\Delta\boldsymbol{h}_i = -J_i^\dagger\boldsymbol{e}_L, \quad i = 1, ..., L-1, \tag{21}$$

*with $J_i = \frac{\partial\boldsymbol{h}_L}{\partial\boldsymbol{h}_i}$ evaluated at $\boldsymbol{h}_i$, $J_i^\dagger$ its Moore-Penrose pseudo-inverse (Moore, 1920; Penrose, 1955) and $\boldsymbol{e}_L = \boldsymbol{h}_L - \boldsymbol{l}$ the output error, with $\boldsymbol{l}$ the output label.*

*Proof.* The output loss function $\mathcal{L}$ for a minibatch size of 1 can be written as:

$$\mathcal{L} = \frac{1}{2}\|\boldsymbol{e}_L\|_2^2 \tag{22}$$

$$\boldsymbol{e}_L \triangleq \frac{\partial L}{\partial\boldsymbol{h}_L} = \boldsymbol{h}_L - \boldsymbol{l}, \tag{23}$$

with $\boldsymbol{h}_L$ the output layer activation and $\boldsymbol{l}$ the output label. As an experiment of thought, imagine that the parameters of our network are the layer activations $\boldsymbol{h}_i$, $i = 1, ..., L$, concatenated in the total activation vector $\bar{\boldsymbol{h}}$, instead of the weights $W_i$. The weights can be seen as fixed values for now. If we want to update the activation values $\bar{\boldsymbol{h}}$ according to the Gauss-Newton method we get the following Gauss-Newton curvature matrix (which serves as an approximation of the Hessian of the output loss with respect to $\bar{\boldsymbol{h}}$):

$$G = \bar{J}^T\bar{J} \tag{24}$$

with $\bar{J}$ the Jacobian of $\boldsymbol{h}_L$ with respect to $\bar{\boldsymbol{h}}$. $\bar{J}$ can be structured in blocks along the column dimension:

$$\text{Block}_i(\bar{J}) = J_i = \frac{\partial\boldsymbol{h}_L}{\partial\boldsymbol{h}_i}, \quad i = 1, ..., L \tag{25}$$

Consequently, $G$ can also be structured in blocks of the form:

$$\text{Block}_{i,j}(G) = J_i^T J_j, \quad i,j = 1, ..., L \tag{26}$$

In the field of Gauss-Newton optimization for deep learning, it is common to approximate $G$ by a block-diagonal matrix $\tilde{G}$ (Martens and Grosse, 2015; Botev et al., 2017; Chen et al., 1990), as Martens and Grosse (2015) show that the GN Hessian matrix is block diagonal dominant for feed-forward neural networks. $\tilde{G}$ then consists of the following diagonal blocks:

$$\text{Block}_{i,i}(\tilde{G}) = J_i^T J_i, \quad i = 1, ..., L \tag{27}$$

while all off-diagonal blocks are zero. Now the following linear system has to be solved to compute the activations update $\Delta \bar{h}$:

$$\tilde{G} \Delta \bar{h} = -\bar{J}^T e_L, \tag{28}$$

As $\tilde{G}$ is block-diagonal, this system factorizes naturally in $L$ linear systems of the form

$$J_i^T J_i \Delta h_i = -J_i^T e_L, \quad i = 1, ..., L \tag{29}$$

$$\Leftrightarrow \Delta h_i = -J_i^\dagger e_L, \quad i = 1, ..., L \tag{30}$$

If $J_i^T J_i$ is not invertible, the Moore-Penrose pseudo-inverse gives the solution $\Delta h_i$ with the smallest norm, which is the most common choice in practice. $\square$

**Theorem S3** (Theorem 2 in main manuscript). *Consider an invertible network specified in Condition S1. Further, assume a mini-batch size of 1 and an $L_2$ output loss function. Under these conditions and in the limit of $\hat{\eta} \to 0$, TP uses Gauss-Newton optimization with a block-diagonal approximation of the Gauss-Newton curvature matrix, with block sizes equal to the layer sizes, to compute the local layer targets $\hat{h}_i$.*

**Proof Sketch.** *Our proof relies on the following insights. First, if the Gauss-Newton method is used to compute a single target update $\Delta h_i$ (while considering $h_i$ as the parameters to optimize) this results in $\Delta h_i = J_{\bar{f}_{i,L}}^\dagger e_L$. Second, due to the invertible network setting, the pseudo-inverse is equal to the real inverse and can be factorized over the layer Jacobians resembling Lemma S1. Finally, when all the target updates $\Delta h_i$, $i = 1, .., L$ are computed at once with GN, the block-diagonal approximation of the total curvature matrix ensures that each target update is computed independently from the others, such that the above interpretation still holds.*

*Proof.* Under the conditions assumed in this theorem, the Gauss-Newton optimization step for the layer activations, with a block-diagonal approximation of the Gauss-Newton Hessian matrix with blocks equal to the layer sizes, is given by (a result of Lemma S2):

$$\Delta h_i^{GN} = -J_i^\dagger e_L, \quad i = 1, ..., L-1, \tag{31}$$

with $J_i = \partial h_L / \partial h_i$. Under Condition S1, $J_i$ is square and invertible, hence the pseudo inverse $J_i^\dagger$ is equal to the real inverse $J_i^{-1}$. As a result of Lemma S1, the TP target update $\Delta h_i^{TP}$ is given by

$$\Delta h_i^{TP} = -\hat{\eta} J_i^{-1} e_L + \mathcal{O}(\hat{\eta}^2), \quad i = 1, ..., L-1, \tag{32}$$

We see that $\Delta h_i^{TP}$ is approximately equal to $\Delta h_i^{GN}$ with stepsize $\hat{\eta}$ with an error of $\mathcal{O}(\hat{\eta}^2)$. In the limit of $\hat{\eta} \to 0$ this error goes to zero, thereby proving the theorem. This proof is inspired by the work of Lillicrap et al. (2016), who discovered that feedback alignment for one-hidden layer networks is closely related to Gauss-Newton optimization.

$\square$

## A.2 Proofs and extra information for section 3.2

**Lemma S4** (Lemma 3 in main manuscript). *Assuming the output target step size $\hat{\eta}$ is small compared to $\|h_L\|_2 / \|e_L\|_2$, the target update $\Delta h_i \triangleq \hat{h}_i - h_i$ of the DTP method can be approximated by*

$$\Delta h_i \triangleq \hat{h}_i - h_i = -\hat{\eta} \left[ \prod_{k=i}^{L-1} J_{g_k} \right] e_L + \mathcal{O}(\hat{\eta}^2), \tag{33}$$

*with $J_{g_k} = \partial g_k(h_{k+1}) / \partial h_{k+1}$ evaluated at $h_{k+1}$.*

*Proof.* In DTP, the target for layer $i$ is computed as $\hat{h}_i = g_i(\hat{h}_{i+1}) + h_i - g_i(h_{i+1})$. A first-order Taylor approximation for $\hat{h}_{L-1}$ around $h_{L-1}$ gives

$$\hat{h}_{L-1} = g_{L-1}(\hat{h}_L) + h_{L-1} - g_{L-1}(h_L) \tag{34}$$

$$= g_{L-1}(h_L - \hat{\eta}e_L) + h_{L-1} - g_{L-1}(h_L) \tag{35}$$

$$= g_{L-1}(h_L) - \hat{\eta}J_{g_{L-1}}e_L + \mathcal{O}(\hat{\eta}^2) + h_{L-1} - g_{L-1}(h_L) \tag{36}$$

$$= h_{L-1} - \hat{\eta}J_{g_{L-1}}e_L + \mathcal{O}(\hat{\eta}^2) \tag{37}$$

with $J_{g_i}$ the Jacobian of $g_i$ with respect to $h_{i+1}$. A further series of first-order Taylor expansions until layer $i$ gives:

$$\hat{h}_i = h_i - \hat{\eta}\left[\prod_{k=i}^{L-1} J_{g_k}\right]e_L + \mathcal{O}(\hat{\eta}^2) \tag{38}$$

Using the definition $\Delta h_i \triangleq \hat{h}_i - h_i$ concludes the proof. $\qquad\square$

**Why doesn't DTP propagate GN targets?** As shown in Lemma S2, for propagating GN targets, the target updates should be equal to:

$$\Delta h_i^{GN} = -J_{\bar{f}_{i,L}}^\dagger e_L \tag{39}$$

with $J_{\bar{f}_{i,L}}$ the Jacobian of the forward mapping from layer $i$ to the output, evaluated at the batch sample $h_i$. In DTP, the feedback pathways $g_i$ are trained on the following layer-wise reconstruction loss (Lee et al., 2015):

$$\mathcal{L}_i^{\text{rec}} = \sum_{b=1}^{B} \left\| g_i\big(f_{i+1}(h_i^{(b)} + \sigma\epsilon)\big) - (h_i^{(b)} + \sigma\epsilon) \right\|_2^2 \tag{40}$$

with $B$ the minibatch size, $\epsilon$ white standard Gaussian noise and $\sigma$ the standard deviation of the noise. For clarity, let us assume that we have linear feedforward mappings $f_i(h_{i-1}) = W_i h_{i-1}$ and linear feedback mappings $g_i(h_{i+1}) = Q_i h_{i+1}$. Furthermore, we assume a batch-setting (i.e. the mini-batch size $B$ is equal to the total batch size). Then the GN target updates are given by:

$$\Delta h_i^{GN} = -J_{\bar{f}_{i,L}}^\dagger e_L = \left[\prod_{k=L}^{i+1} W_k\right]^\dagger e_L \tag{41}$$

Further, the minimum of (40) for $Q_i$ has a closed form solution.

$$Q_i^* \big(W_{i+1}\Gamma_i W_{i+1}^T\big) = \Gamma_i W_{i+1}^T \tag{42}$$

with $\Gamma_i = \sum_{b=1}^{B}(h_i^{(b)} + \sigma\epsilon)(h_i^{(b)} + \sigma\epsilon)^T$, which can be interpreted as a covariance matrix. For contracting networks (diminishing layer sizes) $W_{i+1}\Gamma_i W_{i+1}^T$ is usually invertible, leading to

$$Q_i^* = \Gamma_i W_{i+1}^T \big(W_{i+1}\Gamma_i W_{i+1}^T\big)^{-1} \tag{43}$$

When $\sigma$ is big relative to the norm of $h_i$ (Lee et al. (2015) and Bartunov et al. (2018) take $\sigma$ in the same order of magnitude as $\|h_i\|_2$), $(h_i^{(b)} + \sigma\epsilon)$ can be approximately seen as white noise and $\Gamma_i$ is thus close to a scalar multiple of the identity matrix. For $\Gamma_i = cI$ and in contracting networks, $Q_i$ converges to

$$Q_i^* = W_{i+1}^T \big(W_{i+1}W_{i+1}^T\big)^{-1} = W_{i+1}^\dagger \tag{44}$$

Following Lemma S4, the DTP target updates are given by (the approximation is exact in the linear case):

$$\Delta h_i^{DTP} = \prod_{k=i+1}^{L} W_k^\dagger e_L \tag{45}$$

Despite the resemblance of (41) and (45), $\Delta h_i^{DTP}$ is not equal to the Gauss-Newton target update $\Delta h_i^{GN}$, because in general, $\left[\prod_{k=L}^{i+1} W_k\right]^\dagger$ cannot be factorized as $\prod_{k=i+1}^{L} W_k^\dagger$. The pseudo-inverse of a product of matrices $(AB)^\dagger$ can only be factorized as $B^\dagger A^\dagger$ if one or more of the following conditions hold (Campbell and Meyer, 2009):

- $A$ has orthonormal columns
- $B$ has orthonormal rows
- $B = A^*$ ($B$ is the conjugate transpose of $A$)
- $A$ has all columns linearly independent and $B$ has all rows linearly independent

In general, the weight matrices do not satisfy any of these conditions for a neural network with arbitrary architecture. In section A.4 we show that $\Delta \boldsymbol{h}_i^{DTP}$ leads to inefficient forward parameter updates. In the nonlinear case, it is in general not true that the Jacobian $J_{g_i}^{(b)}$ of the feedback mappings will be a good approximation of $J_{f_{i+1}}^{(b)\dagger}$, pushing $\Delta \boldsymbol{h}_i^{DTP}$ even further away from $\Delta \boldsymbol{h}_i^{GN}$.

## A.3 Proofs and extra information for section 3.3

(a) Reconstruction loss for DTP

1 **Perturb:** $\tilde{\boldsymbol{h}}_i = \boldsymbol{h}_i + \sigma \boldsymbol{\epsilon}$

2 **Propagate forward:** $\tilde{\boldsymbol{h}}_{i+1} = f_{i+1}(\tilde{\boldsymbol{h}}_i)$

3 **Reconstruct:** $\bar{\boldsymbol{h}}_i = g_i(\tilde{\boldsymbol{h}}_{i+1})$

4 **Loss:** $\mathcal{L}_i^{\mathrm{rec}} = \|\bar{\boldsymbol{h}}_i - \tilde{\boldsymbol{h}}_i\|_2^2$

(b) Difference Reconstruction Loss

1 **Perturb:** $\tilde{\boldsymbol{h}}_i = \boldsymbol{h}_i + \sigma \boldsymbol{\epsilon}$

2 **Propagate forward:** $\bar{\boldsymbol{h}}_L = \tilde{\boldsymbol{h}}_L = \bar{f}_{i,L}(\tilde{\boldsymbol{h}}_i)$

3 **Reconstruct:** $\bar{\boldsymbol{h}}_i = g_i(\bar{\boldsymbol{h}}_{i+1}) + \boldsymbol{h}_i - g_i(\boldsymbol{h}_{i+1})$

4 **Loss:** $\mathcal{L}_i^{\mathrm{rec,diff}} = \|\bar{\boldsymbol{h}}_i - \tilde{\boldsymbol{h}}_i\|_2^2 + \mathcal{R}$

Figure S1: Schematic of the reconstruction loss used in DTP and our new Difference Reconstruction Loss (DRL). The reconstruction loop is outlined in red. $\mathcal{R}$ indicates a regularization term, for which weight decay on the feedback parameters is used in practice. Note that we visualized the practical format of DRL, where the expectation is replaced by a single sample. If more samples are wished, the procedure is repeated $n$ times with the same batch sample but different independent samples of $\boldsymbol{\epsilon}$ and the loss is averaged in the end.

**Theorem S5** (Theorem 4 in main manuscript). *The difference reconstruction loss for layer $i$, with $\sigma$ driven in limit to zero, is equal to*

$$\lim_{\sigma \to 0} \mathcal{L}_i^{rec,diff} = \sum_{b=1}^{B} \mathbb{E}_{\boldsymbol{\epsilon}_1 \sim \mathcal{N}(0,1)} \left[ \|J_{\bar{g}_{L,i}}^{(b)} J_{\bar{f}_{i,L}}^{(b)} \boldsymbol{\epsilon}_1 - \boldsymbol{\epsilon}_1\|_2^2 \right] + \mathbb{E}_{\boldsymbol{\epsilon}_2 \sim \mathcal{N}(0,1)} \left[ \lambda \|J_{\bar{g}_{L,i}}^{(b)} \boldsymbol{\epsilon}_2\|_2^2 \right], \quad (46)$$

*with $J_{\bar{g}_{L,i}}^{(b)} = \prod_{k=i}^{L-1} J_{g_k}^{(b)}$ the Jacobian of feedback mapping function $\bar{g}_{L,i}$ evaluated at $\boldsymbol{h}_k^{(b)}$, $k = i+1, .., L$, and $J_{\bar{f}_{i,L}}^{(b)}$ the Jacobian of the forward mapping function $\bar{f}_{i,L}$ evaluated at $\boldsymbol{h}_i^{(b)}$. The minimum of $\lim_{\sigma \to 0} \mathcal{L}_i^{rec,diff}$ is reached if for each batch sample holds that*

$$J_{\bar{g}_{L,i}}^{(b)} = \prod_{k=i}^{L-1} J_{g_k}^{(b)} = J_{\bar{f}_{i,L}}^{(b)T} \left( J_{\bar{f}_{i,L}}^{(b)} J_{\bar{f}_{i,L}}^{(b)T} + \lambda I \right)^{-1}. \quad (47)$$

*When the regularization parameter $\lambda$ is driven in limit to zero, this results in $J_{\bar{g}_{L,i}}^{(b)} = J_{\bar{f}_{i,L}}^{(b)\dagger}$.*

*Proof.* The proof consists of two main parts. First, we show how $\mathcal{L}_i^{\text{rec,diff}}$ looks like in the limit of $\sigma \to 0$. Second, we investigate the minimum of $\mathcal{L}_i^{\text{rec,diff}}$.

For proving the first part, we do the following first-order Taylor expansions.

$$\bar{f}_{i,L}(\boldsymbol{h}_i^{(b)} + \sigma\boldsymbol{\epsilon}_1) = \boldsymbol{h}_L^{(b)} + \sigma J_{\bar{f}_{i,L}}^{(b)} \boldsymbol{\epsilon}_1 + \mathcal{O}(\sigma^2) \tag{48}$$

$$\bar{g}_{L,i}^{\text{diff}}(\bar{f}_{i,L}(\boldsymbol{h}_i^{(b)} + \sigma\boldsymbol{\epsilon}_1), \boldsymbol{h}_L^{(b)}, \boldsymbol{h}_i^{(b)}) = \bar{g}_{L,i}(\bar{f}_{i,L}(\boldsymbol{h}_i^{(b)} + \sigma\boldsymbol{\epsilon}_1)) - \bar{g}_{L,i}(\boldsymbol{h}_L^{(b)}) + \boldsymbol{h}_i^{(b)} \tag{49}$$

$$= \sigma J_{\bar{g}_{L,i}}^{(b)} J_{\bar{f}_{i,L}}^{(b)} \boldsymbol{\epsilon}_1 + \boldsymbol{h}_i^{(b)} + \mathcal{O}(\sigma^2) \tag{50}$$

$$\bar{g}_{L,i}^{\text{diff}}(\boldsymbol{h}_L^{(b)} + \sigma\boldsymbol{\epsilon}_2, \boldsymbol{h}_L^{(b)}, \boldsymbol{h}_i^{(b)}) = \bar{g}_{L,i}(\boldsymbol{h}_L^{(b)} + \sigma\boldsymbol{\epsilon}_2) - \bar{g}_{L,i}(\boldsymbol{h}_L^{(b)}) + \boldsymbol{h}_i^{(b)} \tag{51}$$

$$= \sigma J_{\bar{g}_{L,i}}^{(b)} \boldsymbol{\epsilon}_2 + \boldsymbol{h}_i^{(b)} + \mathcal{O}(\sigma^2) \tag{52}$$

with $J_{\bar{g}_{L,i}}^{(b)} = \prod_{k=i}^{L-1} J_{\bar{g}_k}^{(b)}$ the Jacobian of feedback mapping function $\bar{g}_{L,i}$ with respect to $\boldsymbol{h}_L$ evaluated at $\boldsymbol{h}_k^{(b)}$, $k = i+1, .., L$, and $J_{\bar{f}_{i,L}}^{(b)}$ the Jacobian of the forward mapping function $\bar{f}_{i,L}^{(b)}$ evaluated at $\boldsymbol{h}_i^{(b)}$. Filling these Taylor approximations into the definition of $\mathcal{L}_i^{\text{rec,diff}}$ and taking the limit of $\sigma \to 0$ concludes the first part of the proof.

$$\lim_{\sigma \to 0} \mathcal{L}_i^{\text{rec,diff}} = \sum_{b=1}^{B} \mathop{\mathbb{E}}_{\boldsymbol{\epsilon}_1 \sim \mathcal{N}(0,1)} \left[ \| J_{\bar{g}_{L,i}}^{(b)} J_{\bar{f}_{i,L}}^{(b)} \boldsymbol{\epsilon}_1 - \boldsymbol{\epsilon}_1 \|_2^2 \right] + \mathop{\mathbb{E}}_{\boldsymbol{\epsilon}_2 \sim \mathcal{N}(0,1)} \left[ \lambda \| J_{\bar{g}_{L,i}}^{(b)} \boldsymbol{\epsilon}_2 \|_2^2 \right], \tag{53}$$

We see that $\mathcal{L}_i^{\text{rec,diff}}$ is a sum of positive terms. The absolute minimum of $\mathcal{L}_i^{\text{rec,diff}}$ can thus be reached if for each batch sample $b$, $J_{\bar{g}_{L,i}}^{(b)}$ is independent from $J_{\bar{g}_{L,i}}^{(c \neq b)}$ and when we minimize each positive term separately. When $\bar{g}_{L,i}$ is parameterized by a finite amount of parameters, $J_{\bar{g}_{L,i}}^{(b)}$ is not independent from $J_{\bar{g}_{L,i}}^{(c \neq b)}$ and we will discuss the implications of this assumption below this proof. Each positive term in $\lim_{\sigma \to 0} \mathcal{L}_i^{\text{rec,diff}}$ is given by

$$\lim_{\sigma \to 0} \mathcal{L}_i^{\text{rec,diff},(b)} = \mathop{\mathbb{E}}_{\boldsymbol{\epsilon}_1 \sim \mathcal{N}(0,1)} \left[ \| J_{\bar{g}_{L,i}}^{(b)} J_{\bar{f}_{i,L}}^{(b)} \boldsymbol{\epsilon}_1 - \boldsymbol{\epsilon}_1 \|_2^2 \right] + \mathop{\mathbb{E}}_{\boldsymbol{\epsilon}_2 \sim \mathcal{N}(0,1)} \left[ \lambda \| J_{\bar{g}_{L,i}}^{(b)} \boldsymbol{\epsilon}_2 \|_2^2 \right] \tag{54}$$

for which we introduce the following shorthand notation:

$$\lim_{\sigma \to 0} \mathcal{L}_i^{\text{rec,diff},(b)} = \mathop{\mathbb{E}}_{\boldsymbol{\epsilon}_1 \sim \mathcal{N}(0,1)} \left[ \| J_g J_f \boldsymbol{\epsilon}_1 - \boldsymbol{\epsilon}_1 \|_2^2 \right] + \mathop{\mathbb{E}}_{\boldsymbol{\epsilon}_2 \sim \mathcal{N}(0,1)} \left[ \lambda \| J_g \boldsymbol{\epsilon}_2 \|_2^2 \right] \tag{55}$$

Next, we seek an expression for its gradient with respect to $J_g$. For that, we rewrite equation (55) with traces and use the fact that $\boldsymbol{\epsilon}_i$ are white noise.

$$\lim_{\sigma \to 0} \mathcal{L}_i^{\text{rec,diff},(b)} = \mathop{\mathbb{E}}_{\boldsymbol{\epsilon}_1 \sim \mathcal{N}(0,1)} \left[ \text{Tr}\left(\boldsymbol{\epsilon}_1 \boldsymbol{\epsilon}_1^T\right) + \text{Tr}\left(J_g J_f \boldsymbol{\epsilon}_1 \boldsymbol{\epsilon}_1^T J_f^T J_g^T\right) - 2\text{Tr}\left(J_g J_f \boldsymbol{\epsilon}_1 \boldsymbol{\epsilon}_1^T\right) \right] + ..$$

$$.. \mathop{\mathbb{E}}_{\boldsymbol{\epsilon}_2 \sim \mathcal{N}(0,1)} \left[ \lambda \text{Tr}\left(J_g \boldsymbol{\epsilon}_2 \boldsymbol{\epsilon}_2^T J_g^T\right) \right] \tag{56}$$

$$= \text{Tr}\left(I\right) + \text{Tr}\left(J_g J_f J_f^T J_g^T\right) - 2\text{Tr}\left(J_g J_f\right) + \lambda \text{Tr}\left(J_g J_g^T\right) \tag{57}$$

The gradient with respect to $J_g$ is given by

$$\nabla_{J_g}\left(\lim_{\sigma \to 0} \mathcal{L}_i^{\text{rec,diff},(b)}\right) = 2J_g\left(J_f J_f^T + \lambda I\right) - 2J_f^T \tag{58}$$

Requiring that the gradient is zero gives the following optimality condition for $J_g$:

$$J_g^*\left(J_f J_f^T + \lambda I\right) = J_f^T \tag{59}$$

$$J_g^* = J_f^T\left(J_f J_f^T + \lambda I\right)^{-1}. \tag{60}$$

$\left(J_f J_f^T + \lambda I\right)$ is always invertible, as $J_f J_f^T$ is positive semi-definite and $\lambda$ is strictly positive. Taking the limit of $\lambda \to 0$, this results in

$$\lim_{\lambda \to 0} J_g^* = J_f^\dagger \tag{61}$$

This last step follows from the definition of the Moore-Penrose pseudo-inverse (Campbell and Meyer, 2009) and can be easily seen by taking the singular value decomposition of $J_f$ $\qquad\square$

**The minimum of DRL in a parameterized setting.** Theorem S5 showed the absolute minimum of the difference reconstruction loss and proved that it is reached when for each batch sample holds that

$$J_{\bar{g}_{L,i}}^{(b)} = \prod_{k=i}^{L-1} J_{g_k}^{(b)} = J_{\bar{f}_{i,L}}^{(b)T}\big(J_{\bar{f}_{i,L}}^{(b)} J_{\bar{f}_{i,L}}^{(b)T} + \lambda I\big)^{-1} \tag{62}$$

However, Theorem S5 did not cover whether this absolute minimum will be reached by an actual parameterized feedback path $g_i$. Two things prevent the feedback paths from reaching the exact absolute minimum of DRL.

First, $J_{\bar{g}_{L,i}}^{(b)}$, for different samples $b$, will depend on the same limited amount of parameters of $g_i$. Hence, $J_{\bar{g}_{L,i}}^{(b)}$ cannot be treated independently from $J_{\bar{g}_{L,i}}^{(c \neq b)}$ and a parameter setting will be sought that brings $J_{\bar{g}_{L,i}}^{(b)}$ as close as possible to $J_{\bar{f}_{i,L}}^{(b)\dagger}$ for all batch samples $b$, but they will in general not be equal for each $b$. The more parameters $g_i$ has, the more capacity it has to approximate different $J_{\bar{g}_{L,i}}^{(b)}$ for different batch samples and hence the lower DRL it will reach after convergence.

Second, in contracting networks (diminishing layer sizes), a second issue occurs. Since the network is contracting, different samples of $\boldsymbol{h}_i^{(b)}$ can map to the same output and $\boldsymbol{h}_{k>i}^{(b)}$. Hence, different $J_{\bar{f}_{i,L}}^{(b)}$ will correspond to the same $J_{\bar{g}_{L,i}}^{(b)}$, as $J_{\bar{g}_{L,i}}^{(b)}$ is solely evaluated at $\boldsymbol{h}_{k>i}^{(b)}$. Therefore, it is impossible for $J_{g_i}^{(b)}$ to approximate $J_{\bar{f}_{i,L}}^{(b)\dagger}$ for all $b$. This is especially troubling for direct feedback connections, as then $J_{\bar{g}_{L,i}}^{(b)}$ is solely evaluated at the output value, which is often of much lower dimension than the hidden layers. We can alleviate this issue by drawing inspiration from the synthetic gradient framework ((Jaderberg et al., 2017)). When we provide $\boldsymbol{h}_i$ as an extra input to $g_i(\hat{\boldsymbol{h}}_L, \boldsymbol{h}_i)$, the (direct) feedback connections can use this extra information to compute the hidden layer target or to reconstruct the corrupted hidden layer activation $\boldsymbol{h}_i + \sigma\boldsymbol{\epsilon}$, thereby making it possible to let all $J_{\bar{f}_{i,L}}^{(b)}$ correspond to a different $J_{\bar{g}_{L,i}}^{(b)}$. As a concrete example, we can adjust the DDTP-RHL method from the experimental section to incorporate these recurrent feedback connections. This new *DDTP-RHL(rec)* method is shown in figure S2. In the DDTP-RHL method without recurrent feedback connections, the feedback path $g_i$ is parameterized as

$$g_i(\boldsymbol{h}_L) = \tanh\big(Q_i \tanh(R\boldsymbol{h}_L)\big) \tag{63}$$

If we add the recurrent feedback connections, this results in

$$g_i^{\text{rec}}(\boldsymbol{h}_L, \boldsymbol{h}_i) = \tanh\big(Q_i \tanh(R\boldsymbol{h}_L) + S_i\boldsymbol{h}_i\big) \tag{64}$$

More specific, for propagating targets this would result in

$$g_i^{\text{rec}}(\hat{\boldsymbol{h}}_L, \boldsymbol{h}_i) = \tanh\big(Q_i \tanh(R\hat{\boldsymbol{h}}_L) + S_i\boldsymbol{h}_i\big) \tag{65}$$

In the context of the DRL, this results in

$$g_i^{\text{rec}}\big(\bar{f}_{i,L}(\boldsymbol{h}_i + \sigma\boldsymbol{\epsilon}), \boldsymbol{h}_i\big) = \tanh\Big(Q_i \tanh\big(R\bar{f}_{i,L}(\boldsymbol{h}_i + \sigma\boldsymbol{\epsilon})\big) + S_i\boldsymbol{h}_i\Big) \tag{66}$$

Note that the non-corrupted sample $\boldsymbol{h}_i$ is used in the recurrent feedback connection.

**Approximating Tikhonov regularization by weight decay.** The difference reconstruction loss (DRL) introduces a Tikhonov regularization term by corrupting the output activation with white noise and propagating this output backward:

$$\mathcal{R}_i = \frac{1}{\sigma^2} \sum_{b=1}^{B} \mathbb{E}_{\boldsymbol{\epsilon}_2 \sim \mathcal{N}(0,1)} \Big[\lambda \|\bar{g}_{L,i}^{\text{diff}}(\boldsymbol{h}_L^{(b)} + \sigma\boldsymbol{\epsilon}_2, \boldsymbol{h}_L^{(b)}, \boldsymbol{h}_i^{(b)}) - \boldsymbol{h}_i^{(b)}\|_2^2\Big] \tag{67}$$

However, from a practical side, we would like to remove the need for a second noise source $\boldsymbol{\epsilon}_2$ on the output, as this introduces the need for an extra separate phase for training the feedback parameters, because $\boldsymbol{\epsilon}_2$ may not interfere with $\boldsymbol{\epsilon}_1$. Hence, we show that this regularization term can

Figure S2: Schematic of the DDTP-RHL method with recurrent feedback connections.

be approximated by simple weight decay on the feedback parameters, removing the need for an extra phase. From Theorem S5 follows that, for $\sigma \to 0$, this regularizer term can be written as

$$\lim_{\sigma \to 0} \mathcal{R}_i = \sum_{b=1}^{B} \|J_{\tilde{g}_{L,i}}^{(b)}\|_F^2 \tag{68}$$

with $\|.\|_F$ the Frobenius norm. If $g_i$ is parameterized as $t_i(Q_i \hat{\boldsymbol{h}}_{i+1})$, and using the fact the Frobenius norm is submultiplicative (following from the Cauchy-Schwarz inequality), we can define the following upperbound for $\mathcal{R}_i$.

$$\lim_{\sigma \to 0} \mathcal{R}_i = \sum_{b=1}^{B} \| \prod_{k=i}^{L-1} D_{t_k}^{(b)} Q_k \|_F^2 \tag{69}$$

$$\leq \sum_{b=1}^{B} \prod_{k=i}^{L-1} \|D_{t_k}^{(b)}\|_F^2 \|Q_k\|_F^2 \tag{70}$$

$$\leq B \prod_{k=i}^{L-1} n_i \|Q_k\|_F^2 \tag{71}$$

with $D_{t_k}^b$ the diagonal matrix with the derivatives of $t_i$, evaluated at $Q_i \boldsymbol{h}_{i+1}^{(b)}$ and $n_i$ the size of layer $i$. For the last step, we used the assumption that the derivative of $t_i$ lies in the interval $[0; 1]$, which is true for the conventional non-linearities used in deep learning. Applying weight decay results in adding the following regularizer term to the DRL:

$$\mathcal{R}^{\text{wd}} = \sum_{k=1}^{L-1} \|Q_k\|_F^2 \tag{72}$$

Hence, we see that weight decay penalizes the same terms $\|Q_k\|_F^2$ appearing in the upper bound of $\mathcal{R}_i$. The product is replaced by a sum, so the interactions between $\|Q_k\|_F^2$ will be different for $\mathcal{R}^{\text{wd}}$, however, a similar regularizing effect can be expected, as it puts pressure on the same norms $\|Q_k\|_F^2$. Note that for linear direct feedback connections (corresponding to DDTP-linear in the experiments), $J_{\tilde{g}_{L,i}}^{(b)} = Q_i$, hence $\mathcal{R}_i$ can be replaced exactly by $\mathcal{R}^{\text{wd}}$.

**Interpretation of DRL in a noisy environment.** The difference reconstruction loss is based on a noise-corrupted sample that is propagated forward to the network output and afterwards backpropagated to layer $i$ again, with a reconstruction correction based on the non-corrupted layer activations $\boldsymbol{h}_i$. As example, consider the DRL for layer $L-1$ (without the regularization term for simplicity, as we approximate it in practice by weight decay).

$$\mathcal{L}_{L-1}^{\text{rec,diff}} = \frac{1}{\sigma^2} \sum_{b=1}^{B} \mathop{\mathbb{E}}_{\boldsymbol{\epsilon} \sim \mathcal{N}(0,1)} \left[ \|g_{L-1}\big(f_L(\tilde{\boldsymbol{h}}_{L-1}^{(b)})\big) + \boldsymbol{h}_{L-1}^{(b)} - g_{L-1}(\boldsymbol{h}_L^{(b)}) - (\tilde{\boldsymbol{h}}_{L-1}^{(b)})\|_2^2 \right] \tag{73}$$

with $\tilde{\boldsymbol{h}}_{L-1} = \boldsymbol{h}_{L-1} + \sigma\boldsymbol{\epsilon}$. Hence, the difference reconstruction loss needs both the non-corrupted activations $\boldsymbol{h}_{L-1}$ and $\boldsymbol{h}_L$, and the corrupted activations $\tilde{\boldsymbol{h}}_{L-1}$ and $f_L(\tilde{\boldsymbol{h}}_{L-1})$. At first sight, it seems dubious whether a biological neuron could keep track of its noise-corrupted and non-corrupted activation at the same time. However, for small $\sigma$, the non-corrupted activations are approximately equal to the time average of the noise-corrupted activations. For $\boldsymbol{h}_{L-1}$ it follows directly from the fact that $\boldsymbol{\epsilon}$ is white noise.

$$\underset{\boldsymbol{\epsilon} \sim \mathcal{N}(0,1)}{\mathbb{E}} \left[ \boldsymbol{h}_{L-1} + \sigma\boldsymbol{\epsilon} \right] = \boldsymbol{h}_{L-1} \tag{74}$$

For small $\sigma$, we can do a first-order Taylor approximation for $f_L(\tilde{\boldsymbol{h}}_{L-1})$.

$$\underset{\boldsymbol{\epsilon} \sim \mathcal{N}(0,1)}{\mathbb{E}} \left[ f_L(\tilde{\boldsymbol{h}}_{L-1}) \right] = \underset{\boldsymbol{\epsilon} \sim \mathcal{N}(0,1)}{\mathbb{E}} \left[ f_L(\boldsymbol{h}_{L-1}) + \sigma J_{f_L}\boldsymbol{\epsilon} + \mathcal{O}(\sigma^2) \right] \approx \boldsymbol{h}_L \tag{75}$$

Hence, the time averages can be used for the reconstruction correction term $\boldsymbol{h}_{L-1} - g_{L-1}(\boldsymbol{h}_L)$, such that the neurons only need to represent the noise-corrupted activations, if there exists a mechanism to keep track of the time average of its activations. Similar arguments hold for $\mathcal{L}_i^{\text{rec,diff}}$ of the other layers of the network. Similar to Lee et al. (2015), Akrout et al. (2019) and Kunin et al. (2020), a single layer $i$ needs to be noise-corrupted with additive white noise, while all other layers cannot have additive noise of their own, for training the feedback parameters of layer $i$. Lee et al. (2015) Akrout et al. (2019) and Kunin et al. (2020) allow for training the feedback parameters of half of the network layers simultaneously, while keeping the other layers noise-free, which is less restrictive, but still needs the same fundamental mechanism to keep certain layers noise-free. This is a serious biological constraint, and future research should aim to adjust our difference reconstruction loss such that it can operate in an environment where all layers are noisy at the same time, while keeping its favorable properties.

**Feedback weight updates.** From the biological perspective, it is interesting to know which feedback parameter updates result from the difference reconstruction loss. We consider the difference reconstruction loss with weight decay as regularizer, as this is the version we use in the practical implementation of the methods. Furthermore we use a single noise sample $\boldsymbol{\epsilon}$ to approximate the expectation, as is also done in our practical implementation. For simplicity, we consider the DDTP-linear method, which has direct linear feedback connections $Q_i$. In this case, the difference reconstruction loss for a minibatch size of 1 is defined as

$$\mathcal{L}_i^{\text{rec,diff}} = \frac{1}{\sigma^2} \| Q_i \bar{f}_{i,L}(\tilde{\boldsymbol{h}}_i) - Q_i \bar{f}_{i,L}(\tilde{\boldsymbol{h}}_i) + \boldsymbol{h}_i - \tilde{\boldsymbol{h}}_i \|_2^2 + \lambda \| Q_i \|_F^2 \tag{76}$$

with $\tilde{\boldsymbol{h}}_i = \boldsymbol{h}_i + \sigma\boldsymbol{\epsilon}$. The resulting parameter update is given by

$$\Delta Q_i = -\frac{\eta}{2} \frac{\partial \mathcal{L}_i^{\text{rec,diff}}}{\partial Q_i} = -\eta \big( \tilde{\boldsymbol{h}}_i^{\text{fb}} - \boldsymbol{h}_i^{\text{fb}} - (\tilde{\boldsymbol{h}}_i - \boldsymbol{h}_i) \big) (\tilde{\boldsymbol{h}}_L - \boldsymbol{h}_L)^T - \eta\lambda Q_i \tag{77}$$

with $\tilde{\boldsymbol{h}}_L = \bar{f}_{i,L}(\tilde{\boldsymbol{h}}_i)$, $\tilde{\boldsymbol{h}}_i^{\text{fb}} = Q_i \bar{f}_{i,L}(\tilde{\boldsymbol{h}}_i)$ and $\boldsymbol{h}_i^{\text{fb}} = Q_i \bar{f}_{i,L}(\tilde{\boldsymbol{h}}_i)$. We see that the parameter update is fully local to the individual neurons of each layer. The parameter consists of three differences between a noisy activation (e.g. $\tilde{\boldsymbol{h}}_i$) and a time-average or base activation (e.g. $\boldsymbol{h}_i$). $\boldsymbol{h}_i^{\text{fb}}$ and $\tilde{\boldsymbol{h}}_i^{\text{fb}}$ could represent the activations of a separate feedback neuron or a segregated dendritic compartment of the considered neuron (Sacramento et al., 2018; Guerguiev et al., 2017). $\tilde{\boldsymbol{h}}_i$ and $\boldsymbol{h}_i$ could represent the activation of the considered neuron, or the activation of a segregated dendritic compartment of the considered neuron. Finally, $\tilde{\boldsymbol{h}}_L$ and $\boldsymbol{h}_L$ represent the pre-synaptic activation of the feedback weights $Q_i$. Similar interpretations holds for other parameterizations of the feedback functions $g_i$.

### A.4 Proofs and extra information for section 3.4

**Condition S2** (Gauss-Newton Target method). *The network is trained by GN targets: each hidden layer target is computed by*

$$\Delta \boldsymbol{h}_i^{(b)} \triangleq \hat{\boldsymbol{h}}_i^{(b)} - \boldsymbol{h}_i^{(b)} = -\hat{\eta} J_{\bar{f}_{i,L}}^{(b)\dagger} \boldsymbol{e}_L^{(b)}, \tag{78}$$

*after which the network parameters of each layer $i$ are updated by a gradient descent step on its corresponding local mini-batch loss $\mathcal{L}_i = \sum_b \| \Delta \boldsymbol{h}_i^{(b)} \|_2^2$, while considering $\hat{\boldsymbol{h}}_i$ fixed.*

**Theorem S6** (Theorem 5 in main manuscript). *Consider a contracting linear multilayer network ($n_1 \geq n_2 \geq .. \geq n_L$) trained by GNT according to Condition S2. For a mini-batch size of 1, the parameter updates $\Delta W_i$ are minimum-norm updates under the constraint that $\boldsymbol{h}_L^{(m+1)} = \boldsymbol{h}_L^{(m)} - c^{(m)}\boldsymbol{e}_L^{(m)}$ and when $\Delta W_i$ is considered independent from $\Delta W_{j \neq i}$, with $c^{(m)}$ a positive scalar and $m$ indicating the iteration.*

*Proof.* Let $J_i = \partial \boldsymbol{h}_{i+1}/\partial \boldsymbol{h}_i$ and $\hat{J}_i = J_{L-1}\ldots J_i$, then $\Delta W_i = -\eta (J_{L-1}\ldots J_i)^{\dagger}\boldsymbol{e}_L \boldsymbol{h}_{i-1}^T = -\eta \hat{J}_i^{\dagger}\boldsymbol{e}_L \boldsymbol{h}_{i-1}^T$. In a linear network the GNT method produces the following dynamic:

$$\boldsymbol{h}_L^{(m+1)} = \sum_{i=1}^{L} \hat{J}_i(W_i + \Delta W_i)\boldsymbol{h}_{i-1}^{(m)} = \sum_{i=1}^{L} \hat{J}_i W_i \boldsymbol{h}_{i-1}^{(m)} + \sum_{i=1}^{L} \hat{J}_i \Delta W_i \boldsymbol{h}_{i-1}^{(m)} \tag{79}$$

$$= \boldsymbol{h}_L^{(m)} - \eta \sum_{i=1}^{L} \hat{J}_i \hat{J}_i^{\dagger}\boldsymbol{e}_L^{(m)}\boldsymbol{h}_{i-1}^{(m)T}\boldsymbol{h}_{i-1}^{(m)} = \boldsymbol{h}_L^{(m)} - \eta \boldsymbol{e}_L^{(m)} \sum_{i=1}^{L} \left\|\boldsymbol{h}_{i-1}^{(m)}\right\|_2^2 \tag{80}$$

Letting $c^{(m)} = \eta \sum_{i=1}^{L-1} \left\|\boldsymbol{h}_i^{(m)}\right\|_2^2$ we get

$$\boldsymbol{h}_L^{(m+1)} = \boldsymbol{h}_L^{(m)} - c^{(m)}\boldsymbol{e}_L^{(m)} \tag{81}$$

To show that this update represents the minimum weight choice we solve the following optimization problem:

$$\underset{\Delta W_i}{\arg\min} \qquad \sum_{i=1}^{L} \|\Delta W_i\|_F^2 \tag{82}$$

$$\text{s.t.} \qquad \boldsymbol{h}_L^{(m+1)} = \boldsymbol{h}_L^{(m)} - c^{(m)}\boldsymbol{e}_L^{(m)} \tag{83}$$

If $\Delta W_i$ is considered independent of $\Delta W_{j \neq i}$ the above minimization problem can be split in $L$ optimization problems:

$$\underset{\Delta W_i}{\arg\min} \qquad \|\Delta W_i\|_F^2 \tag{84}$$

$$\text{s.t.} \qquad \hat{J}_i(W_i + \Delta W_i)\boldsymbol{h}_{i-1}^{(m)} = \hat{J}_i W_i \boldsymbol{h}_{i-1}^{(m)} - c_i^{(m)}\boldsymbol{e}_L^{(m)} \tag{85}$$

$$\Longleftrightarrow \quad \hat{J}_i \Delta W_i \boldsymbol{h}_{i-1}^{(m)} = -c_i^{(m)}\boldsymbol{e}_L^{(m)}, \tag{86}$$

with $c^{(m)} = \sum_{i=1}^{L-1} c_i^{(m)}$ since we are working with a linear network. Note that the assumption that $\Delta W_i$ is independent of $\Delta W_{j \neq i}$ is similar to the block-diagonal approximation of the Gauss-Newton curvature matrix in Theorem S3. For ease of notation, we drop the iteration superscripts. We can now write the Lagrangian

$$\mathcal{L} = \|\Delta W_i\|_F^2 + \boldsymbol{\lambda}^T \left(\hat{J}_i \Delta W_i \boldsymbol{h}_{i-1} + c_i \boldsymbol{e}_L\right) \tag{87}$$

As this is a convex optimization problem, the optimal solution can be found by taking the derivatives of the Lagrangian (resulting in the Karush-Kuhn-Tucker conditions).

$$\frac{\partial \mathcal{L}}{\partial W_i} = 2\Delta W_i + \hat{J}_i^T \boldsymbol{\lambda}\boldsymbol{h}_{i-1}^T = 0 \implies \Delta W_i^* = -\frac{1}{2}\hat{J}_i^T \boldsymbol{\lambda}\boldsymbol{h}_{i-1}^T \tag{88}$$

$$\frac{\partial \mathcal{L}}{\partial \boldsymbol{\lambda}} = \hat{J}_i \Delta W_i \boldsymbol{h}_{i-1} + c_i \boldsymbol{e}_L = -\frac{1}{2}\hat{J}_i \hat{J}_i^T \boldsymbol{\lambda} \|\boldsymbol{h}_{i-1}\|_2^2 + c_i \boldsymbol{e}_L = 0 \tag{89}$$

$$\implies \boldsymbol{\lambda}^* = \frac{2c_i}{\|\boldsymbol{h}_{i-1}\|_2^2}(\hat{J}_i \hat{J}_i^T)^{-1}\boldsymbol{e}_L \tag{90}$$

Combining the two conditions

$$\Delta W_i^* = -\frac{c_i}{\|\boldsymbol{h}_{i-1}\|_2^2}\hat{J}_i^T(\hat{J}_i \hat{J}_i^T)^{-1}\boldsymbol{e}_L \boldsymbol{h}_{i-1}^T = -\frac{c_i}{\|\boldsymbol{h}_{i-1}\|_2^2}\hat{J}_i^{\dagger}\boldsymbol{e}_L \boldsymbol{h}_{i-1}^T = -\frac{c_i}{\|\boldsymbol{h}_{i-1}\|_2^2}\Delta W_i^{GNT} \tag{91}$$

We see that $\Delta W_i^{GNT}$ is a positive scalar multiple of the minimum-norm update $\Delta W_i^*$, thereby concluding the proof. $\qquad\square$

**DTP has inefficient updates.** In Theorem S6, we showed that GNT updates on linear networks push the output activation in the negative gradient direction and can be considered minimum-norm in doing so. In Corollary S6.1 below, we show that vanilla DTP updates also push the output activation in the negative gradient direction, but are not minimum-norm. This is due to the fact that $\Delta \vec{W}_i^{DTP}$ has components in the null-space of $\partial \boldsymbol{h}_L / \partial \vec{W}_i$. These null-space components do not contribute to any change in the output of the network and can thus be considered useless. Furthermore, these components will most likely interfere with the parameter updates of other training samples, thereby impairing the training process. The origin of these null-space components lies in the fact that the feedback parameters are trained by layer-wise reconstruction losses. Hence, under the right conditions, each target update $\Delta \boldsymbol{h}_i^{DTP}$ is minimum-norm in pushing the activation $\boldsymbol{h}_{i+1}$ of the layer on top towards its target $\hat{\boldsymbol{h}}_{i+1}$ (which is reflected in $Q_i^* = W_{i+1}^\dagger$ under the layer-wise reconstruction loss, as discussed in section A.2). However, this layer-wise minimum-norm characteristic does not translate to being minimum-norm for pushing the output towards its target. Indeed, a target update $\Delta \boldsymbol{h}_i^{DTP}$ that is minimum-norm for pushing $\boldsymbol{h}_{i+1}$ towards $\hat{\boldsymbol{h}}_{i+1}$ is not in general minimum-norm for pushing $\boldsymbol{h}_L$ towards $\hat{\boldsymbol{h}}_L$ because $\left[ \prod_{k=L}^{i+1} W_k \right]^\dagger$ is in general not factorizable as $\prod_{k=i+1}^{L} W_k^\dagger$. This result helps explaining why DTP cannot scale to more difficult datasets (Bartunov et al., 2018). The GNT updates in contrast, have no components in the null-space of $\partial \boldsymbol{h}_L / \partial \vec{W}_i$. For Corollary S6.1, we assumed that white noise is used for the layer-wise reconstruction losses. This assumption is approximately true in practical use-cases of DTP training (Lee et al., 2015; Bartunov et al., 2018), as the authors corrupt the layer activations $\boldsymbol{h}_i$ with white noise with a standard deviation in the same order of magnitude as the layer activations, for computing the reconstruction loss.

**Corollary S6.1.** *Consider a contracting linear multilayer network trained by DTP where white noise is used to train the feedback parameters. For a mini-batch size of 1, the resulting forward parameter update pushes the current output activation along the negative gradient direction $\boldsymbol{e}_L = \partial \mathcal{L} / \partial \boldsymbol{h}_L$ in the output space.*

$$\boldsymbol{h}_L^{(m+1)} = \boldsymbol{h}_L^{(m)} - c^{(m)} \boldsymbol{e}_L^{(m+1)}, \tag{92}$$

*with $c^{(m)}$ a positive scalar. Furthermore, $\|\Delta W_i^{DTP}\|_F \geq \|\Delta W_i^{GNT}\|_F$ for all layers $i$, with $\Delta W_i^{GNT}$ the parameter updates resulting from the GNT method as described in Condition S2.*

*Proof.* Call $\Delta W_i^{GNT} = (J_{L-1} \ldots J_{i+1})^\dagger \boldsymbol{e}_L^{(m)} \boldsymbol{h}_{i+1}^{(m)T}$ and $\Delta W_i^{DTP} = J_{i+1}^\dagger \ldots J_{L-1}^\dagger \boldsymbol{e}_L^{(m)} \boldsymbol{h}_{i+1}^{(m)T}$ the updates prescribed by GNT and DTP respectively, with $J_i \triangleq \partial \boldsymbol{h}_{i+1} / \partial \boldsymbol{h}_i = W_{i+1}$ (see section A.2 for a derivation of the DTP update for linear networks with white noise for feedback parameter training). The fact that DTP pushes the output activation along the negative gradient direction corresponding to equation (92) can be proved analogously to Theorem S6, as $(J_{L-1} \ldots J_{i+1})(J_{L-1} \ldots J_{i+1})^\dagger = (J_{L-1} \ldots J_{i+1})(J_{i+1}^\dagger \ldots J_{L-1}^\dagger) = I$ for contracting networks. For the second part of the proof, consider $\Delta W_i^{GNT} \boldsymbol{x} = \boldsymbol{g_r} + \boldsymbol{g_n}$ and $\Delta W_i^{DTP} \boldsymbol{x} = \boldsymbol{t_r} + \boldsymbol{t_n}$ with $\|x\|_2 = 1$ an arbitrary vector, where the vectors described by the index $r$ and $n$ are the projections onto the image and kernel of $(J_{L-1} \ldots J_{i+1})$ respectively. Since $im(A^\dagger) = im(A^T) = ker(A)^\perp$ we have that $\boldsymbol{g_n} = 0$. Since the network is contracting

$$(J_{L-1} \ldots J_{i+1})\boldsymbol{g_r} = (J_{L-1} \ldots J_{i+1})\Delta W_i^{GNT} \boldsymbol{x} = (J_{L-1} \ldots J_{i+1})\Delta W_i^{DTP} \boldsymbol{x} \tag{93}$$

$$= (J_{L-1} \ldots J_{i+1})(\boldsymbol{t_r} + \boldsymbol{t_n}) = (J_{L-1} \ldots J_{i+1})\boldsymbol{t_r} \tag{94}$$

For equation (93), we used the fact that $(J_{L-1} \ldots J_{i+1})(J_{L-1} \ldots J_{i+1})^\dagger = (J_{L-1} \ldots J_{i+1})(J_{i+1}^\dagger \ldots J_{L-1}^\dagger) = I$ for contracting networks. By definition, both $\boldsymbol{g_r}$ and $\boldsymbol{t_r}$ are not in the kernel of $(J_{L-1} \ldots J_{i+1})$, hence $\boldsymbol{g_r} = \boldsymbol{t_r}$. Finally by orthogonality of $\boldsymbol{t_r}$ and $\boldsymbol{t_n}$

$$\|\boldsymbol{t_r} + \boldsymbol{t_n}\|_2^2 = \|\boldsymbol{t_r}\|_2^2 + \|\boldsymbol{t_n}\|_2^2 = \|\boldsymbol{g_r}\|_2^2 + \|\boldsymbol{t_n}\|_2^2 \geq \|\boldsymbol{g_r}\|_2^2 \tag{95}$$

and therefore

$$\left\|\Delta W_i^{DTP}\right\|_2^2 \geq \left\|\Delta W_i^{GNT}\right\|_2^2 \tag{96}$$

The theorem follows by the equivalence of matrix norms.

$\square$

**Minimum-norm properties of Gauss-Newton in an over-parameterized setting.** In Theorem S6, we showed that the GNT method can be interpreted as finding the minimum-norm update $\Delta W_i$ under the constraint that the output activation should move in the negative gradient direction in the output space, as a result of the update. This result is closely connected to the properties of Gauss-Newton optimization in an over-parameterized setting. The Gauss-Newton parameter update for a nonlinear model parameterized by $\boldsymbol{\beta}$ is given by

$$\Delta \boldsymbol{\beta} = -J^{\dagger} \boldsymbol{e}_L, \tag{97}$$

with $J$ the Jacobian of the model outputs for each batch sample with respect to $\boldsymbol{\beta}$ and $\boldsymbol{e}_L$ the vector containing the output error for each batch sample. As reviewed in section C, this update can be seen as a linear least-squares regression with $J$ as the design matrix and $\boldsymbol{e}_L$ as the target values. However, this interpretation is only valid for under-parameterized models (fewer model parameters compared to the number of entries in $\boldsymbol{e}_L$), which is the usual setting for doing Gauss-Newton optimization. Theorem S6, however, operates in an over-parameterized setting, as we have many more layer weights $W_i$ than output errors $\boldsymbol{e}_L$ for a minibatch size of 1 in contracting networks. In this case, there are many possible parameter updates $\Delta \boldsymbol{\beta}$ that exactly reduce the error $\boldsymbol{e}_L$ to zero for the linearized model in the current mini-batch. A sensible way to resolve this indefiniteness is to take the minimum-norm update $\Delta \boldsymbol{\beta}$ that exactly reduces the error $\boldsymbol{e}_L$ to zero in the linearized model and this is exactly what the pseudo-inverse $J^{\dagger}$ does. The Gauss-Newton iteration (eq. 97) in an over-parameterized setting can thus be interpreted as a minimum-norm update, which is a fundamentally different interpretation compared to the under-parameterized setting. Gauss-Newton optimization is usually not used in this over-parameterized setting, however, recent research has started with investigating the theoretical properties of GN for over-parameterized networks (Cai et al., 2019; Zhang et al., 2019). The minimum-norm interpretation of GN suggests that the GNT method on linear networks is closely related to GN optimization for the network parameters. Corollary S6.2 makes this relationship explicit.

**Corollary S6.2.** *Consider a contracting linear multilayer network trained by GN targets according to Condition S2. For a mini-batch size of 1 and an L2 loss function, the resulting parameter updates $\Delta W_i^{GNT}$ are a positive scalar multiple of the parameter updates $\Delta W_i^{GN}$ resulting from the GN optimization method with a block-diagonal approximation of the Gauss-Newton curvature matrix, with block sizes equal to the layer weights sizes.*

*Proof.* In order to avoid tensors in the proof, we use a vectorized form of $W_i$, $\vec{W}_i$, which is a vector containing the concatenated rows of $W_i$. Furthermore, let us define $H_i^T$ as:

$$H_i^T \triangleq \begin{bmatrix} \boldsymbol{h}_i^T & \boldsymbol{0}^T & \ldots & \boldsymbol{0}^T \\ \boldsymbol{0}^T & \ddots & & \vdots \\ \vdots & & \ddots & \boldsymbol{0}^T \\ \boldsymbol{0}^T & \ldots & \boldsymbol{0}^T & \boldsymbol{h}_i^T \end{bmatrix} \tag{98}$$

Then it holds that $W_i \boldsymbol{h}_{i-1} = H_{i-1}^T \vec{W}_i$. Finally, let us define $\bar{f}_{i,L}(\boldsymbol{h}_i)$ as the forward mapping from $\boldsymbol{h}_i$ to $\boldsymbol{h}_L$. Analogue to Lemma S2, the Gauss-Newton steps for $\vec{W}_i$, with a block-diagonal approximation of the GN curvature matrix and a mini-batch size of 1 are given by:

$$\Delta \vec{W}_i^{GN} = -J_{\boldsymbol{h}_L, \vec{W}_i}^{\dagger} \boldsymbol{e}_L, \quad i = 1, ..., L-1 \tag{99}$$

with $J_{\boldsymbol{h}_L, \vec{W}_i} \triangleq \partial \boldsymbol{h}_L / \partial \vec{W}_i$. For linear networks, $J_{\boldsymbol{h}_L, \vec{W}_i}$ is equal to

$$J_{\boldsymbol{h}_L, \vec{W}_i} = J_{\boldsymbol{h}_L, \boldsymbol{h}_i} H_{i-1}^T \tag{100}$$

with $J_{\boldsymbol{h}_L, \boldsymbol{h}_i} \triangleq \partial \boldsymbol{h}_L / \partial \boldsymbol{h}_i$. As $\frac{1}{\|\boldsymbol{h}_{i-1}\|_2} H_{i-1}^T$ has orthonormal rows, $J_{\boldsymbol{h}_L, \vec{W}_i}^{\dagger}$ can be factorized as follows (Campbell and Meyer, 2009):

$$J_{\boldsymbol{h}_L, \vec{W}_i}^{\dagger} = \left( H_{i-1}^T \right)^{\dagger} J_{\boldsymbol{h}_L, \boldsymbol{h}_i}^{\dagger} \tag{101}$$

In order to investigate $\left( H_{i-1}^T \right)^{\dagger}$, we take a look at the singular value decomposition (SVD) of $H_{i-1}^T$.

$$H_{i-1}^T = U \Sigma V^T \tag{102}$$

$U$ and $V$ are both square orthogonal matrices and $\Sigma$ contains the singular values. As the SVD is unique and $H_{i-1}^T$ has orthogonal rows, we can find the singular value decomposition intuitively by normalizing the rows of $H_{i-1}^T$.

$$U = I \tag{103}$$

$$V = \left[ \frac{1}{\|\boldsymbol{h}_{i-1}\|_2} H_{i-1} \quad \vdots \quad \tilde{V} \right] \tag{104}$$

$$\Sigma = \left[ \|\boldsymbol{h}_{i-1}\|_2 I \quad \vdots \quad \mathbf{0} \right] \tag{105}$$

with $\tilde{V}$ an orthonormal basis orthogonal to the column space of $\bar{H}_{i-1}$. Hence, the block-diagonal GN updates for $W_i$ can be written as

$$\Delta \vec{W}_i^{GN} = -\frac{1}{\|\boldsymbol{h}_{i-1}\|_2^2} H_{i-1} J_{\boldsymbol{h}_L, \boldsymbol{h}_i}^\dagger \boldsymbol{e}_L, \quad i = 1, ..., L-1 \tag{106}$$

$$\Delta W_i^{GN} = -\frac{1}{\|\boldsymbol{h}_{i-1}\|_2^2} J_{\boldsymbol{h}_L, \boldsymbol{h}_i}^\dagger \boldsymbol{e}_L \boldsymbol{h}_{i-1}^T, \quad i = 1, ..., L-1 \tag{107}$$

For linear networks, the GNT update for $W_i$ is given by

$$\Delta W_i^{GNT} = -\eta_i \hat{\eta} J_{\boldsymbol{h}_L, \boldsymbol{h}_i}^\dagger \boldsymbol{e}_L \boldsymbol{h}_{i-1}^T, \quad i = 1, ..., L-1 \tag{108}$$

with $\eta_i$ the learning rate of layer $i$ and $\hat{\eta}$ the output step size. We see that $\Delta W_i^{GNT}$ is a positive scalar multiple of $\Delta W_i^{GN}$, thereby concluding the proof. □

**Minimum-norm interpretation of error backpropagation.** Gradient descent (the optimization method behind error backpropagation) has also a minimum-norm interpretation, however, with a different constraint compared to the GNT method, as shown in Proposition S7. Gradient descent uses minimum-norm updates, with as a sole purpose to decrease the loss, whereas GNT uses minimum-norm updates in order to move the output in a specific direction indicated by the output target. In the current implementations of TP and its variants, this output direction is specified as the negative gradient direction in output space, but one could also specify other directions if that would be favourable for the considered application.

**Proposition S7.** *The gradient descent update is the solution to the following minimum-norm optimization problem.*

$$\lim_{c \to 0} \underset{\Delta W_i}{\arg\min} \quad \sum_{i=1}^{L} \|\Delta W_i\|_F^2 \tag{109}$$

$$s.t. \quad L^{(m+1)} = L^{(m)} - c \tag{110}$$

*with $L^{(m)}$ the loss at iteration $m$ and $c$ a positive scalar.*

*Proof.* First, let us define $\vec{W}_i$ as the vector with the concatenated rows of $W_i$ and $\vec{W}$ the concatenated vector of all $\vec{W}_i$. Then, the Lagrangian of this constrained optimization problem can be written as:

$$\mathcal{L} = \left\| \Delta \vec{W} \right\|_2^2 + \lambda (L^{(m+1)} - L^{(m)} + c) \tag{111}$$

A first-order Taylor expansion of $L^{(m+1)}$ around $L^{(m)}$ gives:

$$L^{(m+1)} = L^{(m)} + \frac{\partial L}{\partial \vec{W}} \Delta \vec{W} + \mathcal{O}(\|\Delta \vec{W}\|_2^2) \tag{112}$$

We will assume that $\mathcal{O}(\|\Delta \vec{W}\|_2^2)$ vanishes in the limit of $c \to 0$ relative to the first-order Taylor expansion and $c$, and check this assumption in the end of the proof. Hence, the Lagrangian can be approximated as

$$\mathcal{L} \approx \left\| \Delta \vec{W} \right\|_2^2 + \lambda (\frac{\partial L}{\partial \vec{W}} \Delta \vec{W} + c) \tag{113}$$

As this is a convex optimization problem, the optimal solution can be found by solving the following set of equations.

$$\frac{\partial \mathcal{L}}{\partial \lambda} = \frac{\partial L}{\partial \vec{W}} \Delta \vec{W} + c = 0 \tag{114}$$

$$\frac{\partial \mathcal{L}}{\partial \Delta \vec{W}} = 2 \Delta \vec{W} + \lambda \frac{\partial L}{\partial \vec{W}}^T \tag{115}$$

This set of equations has as solutions

$$\lambda^* = \frac{2c}{\left\| \frac{\partial L}{\partial \vec{W}} \right\|_2^2} \tag{116}$$

$$\Delta \vec{W}^* = -\frac{c}{\left\| \frac{\partial L}{\partial \vec{W}} \right\|_2^2} \frac{\partial L}{\partial \vec{W}}^T \tag{117}$$

We see that the minimum-norm solution for $\Delta \vec{W}$ is the gradient descent update with stepsize $c / \| \frac{\partial L}{\partial \vec{W}} \|_2^2$. Further, $\mathcal{O}(\| \Delta \vec{W} \|_2^2) \sim \mathcal{O}(c^2)$, thus $\mathcal{O}(\| \Delta \vec{W} \|_2^2)$ vanishes in the limit of $c \to 0$ relative to the first-order Taylor expansion and $c$, thereby concluding the proof. □

**GN optimization for minibatches bigger than 1.** In Corollary S6.2 we showed that the GNT method is equal to a block-diagonal approximation of GN for a minibatch size equal to 1. We emphasize, however, that this relation only holds for minibatch sizes equal to 1 and does not generalize to larger minibatches. The Gauss-Newton iteration with a minibatch size of B and a block-diagonal approximation is given by

$$\Delta \vec{W}_i^{GN} = \lim_{\lambda \to 0} \left[ \sum_{b=1}^{B} G^{(b)} + \lambda I \right]^{-1} \left[ \sum_{b=1}^{B} J^{(b)T} e_L^{(b)} \right] \tag{118}$$

$$G^{(b)} \triangleq J^{(b)T} J^{(b)} \tag{119}$$

with $G$ the Gauss-Newton curvature matrix, $J^{(b)} \triangleq \partial \boldsymbol{h}_L / \partial \vec{W}_i$ evaluated on mini-batch sample $b$ and $\lambda$ a damping parameter. For linear networks, the GNT parameter updates with a minibatch size of B can be written as:

$$\Delta \vec{W}_i^{GNT} = \lim_{\lambda \to 0} \sum_{b=1}^{B} \left[ (G^{(b)} + \lambda I)^{-1} J^{(b)T} e_L^{(b)} \right] \tag{120}$$

$$= \sum_{b=1}^{B} \left[ J^{(b)\dagger} e_L^{(b)} \right] \tag{121}$$

We see that for $B = 1$, these expressions overlap, but for $B > 1$ they are not equal anymore, because the order of the sum and inverse operation is switched. GNT can thus be best interpreted in the framework of minimum-norm parameter updates, even if bigger batch-sizes are used, whereas GN optimization has a different interpretation for bigger minibatch sizes.

**Linear networks trained with GNT converge to the global minimum.** Even though the GNT method does not correspond with the GN method for bigger mini-batch sizes, we can still show that GNT on linear networks converges to the global minimum. Theorem S8 proves this for arbitrary batch sizes and an infinitesimally small learning rate. Note that a batch setting instead of a minibatch setting is used (i.e. one minibatch that is equal to the total training batch).

**Theorem S8** (Theorem 6 in main manuscript). *Consider a linear multilayer network of arbitrary architecture, trained with GNT according to Condition S2 and with arbitrary batch size. The resulting parameter updates lie within 90 degrees of the corresponding loss gradients. Moreover, for an infinitesimal small learning rate, the network converges to the global minimum.*

*Proof.* Consider a batch of size $B$ with inputs $\{\boldsymbol{h}_0^{(b)}\}_{1 \leq b \leq B}$ and errors $\{\boldsymbol{e}_L^{(b)}\}_{1 \leq b \leq B}$, and let $A_i \triangleq \sum_{b=1}^{B}[\boldsymbol{e}_L^{(b)}\boldsymbol{h}_i^{(b)T}]$. Given the linear setting, $J_i$ is independent of the presented input for each batch, so

$$\Delta W_i^{GNT} = \sum_{b=1}^{B}[(J_{L-1}\ldots J_{i+1})^{\dagger}\boldsymbol{e}_L^{(b)}\boldsymbol{h}_i^{(b)T}] = (J_{L-1}\ldots J_{i+1})^{\dagger}A_i \tag{122}$$

$$\Delta W_i^{BP} = \sum_{b=1}^{B}[(J_{L-1}\ldots J_{i+1})^{T}\boldsymbol{e}_L^{(b)}\boldsymbol{h}_i^{(b)T}] = (J_{L-1}\ldots J_{i+1})^{T}A_i \tag{123}$$

To compute the angle between the updates we take the Frobenius inner product

$$\langle\Delta W_i^{BP}, \Delta W_i^{GNT}\rangle_F = Tr(A_i^T(J_{L-1}\ldots J_{i+1})(J_{L-1}\ldots J_{i+1})^{\dagger}A_i) \tag{124}$$

$$= Tr(A_iA_i^T(J_{L-1}\ldots J_{i+1})(J_{L-1}\ldots J_{i+1})^{\dagger}) = Tr(B\Sigma) \tag{125}$$

In the last equality a change of basis has been performed with the SVD basis of $(J_{L-1}\ldots J_{i+1})(J_{L-1}\ldots J_{i+1})^{\dagger}$ (which is PSD). $B$ is $A_iA_i^T$ under the new basis while $\Sigma$ is a diagonal matrix with non-negative entries. $A_iA_i^T$ is positive semi-definite and remains so even after a change of basis, so it always has non-negative diagonal entries. Clearly then $Tr(B\Sigma) > 0$ and therefore

$$\langle\Delta W_i^{BP}, \Delta W_i^{GNT}\rangle_F > 0 \tag{126}$$

The angle between the updates hence remains within 90 degrees.

To show that the procedures converge to the same critical point, we prove that

$$\Delta W_i^{BP} = 0 \iff \Delta W_i^{GNT} = 0 \tag{127}$$

This follows from the fact that $ker(M^T) = ker(M^{\dagger})$ for any matrix $M$, since

$$\Delta W_i^{BP} = 0 \iff im(A_i) \subset ker((J_{L-1}\ldots J_{i+1})^T) \tag{128}$$

$$\iff im(A_i) \subset ker((J_{L-1}\ldots J_{i+1})^{\dagger}) \iff \Delta W_i^{GNT} = 0 \tag{129}$$

Finally, for small enough step size we can compare the gradient flow of both GNT and backpropagation, in particular

$$\tau^{BP}\frac{\partial W_i}{\partial t} = -(J_{L-1}\ldots J_{i+1})^T \sum_{b=1}^{B}[\boldsymbol{e}_L^{(b)}\boldsymbol{h}_i^{(b)}] \tag{130}$$

$$\tau^{GNT}\frac{\partial W_i}{\partial t} = -(J_{L-1}\ldots J_{i+1})^{\dagger} \sum_{b=1}^{B}[\boldsymbol{e}_L^{(b)}\boldsymbol{h}_i^{(b)}] \tag{131}$$

Given that backpropagation is guaranteed to always descend the gradient assuming an infinitesimal stepsize, GNT will always reduce the loss under the same assumption, as the directions are always within 90 degrees. As the equilibrium is unique for backpropagation (convex objective) and it is the same as GNT, the two procedures will both converge to the global minimum. $\quad\square$

**Minimum-norm target update interpretation for nonlinear networks.** The direct connection between GNT and GN optimization on the parameters for a minibatch size of 1 does not hold exactly for nonlinear networks. This is due to the nonlinear dependence of $\boldsymbol{h}_i$ on $W_i$ in the local layer loss $\mathcal{L}_i = \|\boldsymbol{h}_i - \hat{\boldsymbol{h}}_i\|_2^2$. For nonlinear networks, the GNT parameter update for a minibatch size of 1 results in

$$\Delta\vec{W}_i^{GNT} = -\eta H_{i-1}D_iJ_i^{\dagger}\boldsymbol{e}_L \tag{132}$$

with $D_i = \partial\boldsymbol{h}_i/\partial\boldsymbol{a}_i$ evaluated at $\boldsymbol{a}_i = W_i\boldsymbol{h}_{i-1}$, $J_i = \partial\boldsymbol{h}_L/\partial\boldsymbol{h}_i$, and $\vec{W}_i$ and $H_{i-1}$ as defined in Corollary S6.2. With the same reasoning as in Corollary S6.2, the GN parameter update for a minibatch size of 1 is given by

$$\Delta\vec{W}_i^{GN} = -\frac{1}{\|\boldsymbol{h}_{i-1}\|_2^2}H_{i-1}(J_iD_i)^{\dagger}\boldsymbol{e}_L \tag{133}$$

Figure S3: The direction of the change in output activation, resulting from GNT update (blue) and BP update (red), plotted together with the loss contours. A nonlinear 2-hidden-layer network with two output neurons ($y_1$ and $y_2$) was used on a synthetic regression dataset.

We see that $\vec{W}_i^{GNT}$ is not equal anymore to $\vec{W}_i^{GN}$. Hence, for an exact interpretation of GNT on nonlinear networks, we should return to the hybrid view. First, the GNT method uses Gauss-Newton optimization to compute its hidden layer targets, after which it does gradient descent on the local loss $\mathcal{L}_i = \|\boldsymbol{h}_i - \hat{\boldsymbol{h}}_i\|_2^2$ to update the network parameters. As the Gauss-Newton optimization for the targets operates in an over-parameterized regime in contracting networks, the GNT target updates should be interpreted as minimum-norm target updates, with as constraint to push the output activation towards its target, similar to Theorem S6. The network parameters are then updated by gradient descent on $\mathcal{L}_i$ to push the layer activation closer to its minimum-norm target. The parameter updates can thus not exactly be interpreted in a minimum-norm sense, but intuitively, the same effect is pursued through the minimum-norm targets. Furthermore, Fig. S3 indicates that for nonlinear networks, the GNT parameter updates push the output activation approximately along the negative gradient direction, similar to Theorem S6.

**Minimum-norm target updates under various norms.** We established an interpretation of GNT, connecting it to finding minimum-norm target updates that push the output activation towards its target. Besides the possibility to vary the direction specified by the output target, one can also use different norms for defining the 'minimum-norm' targets. For the difference reconstruction loss, we used the 2-norm, which gives rise to the link with Moore-Penrose pseudo-inverses and Gauss-Newton optimization. However, the difference reconstruction loss is not restricted to this 2-norm and when using other norms, this will give rise to minimum-norm target updates under the corresponding norm. Ororbia and Mali (2019) indicated that other distance measures might be beneficial for defining local loss functions $\mathcal{L}_i(\hat{\boldsymbol{h}}_i, \boldsymbol{h}_i)$ in the DTP framework. Using our framework, future research can now investigate whether using other norms for the difference reconstruction loss results in better performance.

**Does there exist an energy function for GNT?** In the previous paragraphs, we showed that GNT is closely related to Gauss-Newton optimization and that it converges for linear networks. For non-linear networks, we indicated that GNT can best be interpreted as a hybrid method between GN optimization for finding the targets and gradient descent for updating the parameters. However, it remains an open question whether general convergence can be proved for the nonlinear case, and if yes, to which minimum the GNT method converges. The convergence proofs for error-backpropagation all use the fact that error-backpropagation is a form of gradient descent, and it thus follows the gradient of a loss function (White, 1989; Tesauro et al., 1989). For being able to follow a similar approach for a convergence proof, the GNT method should as well follow the gradient of some *energy function*. However, in Proposition S9 we show that GNT does not follow the gradient of any function, for which we followed a similar approach to Lillicrap et al. (2016). The proposition applies for both linear and non-linear networks. Another path forward towards a general convergence proof for GNT would be to find an energy function for which the GNT updates are *steepest descent updates* under a certain norm (Boyd et al., 2004). The Gauss-Newton parameter updates can be interpreted as steepest descent updates under the norm induced by $G$, the Gauss-Newton curvature matrix. Similarly, one could hope that there exist an energy function and a certain norm for which the GNT updates represent a steepest descent direction. We hypothesize that such energy function and norm do not exist for batch sizes bigger than one, because the order of the summation and inverse operation in equation (120) are reversed compared to equation (118), making it hard to disentangle a norm-inducing matrix and a gradient on the training batch. However, we have not yet proven this rigorously.

**Proposition S9.** *The GNT updates as described in Condition S2 do not produce a conservative dynamical system, hence it does not follow the gradient of any function.*

*Proof.* We will restrict ourselves to the case of one hidden layer, with each layer having only one unit, to avoid cumbersome notations. This choice does not impact the generality of the statement. Let $h_0$ be the input, $h_1 = w_1 h_0$ be the state of the hidden layer, and $h_2 = w_2 h_1$ be the output layer. The error signal is $e_L = h_2 - t$, with $t$ being the target. The updates to $w_2$ and $w_1$ after presentation of one sample are

$$\dot{w}_2 = e_L h_1$$
$$\dot{w}_1 = j^{-1} e_L h_0$$

with $j = \partial h_2 / \partial h_1 = w_2$. If $\dot{w}_1$ and $\dot{w}_1$ follow the gradient of any function, the Hessian of this function should be symmetric:

$$\frac{\partial \dot{w}_2}{\partial w_1} = \frac{\partial \dot{w}_1}{\partial w_2} \tag{134}$$

This condition is also known as *conservative dynamics* in dynamical systems. However

$$\frac{\partial \dot{w}_2}{\partial w_1} = \frac{\partial e_L h_1}{\partial w_1} = \frac{\partial e_L}{\partial w_1} h_1 + e_L \frac{\partial h_1}{\partial w_1} = w_2 h_0 h_1 + e_L h_0 \tag{135}$$

and

$$\frac{\partial \dot{w}_1}{\partial w_2} = \frac{\partial j^{-1} e_L h_0}{\partial w_2} = \left( \frac{\partial j^{-1}}{\partial w_2} e_L + j^{-1} \frac{\partial e_L}{\partial w_2} \right) h_0 = \frac{\partial j^{-1}}{\partial w_2} e_L h_0 + j^{-1} h_1 h_0 \tag{136}$$

$$= -\frac{1}{w_2^2} e_L h_0 + \frac{1}{w_2} h_1 h_0 \tag{137}$$

In general

$$\frac{\partial \dot{w}_2}{\partial w_1} \neq \frac{\partial \dot{w}_1}{\partial w_2} \tag{138}$$

and the dynamical system is non-conservative. ☐

**GNT updates align with gradients in contracting networks.** Despite the lack of a general convergence proof for the GNT method in the nonlinear case, experimental results indicate that our DTP variants (which are an approximation of the GNT method) succeed in decreasing the loss sufficiently, even for nonlinear networks. To help explain this observation, we indicate that the GNT updates approximately align with the gradient direction in high probability, if the network has large hidden layers compared to the output layer, which is the case for many classification and regression problems. We start from the assumption that the properties of $J_{\bar{f}_{i,L}}$ are in general similar to those of a random matrix (Arora et al., 2015). If a zero-mean initialization for the weights is used, we assume that $J_{\bar{f}_{i,L}}$ is also approximately random and zero-mean. For this case, we prove that a scalar multiple of its transpose is a good approximation of its pseudo-inverse. Intuitively, this is easy to understand, as $J^\dagger = J^T (J J^T)^{-1}$ and $J J^T$ is close to a scalar multiple of the identity matrix for zero-mean random matrices in $\mathbb{R}^{m \times n}$ with $n \gg m$:

$$\left[ J J^T \right]_{ii} = \sum_k^n J_{i,k}^2 \approx n \sigma^2 \tag{139}$$

$$\left[ J J^T \right]_{ij} = \sum_k^n J_{i,k} J_{j,k} \approx 0, \quad \forall i \neq j \tag{140}$$

with $J_{i,k}$ the element on the $i$-th row and $k$-th column of $J$, and $\sigma^2$ the variance of the random variables. The GNT update and the gradient (BP) update are given by respectively:

$$\Delta W_i^{GNT} = -\eta \sum_b^B D_i^{(b)} J_i^{(b)\dagger} e_L^{(b)} h_{i-1}^{(b)T} \tag{141}$$

$$\Delta W_i^{BP} = -\eta \sum_b^B D_i^{(b)} J_i^{(b)T} e_L^{(b)} h_{i-1}^{(b)T} \tag{142}$$

with $J_i^{(b)} = \partial \boldsymbol{h}_L / \partial \boldsymbol{h}_i$ evaluated at training sample $b$. We see that if $J_i^{(b)T}$ is approximately equal to $J_i^{(b)\dagger}$, $\Delta W_i^{GNT}$ will align with $\Delta W_i^{BP}$. In Theorem S12, we prove formally that $J_i^{(b)T}$ aligns with $J_i^{(b)\dagger}$ if it is considered as a zero-mean random matrix with $n \gg m$. Before stating the theorem, we need to introduce some auxiliary concepts and lemmas.

**Definition S1.** *Let $A \in \mathbb{R}^{m \times n}$ and $f : \mathbb{R}^{m \times n} \to \mathbb{R}^{n \times m}$. $f(A)$ is an $\epsilon-$approximate Moore-Penrose pseudoinverse (or $\epsilon - $pseudoinverse) of $A$ if it satisfies the following conditions:*

1. $\|Af(A)A - A\|_F^2 \leq \epsilon$

2. $\|f(A)Af(A) - f(A)\|_F^2 \leq \epsilon$

3. $Af(A)$ and $f(A)A$ are Hermitian

*These are an $\epsilon - $approximate version of the Penrose conditions.*

**Lemma S10.** *A classical result from Paul Lévy prescribes that the n-dimensional unit spheres $\mathbb{S}^n$ equipped with their respective uniform probability measures form a normal Lévy family. Therefore given one $\boldsymbol{x} \in \mathbb{S}^n$, for any $\boldsymbol{x}' \in \mathbb{S}^n$ we have*

$$\mathbb{P}(\|\langle \boldsymbol{x}, \boldsymbol{x}' \rangle\| \geq t) \leq C_1 \exp\left(-C_2 n t^2\right) \tag{143}$$

*where $C_1, C_2$ are universal constants.*

*Proof.* For a proof refer to Brazitikos et al. (2014). $\square$

**Lemma S11.** *Given a family $\{\boldsymbol{x}_i\}_{1 \leq i \leq m}$ of $m$ randomly chosen vectors in $\mathbb{S}^n$, then for each $i$*

$$\mathbb{P}(\bigcup_{j \neq i}(\|\langle \boldsymbol{x}_i, \boldsymbol{x}_j \rangle\| \geq t)) \leq C_1 m \exp\left(-C_2 n t^2\right) \tag{144}$$

*Proof.* The proof is a straightforward application of the union bound to Lemma S10. $\square$

**Theorem S12.** *Let $A \in \mathbb{R}^{m \times n}$ be a matrix with $n \gg m$ and with rows sampled uniformly from the unit ball $\mathbb{B}^n$, scaled appropriately. Moreover let $s = \max_i \sqrt{\sum_{j=1}^n a_{ij}^2}$. Then $s^{-1}A^T$ is an $\epsilon - $pseudoinverse of $s^{-1}A$ with high probability.*

*Proof.* Call $B \triangleq s^{-1}A$. $AA^T$ and $A^TA$ are Hermitian, hence the corresponding forms with $B$ are Hermitian as well and they satisfy condition 3 of Definition S1. Consider the row vectors $\{\boldsymbol{x}_i\}_{1 \leq i \leq m}$ of $A$. It is easy to see that

$$(BB^T)_{ij} = s^{-2}\langle \boldsymbol{x}_i, \boldsymbol{x}_j \rangle \tag{145}$$

The row vectors of $B$ are by construction contained in the unit sphere, so it is possible to apply Lemma S11. With probability greater than $1 - C_1 m \exp\left(-C_2 n t^2\right)$

$$\left\|(BB^TB - B)_{ij}\right\|^2 = \left\|\sum_{l=1}^m s^{-3}\langle \boldsymbol{x}_i, \boldsymbol{x}_l \rangle a_{lj} - s^{-1}a_{ij}\right\|^2 \tag{146}$$

$$\leq \left(s^{-3}\left\|\sum_{\substack{l=1 \\ l \neq i}}^m \langle \boldsymbol{x}_i, \boldsymbol{x}_l \rangle a_{lj}\right\| + \left\|(s^{-3}\langle \boldsymbol{x}_i, \boldsymbol{x}_i \rangle - s^{-1})a_{ij}\right\|\right)^2 \tag{147}$$

$$\overset{\text{Cauchy-Schwarz}}{\leq} \left(\frac{t\sqrt{m-1}}{s^3}\sqrt{\sum_{\substack{l=1 \\ l \neq i}}^m a_{lj}^2} + \left\|(s^{-3}\langle \boldsymbol{x}_i, \boldsymbol{x}_i \rangle - s^{-1})a_{ij}\right\|\right)^2 \tag{148}$$

For ease of notation let $c_i = s^{-3}\langle \boldsymbol{x}_i, \boldsymbol{x}_i\rangle - s^{-1}$. Now

$$\left\|BB^T B - B\right\|_F^2 < \sum_{j=1}^{n}\sum_{i=1}^{m}\left(\frac{t\sqrt{m-1}}{s^3}\sqrt{\sum_{\substack{l=1\\l\neq i}}^{m}a_{lj}^2} + \|c_i a_{ij}\|\right)^2 \tag{149}$$

$$= \sum_{j=1}^{n}\sum_{i=1}^{m}\left(\frac{t^2(m-1)}{s^6}\sum_{\substack{l=1\\l\neq i}}^{m}a_{lj}^2 + 2\|c_i a_{ij}\|\frac{t\sqrt{m-1}}{s^3}\sqrt{\sum_{\substack{l=1\\l\neq i}}^{m}a_{lj}^2} + (c_i a_{ij})^2\right)^2 \tag{150}$$

$$= \underbrace{t^2(m-1)s^{-6}\sum_{i=1}^{m}\sum_{j=1}^{n}\sum_{\substack{l=1\\l\neq i}}^{m}a_{lj}^2}_{\text{Part 1}} + \underbrace{2s^{-3}t\sqrt{m-1}\sum_{j=1}^{n}\sum_{i=1}^{m}\|c_i a_{ij}\|\sqrt{\sum_{\substack{l=1\\l\neq i}}^{m}a_{lj}^2}}_{\text{Part 2}} \tag{151}$$

$$+ \underbrace{\sum_{j=1}^{n}\sum_{i=1}^{m}(c_i a_{ij})^2}_{\text{Part 3}} \tag{152}$$

We can easily bound both Part 1 and Part 3 using the definition of $s$, in particular

$$\text{Part 1} = t^2(m-1)s^{-6}\sum_{\substack{l=1\\l\neq i}}^{m}\sum_{i=1}^{m}\sum_{j=1}^{n}a_{lj}^2 \leq t^2(m-1)s^{-6}\sum_{\substack{l=1\\l\neq i}}^{m}\sum_{i=1}^{m}s^2 = t^2 m(m-1)^2 s^{-4} \tag{153}$$

$$\text{Part 3} = \sum_{i=1}^{m}c_i^2\sum_{j=1}^{n}a_{ij}^2 \leq s^2\sum_{i=1}^{m}c_i^2 \tag{154}$$

We will bound Part 2 using a rougher estimate, in particular (we do not consider the constants here)

$$\text{Part 2} \propto \sum_{j=1}^{n}\sum_{i=1}^{m}\|c_i a_{ij}\|\sqrt{\sum_{\substack{l=1\\l\neq i}}^{m}a_{lj}^2} \leq \left(\sum_{i=1}^{m}\|c_i\|\right)\sum_{j=1}^{n}\sum_{i=1}^{m}\|a_{ij}\|\sqrt{\sum_{\substack{l=1\\l\neq i}}^{m}a_{lj}^2} \tag{155}$$

$$\leq \left(\sum_{i=1}^{m}\|c_i\|\right)\sum_{j=1}^{n}\sum_{i=1}^{m}\sqrt{\sum_{l=1}^{m}a_{lj}^2}\sqrt{\sum_{\substack{l=1\\l\neq i}}^{m}a_{lj}^2} \leq \left(\sum_{i=1}^{m}\|c_i\|\right)\sum_{j=1}^{n}\sum_{i=1}^{m}\sqrt{\sum_{l=1}^{m}a_{lj}^2}\sqrt{\sum_{l=1}^{m}a_{lj}^2} \tag{156}$$

$$= \left(\sum_{i=1}^{m}\|c_i\|\right)\sum_{j=1}^{n}\sum_{i=1}^{m}\sum_{l=1}^{m}a_{lj}^2 \leq m^2 s^2\sum_{i=1}^{m}\|c_i\| \tag{157}$$

Now

$$\text{Part 2} + \text{Part 3} < 2s^{-1}tm^2\sqrt{m-1}\sum_{i=1}^{m}\|c_i\| + s^2\sum_{i=1}^{m}c_i^2 < \epsilon/2 \tag{158}$$

for

$$t < \frac{\epsilon - 2s^2\sum_{i=1}^{m}c_i^2}{4s^{-1}m^2\sqrt{m-1}\sum_{i=1}^{m}\|c_i\|} \triangleq K_1 \tag{159}$$

Here we assume necessarily that $\epsilon > 2s^2\sum_{i=1}^{m}c_i^2$. This assumption relies on $c_i$, a matrix dependent measure of the discrepancy in the magnitude of row vectors. It is equal to zero when each row vector is normalised. We also want Part 1 $< \epsilon/2$, so we need

$$t < \frac{\sqrt{\epsilon}}{m\sqrt{m-1}s^2} \triangleq K_2 \tag{160}$$

Let $t < \min\{K_1, K_2\}$, putting everything together

$$\left\| BB^T B - B \right\|_F^2 < \frac{\epsilon}{2} + \frac{\epsilon}{2} = \epsilon \tag{161}$$

Note that the bound $K_1$ is infinite when $c_i$, the row magnitude discrepancy, is zero for all $i$. This was expected, as Part 2 and Part 3 vanish for $c_i = 0$, and Part 1 is bounded by $K_2$. $\left\| B^T BB^T - B^T \right\|_F^2$ can be bounded analogously, since the Frobenius norm is invariant to transposition.

We have that with probability greater than $1 - C_1 m \exp\left(-C_2 n \min\left\{K_1^2, K_2^2\right\}\right)$

1. $\left\| BB^T B - B \right\|_F^2 < \epsilon$

2. $\left\| B^T BB^T - B^T \right\|_F^2 < \epsilon$

Finally for $n = \Omega(\max\left\{\left(\frac{1}{K_1}\right)^2, \left(\frac{1}{K_2}\right)^2\right\} \log(C_1 m/\delta))$ $B^T$ satisfies all the conditions of Definition S1 with probability greater than $1 - \delta$, so it is an $\epsilon -$ pseudoinverse of $B$. $\square$

If we restrict ourselves to matrices whose row vectors have constant magnitude, the proof becomes much easier, as Part 2 and Part 3 vanish. This assumption is reasonable when working in very high dimensional spaces, however, we wanted to provide a point of view that is as general as possible. Further, we tailored the proof towards a uniform distribution of the rows in a scaled version of $\mathbb{B}^n$, such that Lemma S11 can be applied directly. However, the weight matrices in neural networks are often initialized with a uniform random distribution for each element separately, which results in a uniform distribution of the rows in a scaled hyper-cube, instead of the unit hyper-ball. We envision that a variation of Theorem S12 can be made for this case, as the hyper-cube is contained in the unit ball after multiplying the matrix $A$ with $1/s$ and the angles between the randomly drawn rows will have also have a distribution concentrated around 90 degrees (on which Lemma S10 is based).

**Conclusion theoretical results.** Our theoretical analysis of TP shows that TP for invertible networks can best be interpreted as a hybrid method between GN and gradient descent. For non-invertible networks, we showed that DTP leads to inefficient parameter updates, which can be resolved by using our new difference reconstruction loss that restores the hybrid method between GN and gradient descent. We highlighted that the GN-part of the hybrid method implicitly operates in an over-parameterized setting, which leads to minimum-norm parameter and target updates and cannot be compared to GN in an under-parameterized setting. For linear networks, we proved that GNT (the idealized version of DTP combined with DRL) is an approximation of GN optimization with a minibatch size of 1 and that the method converges to the global minimum. For nonlinear networks, we established the minimum-norm target update interpretation and showed that for strongly contracting networks, the GNT parameter update aligns in high probability with the loss gradients.

## B    Further details on experimental results

In this section, we provide further details on the experimental results.

### B.1    Details on the used methods

**Using softmax and cross-entropy loss in the TP framework.** For all our experiments, we used a softmax output layer combined with the cross-entropy loss. In order to incorporate this in the TP framework, it is best to combine the softmax and cross-entropy loss into one output loss:

$$\mathcal{L}^{\text{combined}} = -\sum_{b=1}^{B} \boldsymbol{l}^{(b)T} \log\left(\text{softmax}(\boldsymbol{h}_L^{(b)})\right) \tag{162}$$

with $\boldsymbol{l}^{(b)}$ the one-hot vector representing the class label of training sample $b$ and $\log$ the element-wise logarithmic function. Then the network has a linear output layer $\boldsymbol{h}_L$ and the output target is computed

as

$$\hat{\boldsymbol{h}}_L^{(b)} = \boldsymbol{h}_L^{(b)} - \hat{\eta}\frac{\partial \mathcal{L}^{\text{combined}}}{\partial \boldsymbol{h}_L^{(b)}} \tag{163}$$

The feedback weights are then trained with the difference reconstruction loss, assuming a linear output layer. For this combined loss, the minimum-norm target update interpretation from section A.4 holds, as this interpretation is independent from the used loss function. The connection with GN optimization however is based on an $L_2$ output loss, thus does not hold anymore entirely for $\mathcal{L}^{\text{combined}}$. The Gauss-Newton optimization method can be generalized towards other loss functions by incorporating the *Generalized Gauss-Newton* framework (Schraudolph, 2002; Martens, 2016). In appendix D we show theoretically how the target propagation framework can be adapted to incorporate this generalized Gauss-Newton framework in the computation of its hidden layer targets.

**Algorithms of the methods.** Algorithms 1, 2, 3, 4 and 5 provide the training details for DTP, DTPDRL, DDTP, DDTP-rec and DDTP-control respectively. The DDTP-rec method is explained in section A.3. For all methods, $f_i$ is parameterized as $f_i(\boldsymbol{h}_{i-1}) = \tanh(W_i\boldsymbol{h}_{i-1} + \boldsymbol{b}_i)$. We chose the $\tanh$ nonlinearity, as this gave the best experimental results for the target propagation methods, which was also observed in Lee et al. (2015) and Bartunov et al. (2018). Depending on the method, different parameterizations for $g_i$ are used.

- DTP: $g_i(\boldsymbol{h}_{i+1}) = \tanh(Q_i\boldsymbol{h}_{i+1} + \boldsymbol{c}_i)$
- DTPDRL: $g_i(\boldsymbol{h}_{i+1}) = \tanh(Q_i\boldsymbol{h}_{i+1} + \boldsymbol{c}_i)$
- DDTP-linear: $g_i(\boldsymbol{h}_L) = Q_i\boldsymbol{h}_L + \boldsymbol{c}_i$
- DDTP-control: $g_i(\boldsymbol{h}_L) = Q_i\boldsymbol{h}_L + \boldsymbol{c}_i$
- DDTP-RHL: $g_i(\boldsymbol{h}_L) = \tanh\big(Q_i\tanh(R\boldsymbol{h}_L + \boldsymbol{d}) + \boldsymbol{c}_i\big)$
- DDTP-RHL (rec): $g_i(\boldsymbol{h}_L, \boldsymbol{h}_i) = \tanh\big(Q_i\tanh(R\boldsymbol{h}_L + \boldsymbol{d}) + S_i\boldsymbol{h}_i + \boldsymbol{c}_i\big)$

In the DDTP-RHL variants, $R$ and $\boldsymbol{d}$ are a fixed random matrix and vector, which are shared for all $g_i$.

**Algorithm 1:** DTP training iteration on one minibatch

---

Propagate minibatch activities forwards:
**for** *i in range(1,L)* **do**
   **for** *b in range(1,B)* **do**
      $\boldsymbol{h}_i^{(b)} = f_i(\boldsymbol{h}_{i-1}^{(b)})$

Compute output loss $\mathcal{L}$ on the current minibatch
Compute output targets:
**for** *b in range(1,B)* **do**
   $\hat{\boldsymbol{h}}_L^{(b)} = \boldsymbol{h}_L^{(b)} - \hat{\eta}\frac{\partial \mathcal{L}}{\boldsymbol{h}_L^{(b)}}$

Propagate target backwards:
**for** *i in range(L-1,1)* **do**
   **for** *b in range(1,B)* **do**
      $\hat{\boldsymbol{h}}_i^{(b)} = g_i(\hat{\boldsymbol{h}}_{i+1}^{(b)}) + \boldsymbol{h}_i^{(b)} - g_i(\boldsymbol{h}_{i+1}^{(b)})$

Train the feedback parameters:
**for** *i in range(1, L-1)* **do**
   **for** *b in range(1,B)* **do**
      Generate corrupted activity:
      $\tilde{\boldsymbol{h}}_i^{(b)} = \boldsymbol{h}_i^{(b)} + \sigma\boldsymbol{\epsilon}, \quad \boldsymbol{\epsilon} \sim \mathcal{N}(0,1)$
   Compute the local reconstruction loss:
   $\mathcal{L}_i^{\text{rec}} = \frac{1}{B}\sum_b \|g_i\big(f_{i+1}(\tilde{\boldsymbol{h}}_i^{(b)})\big) - \tilde{\boldsymbol{h}}_i^{(b)}\|_2^2$
   Update the parameters of $g_i$ with a gradient descent step on $\mathcal{L}_i^{\text{rec}}$

Train the forward parameters:
**for** *i in range(1, L)* **do**
   Compute the local loss:
   $\mathcal{L}_i = \frac{1}{B}\sum_b \|\hat{\boldsymbol{h}}_i^{(b)} - \boldsymbol{h}_i^{(b)}\|_2^2$
   Update the parameters of $f_i$ with a gradient descent step on $\mathcal{L}_i$

---

---
**Algorithm 2:** DTPDRL training iteration on one minibatch
---

Propagate minibatch activities forwards:
**for** *i in range(1,L)* **do**
    **for** *b in range(1,B)* **do**
        $\boldsymbol{h}_i^{(b)} = f_i(\boldsymbol{h}_{i-1}^{(b)})$

Compute output loss $\mathcal{L}$ on the current minibatch
Compute output targets:
**for** *b in range(1,B)* **do**
    $\hat{\boldsymbol{h}}_L^{(b)} = \boldsymbol{h}_L^{(b)} - \hat{\eta}\frac{\partial \mathcal{L}}{\boldsymbol{h}_L^{(b)}}$

Propagate target backwards:
**for** *i in range(L-1,1)* **do**
    **for** *b in range(1,B)* **do**
        $\hat{\boldsymbol{h}}_i^{(b)} = g_i(\hat{\boldsymbol{h}}_{i+1}^{(b)}) + \boldsymbol{h}_i^{(b)} - g_i(\boldsymbol{h}_{i+1}^{(b)})$

Train the feedback parameters:
**for** *i in range(1, L-1)* **do**
    **for** *b in range(1,B)* **do**
        Generate corrupted activity:
        $\tilde{\boldsymbol{h}}_i^{(b)} = \boldsymbol{h}_i^{(b)} + \sigma\boldsymbol{\epsilon}, \quad \boldsymbol{\epsilon} \sim \mathcal{N}(0,1)$
        Propagate the corrupted activity to the output layer:
        **for** *k in range(i+1, L)* **do**
            $\tilde{\boldsymbol{h}}_k^{(b)} = f_i(\tilde{\boldsymbol{h}}_{k-1}^{(b)})$

        Propagate the corrupted output activity backwards to reconstruct $\tilde{\boldsymbol{h}}_i$:
        $\boldsymbol{h}_L^{\mathrm{rec}(b)} = \tilde{\boldsymbol{h}}_L^{(b)}$
        **for** *k in range(L-1,i)* **do**
            $\boldsymbol{h}_k^{\mathrm{rec}(b)} = g_k(\boldsymbol{h}_{k+1}^{\mathrm{rec}(b)}) + \boldsymbol{h}_k^{(b)} - g_k(\boldsymbol{h}_{k+1}^{(b)})$

    Compute the difference reconstruction loss:
    $\mathcal{L}_i^{\mathrm{diff,rec}} = \frac{1}{B}\sum_b \|\boldsymbol{h}_i^{\mathrm{rec}(b)} - \tilde{\boldsymbol{h}}_i^{(b)}\|_2^2$
    Update the parameters of $g_i$ with a gradient descent step on $\mathcal{L}_i^{\mathrm{diff,rec}}$ + weight decay

Train the forward parameters:
**for** *i in range(1, L)* **do**
    Compute the local loss:
    $\mathcal{L}_i = \frac{1}{B}\sum_b \|\hat{\boldsymbol{h}}_i^{(b)} - \boldsymbol{h}_i^{(b)}\|_2^2$
    Update the parameters of $f_i$ with a gradient descent step on $\mathcal{L}_i$

---

---
**Algorithm 3:** DDTP training iteration on one minibatch
---

Propagate minibatch activities forwards:
**for** *i in range(1,L)* **do**

    **for** *b in range(1,B)* **do**

        $\boldsymbol{h}_i^{(b)} = f_i(\boldsymbol{h}_{i-1}^{(b)})$

Compute output loss $\mathcal{L}$ on the current minibatch
Compute output targets:
**for** *b in range(1,B)* **do**

    $\hat{\boldsymbol{h}}_L^{(b)} = \boldsymbol{h}_L^{(b)} - \hat{\eta}\frac{\partial \mathcal{L}}{\boldsymbol{h}_L^{(b)}}$

Propagate target backwards:
**for** *i in range(L-1,1)* **do**

    **for** *b in range(1,B)* **do**

        $\hat{\boldsymbol{h}}_i^{(b)} = g_i(\hat{\boldsymbol{h}}_L^{(b)}) + \boldsymbol{h}_i^{(b)} - g_i(\boldsymbol{h}_L^{(b)})$

Train the feedback parameters:
**for** *i in range(1, L-1)* **do**

    **for** *b in range(1,B)* **do**

        Generate corrupted activity:

        $\tilde{\boldsymbol{h}}_i^{(b)} = \boldsymbol{h}_i^{(b)} + \sigma\boldsymbol{\epsilon}, \quad \boldsymbol{\epsilon} \sim \mathcal{N}(0,1)$

        Propagate the corrupted activity to the output layer:

        **for** *k in range(i+1, L)* **do**

            $\tilde{\boldsymbol{h}}_k^{(b)} = f_i(\tilde{\boldsymbol{h}}_{k-1}^{(b)})$

        Propagate the corrupted output activity backwards to reconstruct $\tilde{\boldsymbol{h}}_i$:

        $\boldsymbol{h}_i^{\mathrm{rec}(b)} = g_i(\tilde{\boldsymbol{h}}_L^{(b)}) + \boldsymbol{h}_i^{(b)} - g_i(\boldsymbol{h}_L^{(b)})$

    Compute the difference reconstruction loss:

    $\mathcal{L}_i^{\mathrm{diff,rec}} = \frac{1}{B}\sum_b \|\boldsymbol{h}_i^{\mathrm{rec}(b)} - \tilde{\boldsymbol{h}}_i^{(b)}\|_2^2$

    Update the parameters of $g_i$ with a gradient descent step on $\mathcal{L}_i^{\mathrm{diff,rec}}$ + weight decay

Train the forward parameters:
**for** *i in range(1, L)* **do**

    Compute the local loss:

    $\mathcal{L}_i = \frac{1}{B}\sum_b \|\hat{\boldsymbol{h}}_i^{(b)} - \boldsymbol{h}_i^{(b)}\|_2^2$

    Update the parameters of $f_i$ with a gradient descent step on $\mathcal{L}_i$

---
**Algorithm 4:** DDTP (rec) training iteration on one minibatch
---
Propagate minibatch activities forwards:
**for** *i in range(1,L)* **do**
  **for** *b in range(1,B)* **do**
    $h_i^{(b)} = f_i(h_{i-1}^{(b)})$

Compute output loss $\mathcal{L}$ on the current minibatch
Compute output targets:
**for** *b in range(1,B)* **do**
  $\hat{h}_L^{(b)} = h_L^{(b)} - \hat{\eta} \frac{\partial \mathcal{L}}{h_L^{(b)}}$

Propagate target backwards:
**for** *i in range(L-1,1)* **do**
  **for** *b in range(1,B)* **do**
    $\hat{h}_i^{(b)} = g_i(\hat{h}_L^{(b)}, h_i) + h_i^{(b)} - g_i(h_L^{(b)}, h_i)$

Train the feedback parameters:
**for** *i in range(1, L-1)* **do**
  **for** *b in range(1,B)* **do**
    Generate corrupted activity:
    $\tilde{h}_i^{(b)} = h_i^{(b)} + \sigma \epsilon, \quad \epsilon \sim \mathcal{N}(0, 1)$
    Propagate the corrupted activity to the output layer:
    **for** *k in range(i+1, L)* **do**
      $\tilde{h}_k^{(b)} = f_i(\tilde{h}_{k-1}^{(b)})$

    Propagate the corrupted output activity backwards to reconstruct $\tilde{h}_i$:
    $h_i^{\mathrm{rec}(b)} = g_i(\tilde{h}_L^{(b)}, h_i) + h_i^{(b)} - g_i(h_L^{(b)}, h_i)$
  Compute the difference reconstruction loss:
  $\mathcal{L}_i^{\mathrm{diff,rec}} = \frac{1}{B} \sum_b \|h_i^{\mathrm{rec}(b)} - \tilde{h}_i^{(b)}\|_2^2$
  Update the parameters of $g_i$ with a gradient descent step on $\mathcal{L}_i^{\mathrm{diff,rec}}$ + weight decay

Train the forward parameters:
**for** *i in range(1, L)* **do**
  Compute the local loss:
  $\mathcal{L}_i = \frac{1}{B} \sum_b \|\hat{h}_i^{(b)} - h_i^{(b)}\|_2^2$
  Update the parameters of $f_i$ with a gradient descent step on $\mathcal{L}_i$
---

**Algorithm 5:** DDTP-control training iteration on one minibatch

Propagate minibatch activities forwards:
**for** *i in range(1,L)* **do**
    **for** *b in range(1,B)* **do**
        $\boldsymbol{h}_i^{(b)} = f_i(\boldsymbol{h}_{i-1}^{(b)})$

Compute output loss $\mathcal{L}$ on the current minibatch
Compute output targets:
**for** *b in range(1,B)* **do**
    $\hat{\boldsymbol{h}}_L^{(b)} = \boldsymbol{h}_L^{(b)} - \hat{\eta}\frac{\partial \mathcal{L}}{\boldsymbol{h}_L^{(b)}}$

Propagate target backwards:
**for** *i in range(L-1,1)* **do**
    **for** *b in range(1,B)* **do**
        $\hat{\boldsymbol{h}}_i^{(b)} = g_i(\hat{\boldsymbol{h}}_L^{(b)}) + \boldsymbol{h}_i^{(b)} - g_i(\boldsymbol{h}_L^{(b)})$

Train the feedback parameters:
**for** *i in range(1, L-1)* **do**
    **for** *b in range(1,B)* **do**
        Generate corrupted activity:
        $\tilde{\boldsymbol{h}}_i^{(b)} = \boldsymbol{h}_i^{(b)} + \sigma\boldsymbol{\epsilon}, \quad \boldsymbol{\epsilon} \sim \mathcal{N}(0,1)$
        Propagate the corrupted activity to the output layer:
        **for** *k in range(i+1, L)* **do**
            $\tilde{\boldsymbol{h}}_k^{(b)} = f_i(\tilde{\boldsymbol{h}}_{k-1}^{(b)})$

        Propagate the corrupted output activity backwards to reconstruct $\tilde{\boldsymbol{h}}_i$ (without the difference correction):
        $\boldsymbol{h}_i^{\text{rec}(b)} = g_i(\tilde{\boldsymbol{h}}_L^{(b)})$
    Compute the reconstruction loss:
    $\mathcal{L}_i^{\text{rec}} = \frac{1}{B}\sum_b \|\boldsymbol{h}_i^{\text{rec}(b)} - \tilde{\boldsymbol{h}}_i^{(b)}\|_2^2$
    Update the parameters of $g_i$ with a gradient descent step on $\mathcal{L}_i^{\text{rec}}$ + weight decay

Train the forward parameters:
**for** *i in range(1, L)* **do**
    Compute the local loss:
    $\mathcal{L}_i = \frac{1}{B}\sum_b \|\hat{\boldsymbol{h}}_i^{(b)} - \boldsymbol{h}_i^{(b)}\|_2^2$
    Update the parameters of $f_i$ with a gradient descent step on $\mathcal{L}_i$

**CNN implementation.** The convolutional blocks consist of a sequence of a convolutional layer, a $\tanh$ activation function and a maxpool layer. For the TP variants, the feedback pathways propagate a target $\hat{h}_i$ for the activations $h_i$ of the maxpool layer. The filter weights of the convolutional layer are then updated with a gradient step on $\|\hat{h}_i - h_i\|_2^2$, in line with the above algorithms. For DFA, the feedback connections randomly project the output error to an error for the maxpool layer, after which automatic differentiation is used to compute the update for the filter weights.

**Training details.** For all target propagation variants, we trained both the forward and feedback parameters on each minibatch. Note that Lee et al. (2015) alternated between training the feedback weights for one epoch while keeping the forward weights fixed and training the forward weights for one epoch while keeping the feedback weights fixed. The difference between this alternating approach and the normal approach of training all parameters was investigated in Bartunov et al. (2018). As in our methods, the feedback paths should approximate the pseudo inverse of the forward paths (which is continuously changing), we chose the normal approach. Furthermore, to improve the training of the feedback paths, we pre-train the feedback parameters for some epochs before the start of the training and we insert one or more epochs for training only the feedback parameters, between each epoch of training both forward and feedback parameters:

- MNIST: 6 epochs of pretraining and 1 epoch of pure feedback training between each epoch of training both forward and feedback parameters
- Fashion-MNIST: 6 epochs of pretraining and 1 epoch of pure feedback training between each epoch of training both forward and feedback parameters
- MNIST-frozen: 10 epochs of pretraining and 0 epochs of pure feedback training between each epoch of training both forward and feedback parameters
- CIFAR10: 10 epochs of pretraining and 2 epoch of pure feedback training between each epoch of training both forward and feedback parameters
- CIFAR10 with CNNs: 10 epochs of pretraining and 1 epoch of pure feedback training between each epoch of training both forward and feedback parameters

These periods of pure feedback training could be interpreted as a sleeping-phase of the network. For the DDTP-RHL method, we observed that the feedback parameters needed extra time for training, as the feedback weights are of much bigger dimension than the forward weights. Therefore, we investigated the DDTP-RHL methods with extra feedback training (indicated by DDTP-RHL (extra FB) and DDTP-RHL(rec. and extra FB) in the experiments):

- MNIST: 10 epochs of pretraining and 4 epochs of pure feedback training between each epoch of training both forward and feedback parameters
- Fashion-MNIST: 10 epochs of pretraining and 4 epochs of pure feedback training between each epoch of training both forward and feedback parameters
- MNIST-frozen: 10 epochs of pretraining and 4 epochs of pure feedback training between each epoch of training both forward and feedback parameters
- CIFAR10: 10 epochs of pretraining and 4 epochs of pure feedback training between each epoch of training both forward and feedback parameters

We use a separate ADAM optimizer (Kingma and Ba, 2014) for training the forward and feedback parameters respectively.

**Architecture details.** We use fully connected (FC) layers for all datasets and architectures corresponding to Bartunov et al. (2018).

- MNIST: 5 FC hidden layers of 256 neurons + 1 softmax layer of 10 neurons (+ one random hidden feedback layer of 1024 neurons for the DDTP-RHL variants)
- Fashion-MNIST: 5 FC hidden layers of 256 neurons + 1 softmax layer of 10 neurons (+ one random hidden feedback layer of 1024 neurons for the DDTP-RHL variants)
- MNIST-frozen: 5 FC hidden layers of 256 neurons + 1 softmax layer of 10 neurons (+ one random hidden feedback layer of 1024 neurons for the DDTP-RHL variants)

- CIFAR10: 3 FC hidden layers of 1024 neurons + 1 softmax layer of 10 neurons (+ one random hidden feedback layer of 2048 neurons for the DDTP-RHL variants)
- CIFAR10 with CNN:
  - Conv 5x5x32, stride 1 (with $\tanh$ nonlinearity)
  - Maxpool 3x3, stride 2
  - Conv 5x5x64, stride 1 (with $\tanh$ nonlinearity)
  - Maxpool 3x3, stride 2
  - FC hidden layer with 512 neurons (with $\tanh$ nonlinearity)
  - Softmax layer of 10 neurons

**Numerical errors.** We observed that using the standard data type `float32` for the tensors in PyTorch (Paszke et al., 2019) leads to numerical errors when computing the weight updates of the network. The output target is computed as $\hat{\boldsymbol{h}}_L = \boldsymbol{h}_L - \hat{\eta} \boldsymbol{e}_L$. When the output gradient $\boldsymbol{e}_L$ becomes very small during training, `float32` cannot represent the small addition $\hat{\eta} \boldsymbol{e}_L$ to the relatively large output activation $\boldsymbol{h}_L$. Hence, when you compute the target updates $\Delta \boldsymbol{h}_i = \hat{\boldsymbol{h}}_i - \boldsymbol{h}_i$, $\hat{\eta} \boldsymbol{e}_L$ is lost and $\Delta \boldsymbol{h}_i \approx 0$. This problem can be resolved by using `float64`.

**Hyperparameters.** All hyperparameter searches were done based on the best validation error over 100 epochs, with a validation set of 5000 samples, taken from the training set (60000 training samples in MNIST and Fashion-MNIST, 50000 training samples in CIFAR10). We used the Tree of Parzen Estimators algorithm (Bergstra et al., 2011) for performing the hyperparameter search, based on the Hyperopt (Bergstra et al., 2013) and Ray Tune (Liaw et al., 2018) Python packages. Table S1 summarizes which hyperparameters were used for the various methods. The hyperparameter search intervals for the various methods are given in Table S2, S3, S4 and S5. Note that we used different search intervals for $\hat{\eta}$ and $\sigma$ in for our new methods (DDTP and DTPDRL) compared to the original DTP methods. Our theory is based on small values for $\hat{\eta}$ and $\sigma$, whereas the implementations of DTP in the literature used relatively big values for $\hat{\eta}$ and $\sigma$ (Lee et al., 2015; Bartunov et al., 2018). The original DTP method does not use weight decay on the feedback parameters, hence we did not include it for this method. For enabling a fair comparison with our methods, we included the possibility of weight decay on the feedback parameters in DTP-pretrained. The specific hyperparameter configurations for all methods can be found in our code base.[3] In the hyperparameter searches, we treated $\text{lr} \cdot \hat{\eta}$ and $\hat{\eta}$ as hyperparameters instead of lr and $\hat{\eta}$ separately, as the stepsize of the forward parameter updates $\Delta W_i$ is determined by $\text{lr} \cdot \hat{\eta}$. The value of $\hat{\eta}$ then decides the trade-off between lr and $\hat{\eta}$: the smaller $\hat{\eta}$, the better the Taylor approximations in the theory hold and thus the closer the method is related to GN, however, the implementation is more prone to numerical errors for small $\hat{\eta}$. For the CNNs, we have a separate Adam optimizer for each layer to train the forward weights and one shared Adam optimizer for training all feedback weights. We fix all $\beta$ values to $\beta_1 = \beta_{1,\text{fb}} = 0.9$ and $\beta_2 = \beta_{2,\text{fb}} = 0.999$

Table S1: Hyperparameter symbols

| Symbol | Hyperparameter |
|---|---|
| lr | learning rate of the Adam optimizer for the forward parameters |
| $\beta_1$ | $\beta_1$ parameter of the Adam optimizer for the forward parameters |
| $\beta_2$ | $\beta_2$ parameter of the Adam optimizer for the forward parameters |
| $\epsilon$ | $\epsilon$ parameter of the Adam optimizer for the forward parameters |
| $\text{lr}_{\text{fb}}$ | learning rate of the Adam optimizer for the feedback parameters |
| $\beta_{1,\text{fb}}$ | $\beta_1$ parameter of the Adam optimizer for the feedback parameters |
| $\beta_{2,\text{fb}}$ | $\beta_2$ parameter of the Adam optimizer for the feedback parameters |
| $\epsilon_{\text{fb}}$ | $\epsilon$ parameter of the Adam optimizer for the feedback parameters |
| $\hat{\eta}$ | output target stepsize |
| $\sigma$ | standard deviation of noise perturbations in the reconstruction loss |
| $\text{wd}_{\text{fb}}$ | weight decay for the feedback parameters |

Table S2: Hyperparameter search intervals for DTPDRL and the DDTP variants

| Hyperparameter | Search interval |
|---|---|
| $\text{lr} \cdot \hat{\eta}$ | $[10^{-6} : 10^{-4}]$ |
| $\beta_1$ | $\{0.9; 0.99; 0.999\}$ |
| $\beta_2$ | $\{0.9; 0.99; 0.999\}$ |
| $\epsilon$ | $[10^{-8} : 10^{-4}]$ |
| $\text{lr}_{\text{fb}}$ | $[3 \cdot 10^{-5} : 5 \cdot 10^{-3}]$ |
| $\beta_{1,\text{fb}}$ | $\{0.9; 0.99; 0.999\}$ |
| $\beta_{2,\text{fb}}$ | $\{0.9; 0.99; 0.999\}$ |
| $\epsilon_{\text{fb}}$ | $[10^{-8} : 10^{-4}]$ |
| $\hat{\eta}$ | $[10^{-2} : 10^{-1}]$ |
| $\sigma$ | $[10^{-3} : 10^{-1}]$ |
| $\text{wd}_{\text{fb}}$ | $[10^{-7} : 10^{-4}]$ |

Table S3: Hyperparameter search intervals for DTP

| Hyperparameter | Search interval |
|---|---|
| $\text{lr} \cdot \hat{\eta}$ | $[10^{-6} : 10^{-4}]$ |
| $\beta_1$ | $\{0.9; 0.99; 0.999\}$ |
| $\beta_2$ | $\{0.9; 0.99; 0.999\}$ |
| $\epsilon$ | $[10^{-8} : 10^{-4}]$ |
| $\text{lr}_{\text{fb}}$ | $[3 \cdot 10^{-5} : 5 \cdot 10^{-3}]$ |
| $\beta_{1,\text{fb}}$ | $\{0.9; 0.99; 0.999\}$ |
| $\beta_{2,\text{fb}}$ | $\{0.9; 0.99; 0.999\}$ |
| $\epsilon_{\text{fb}}$ | $[10^{-8} : 10^{-4}]$ |
| $\hat{\eta}$ | $[10^{-2} : 3 \cdot 10^{-1}]$ |
| $\sigma$ | $[10^{-3} : 3 \cdot 10^{-1}]$ |
| $\text{wd}_{\text{fb}}$ | $0$ |

Table S4: Hyperparameter search intervals for DTP (pre-trained)

| Hyperparameter | Search interval |
|---|---|
| $\text{lr} \cdot \hat{\eta}$ | $[10^{-6} : 10^{-4}]$ |
| $\beta_1$ | $\{0.9; 0.99; 0.999\}$ |
| $\beta_2$ | $\{0.9; 0.99; 0.999\}$ |
| $\epsilon$ | $[10^{-8} : 10^{-4}]$ |
| $\text{lr}_{\text{fb}}$ | $[3 \cdot 10^{-5} : 5 \cdot 10^{-3}]$ |
| $\beta_{1,\text{fb}}$ | $\{0.9; 0.99; 0.999\}$ |
| $\beta_{2,\text{fb}}$ | $\{0.9; 0.99; 0.999\}$ |
| $\epsilon_{\text{fb}}$ | $[10^{-8} : 10^{-4}]$ |
| $\hat{\eta}$ | $[10^{-2} : 3 \cdot 10^{-1}]$ |
| $\sigma$ | $[10^{-3} : 3 \cdot 10^{-1}]$ |
| $\text{wd}_{\text{fb}}$ | $[10^{-7} : 10^{-4}]$ |

Table S5: Hyperparameter search intervals for BP and DFA

| Hyperparameter | Search interval |
|---|---|
| $\text{lr}$ | $[10^{-5} : 10^{-2}]$ |
| $\beta_1$ | $\{0.9; 0.99; 0.999\}$ |
| $\beta_2$ | $\{0.9; 0.99; 0.999\}$ |
| $\epsilon$ | $[10^{-8} : 10^{-3}]$ |

## B.2 Extended experimental results

Here, we provide extra experimental results and figures, complementing the results of the main manuscript. Table S6 provides extra test performance results and Fig. S4, S5, S6 and S7 give the alignment angles with the loss gradients and GNT updates for all methods and datasets (with hyperparemeters tuned for test performance). Tables S8, S9 and S10 provide the training losses for all experiments.

Table S6: Test errors corresponding to the epoch with the best validation error over a training of 100 epochs. The mean and standard deviation over 10 randomly initialized weight configurations are given. The best test errors (except BP) are displayed in bold.

|  | MNIST | Frozen-MNIST | Fashion-MNIST | CIFAR10 |
|---|---|---|---|---|
| BP | $1.98 \pm 0.14\%$ | $4.39 \pm 0.13\%$ | $10.74 \pm 0.16\%$ | $45.60 \pm 0.50\%$ |
| DDTP-linear | $\mathbf{2.04 \pm 0.08}\%$ | $6.42 \pm 0.17\%$ | $\mathbf{11.11 \pm 0.35}\%$ | $\mathbf{50.36 \pm 0.26}\%$ |
| DDTP-RHL | $2.10 \pm 0.14\%$ | $\mathbf{5.11 \pm 0.19}\%$ | $11.53 \pm 0.31\%$ | $51.94 \pm 0.49\%$ |
| DDTP-RHL (extra FB) | $2.12 \pm 0.13\%$ | $5.34 \pm 0.17\%$ | $11.13 \pm 0.19\%$ | $51.24 \pm 0.44\%$ |
| DDTP-RHL (rec. and extra FB) | $2.21 \pm 0.10\%$ | $5.36 \pm 0.18\%$ | $11.74 \pm 0.34\%$ | $52.31 \pm 0.44\%$ |
| DTPDRL | $2.21 \pm 0.09\%$ | $6.10 \pm 0.17\%$ | $11.22 \pm 0.20\%$ | $50.80 \pm 0.43\%$ |
| DDTP-control | $2.51 \pm 0.08\%$ | $9.70 \pm 0.31\%$ | $11.71 \pm 0.28\%$ | $51.75 \pm 0.43\%$ |
| DTP | $2.39 \pm 0.19\%$ | $10.64 \pm 0.53\%$ | $11.49 \pm 0.23\%$ | $51.74 \pm 0.30\%$ |
| DTP (pre-trained) | $2.26 \pm 0.18\%$ | $9.31 \pm 0.40\%$ | $11.52 \pm 0.31\%$ | $52.20 \pm 0.50\%$ |
| DFA | $2.17 \pm 0.14\%$ | / | $11.26 \pm 0.25\%$ | $51.28 \pm 0.41\%$ |

**Extra feedback weight training.** Although DDTP-RHL provides the best feedback signals in the frozen MNIST task, we observed that it needs extra training iterations for its many feedback parameters to enable decent performance for the more complex tasks. **DDTP-RHL (extra FB)** uses a couple of extra epochs of pure feedback parameter training between each epoch of both forward and feedback parameter training (see Section B.1 for details). On Fashion MNIST, this extra feedback weight training brings the performance of DDTP-RHL (extra FB) on the same level of DDTP-linear. On CIFAR10, DDTP-RHL (extra FB) has a significantly better performance compared to DDTP-RHL, however, it does not perform as good as DDTP-linear. It appears that on CIFAR10, the possibility for better feedback signals does not outweigh the training challenges caused by the increased complexity of the feedback connections in DDTP-RHL. Hence, future research should explore inductive biases and training methods that can improve the feedback parameter training, while harvesting the benefits of the better feedback signals provided by DDTP-RHL.

**Recurrent feedback connections.** In Section A.3 we discussed that adding recurrent connections to the feedback path could improve the alignment of the feedback signals with the ideal GNT updates. To test this hypothesis, we used the DDTP-RHL (rec.) method (see Section A.3 and Algorithm 4) on the various datasets and computed the alignment angles with the loss gradients and damped GNT updates. Similar to DDTP-RHL (extra FB), we used extra training epochs for training the feedback connections, resulting in **DDTP-RHL (rec. and extra FB)**. Fig. S4 shows that the alignment with both the loss gradients and the damped GNT methods indeed improved by adding recurrent feedback connections. However, the performance results in Table S6 and Fig. S5 and S6 indicate that the training challenges originating from the added complexity outweigh the capacity for better feedback signals and result in worse performance compared to DDTP-linear. Similar to DDTP-RHL, future research can explore whether better optimizers and inductive biases can alleviate these training challenges and improve performance.

**Damped Gauss-Newton targets.** In our theory, we focussed on undamped GN targets. For practical reasons of stability, however, it is advised to damp the GN targets, which we did by using weight decay on the feedback parameters. Condition S3 formalizes these damped GN targets.

**Condition S3** (Damped Gauss-Newton Target method). *The network is trained by damped GN targets: each hidden layer target is computed by*

$$\Delta \boldsymbol{h}_i^{(b)} \triangleq \hat{\boldsymbol{h}}_i^{(b)} - \boldsymbol{h}_i^{(b)} = -\hat{\eta} J_{\bar{f}_{i,L}}^{(b)T} \big( J_{\bar{f}_{i,L}}^{(b)} J_{\bar{f}_{i,L}}^{(b)T} + \lambda I \big)^{-1} \boldsymbol{e}_L^{(b)}, \tag{164}$$

*with $\lambda$ a Tikhonov damping constant, after which the network parameters of each layer $i$ are updated by a gradient descent step on its corresponding local mini-batch loss $\mathcal{L}_i = \sum_b \|\Delta \boldsymbol{h}_i^{(b)}\|_2^2$, while considering $\hat{\boldsymbol{h}}_i$ fixed.*

For computing the alignment angles in Fig. S4, S5 and S6, the weight updates $W_i$ should be compared with the damped GN targets of Condition S3. To find the damping constant $\lambda$ that best corresponds to the weight updates $\Delta W_i$, we computed the alignment angles for $\lambda \in \{0, 10^{-5}, 10^{-4}, 10^{-3}, 10^{-2}, 10^{-1}, 1, 10\}$ and selected the damping value that resulted in the best alignment. The alignment angles were averaged over one epoch, after two epochs of training (such that the tanh activation functions are already partially pushed in their nonlinear regime). Table S7 provides the selected $\lambda$ for all methods on Fashion-MNIST. Our code base contains the specific damping values used for the other datasets. Interestingly, the DDTP variants and DTPDRL method align best with GNT updates that are highly damped ($\lambda = 1$ for all layers except the last hidden layer), while the weight decay we used on the feedback parameters is in the order of magnitude of $10^{-4}$ or smaller. Hence, our methods experience *implicit damping*, on top of the explicit damping introduced by the weight decay. We hypothesize that this implicit damping originates from the limited amount of parameters in the feedback paths, that prevent the DRL of being minimized to its absolute minimum, as discussed in Section A.3. Hence, it is not possible for each batch sample $b$ that $J_{\bar{g}_{L,i}}^{(b)} = J_{\bar{f}_{i,L}}^{(b)T} \big( J_{\bar{f}_{i,L}}^{(b)} J_{\bar{f}_{i,L}}^{(b)T} + \lambda I \big)^{-1}$, and the extreme singular values of $J_{\bar{g}_{L,i}}^{(b)}$ will be damped first, as they interfere the most with $J_{\bar{g}_{L,i}}^{(k \neq b)}$ of other batch samples, resulting in a similar behaviour as Tikhonov damping. In line with this hypothesis, DDTP-RHL (rec. and extra FB) experiences less implicit damping ($\lambda = 0.1$), as it has more feedback parameters. We emphasize that the DDTP variants and DTPDRL method did not select the highest damping value $\lambda = 10$, indicating that it aligns better with damped GNT updates instead of loss gradients, which is confirmed by Fig. S4, S5 and S6.

Table S7: Damping values $\lambda$ for the damped GNT updates according to Condition S3, which are used to compute the alignment angles on Fashion-MNIST in Fig. S4b.

|  | Layer 1 | Layer 2 | Layer 3 | Layer 4 | Layer 5 |
|---|---|---|---|---|---|
| DDTP-linear | 1 | 1 | 1 | 1 | 0 |
| DDTP-RHL | 1 | 1 | 1 | 1 | 0.001 |
| DDTP-RHL (extra FB) | 1 | 1 | 1 | 1 | 0.001 |
| DDTP-RHL (rec. and extra FB) | 0.1 | 0.1 | 0.1 | 0.1 | 0.0001 |
| DTPDRL | 1 | 1 | 1 | 1 | 0.01 |
| DDTP-control | 10 | 1 | 1 | 10 | 10 |
| DTP | 10 | 10 | 10 | 10 | 10 |
| DTP (pre-trained) | 1 | 1 | 1 | 1 | 1 |
| DFA | 10 | 1 | 1 | 10 | 1 |

**A need for customized optimizers for TP methods.** We used the Adam optimizer (Kingma and Ba, 2014) for all experiments, because our methods aligned well with the loss gradients, and the Adam optimizer is tuned for gradient descent. However, our theory and experiments showed that our DTP variants can best be compared to damped GNT updates. This raises the question whether other optimizers can be developed, that can make use of the specific characteristics of the TP methods, such as its minimum-norm property. We hypothesize that with a tailored optimizer, that mitigates the training challenges originating from the complex interplay of feedforward and feedback parameter training and that harvests the beneficial minimum-norm characteristics of the TP methods, the current performance gap with BP can be further closed or even exceeded for some applications.

(a) Angles between $\Delta W_i$ and the loss gradients.

(b) Angles between $\Delta W_i$ and damped GNT updates.

Figure S4: Angles between the weight updates $\Delta W_i$ of all hidden layers with (a) the loss gradient directions and (b) the damped GNT weight updates according to Condition S3 on Fashion-MNIST. A window-average of the angles is plotted, together with the window–standard–deviation.

(a) Angles between $\Delta W_i$ and the loss gradients.

(b) Angles between $\Delta W_i$ and damped GNT updates.

Figure S5: Angles between the weight updates $\Delta W_i$ of all hidden layers with (a) the loss gradient directions and (b) the damped GNT weight updates according to Condition S3 on MNIST. A window-average of the angles is plotted, together with the window-standard-deviation.

(a) Angles between $\Delta W_i$ and the loss gradients.

(b) Angles between $\Delta W_i$ and damped GNT updates.

Figure S6: Angles between the weight updates $\Delta W_i$ of all hidden layers with (a) the loss gradient directions and (b) the damped GNT weight updates according to Condition S3 on CIFAR10. A window-average of the angles is plotted, together with the window-standard-deviation.

(a) Angles between $\Delta W_i$ and the loss gradients.

(b) Angles between $\Delta W_i$ and damped GNT updates.

Figure S7: Angles between the weight updates $\Delta W_i$ of all hidden layers of the small CNN with (a) the loss gradient directions and (b) the damped GNT weight updates according to Condition S3 on CIFAR10. The first two layers are convolutional layer, while the third hidden layer is fully connected. A window–average of the angles is plotted, together with the window–standard–deviation.

**Training losses, fully connected networks.** In Table S8 we provide the training losses that were achieved after 100 epochs for the experiments of Table S6. Note that these training losses are different from those of Table 2, as the hyperparameters are different (here, the hyperparameters are optimized for validation error, while in Table 2 they are optimized for training loss). Better test errors can result from finding a minimum of better quality (e.g. lower and/or flatter), implicit regularization (as we did not use explicit regularization), or a combination of both. Table S8 clearly shows that the improved test performance of the DRL methods compared to DTP and the controls result from finding lower minima.

Table S8: Training loss after 100 epochs for the hyperparameter configurations of Table S6 (optimized for best validation error). The mean and standard deviation over 10 randomly initialized weight configurations are given. The best training losses (except BP) are displayed in bold.

|  | MNIST | Frozen-MNIST | Fashion-MNIST | CIFAR10 |
|---|---|---|---|---|
| BP | $2.62^{\pm 1.45} \cdot 10^{-3}$ | $0.199^{\pm 0.013}$ | $6.22^{\pm 0.19} \cdot 10^{-2}$ | $3.39^{\pm 0.09} \cdot 10^{-5}$ |
| DDTP-linear | $7.51^{\pm 0.69} \cdot 10^{-7}$ | $0.355^{\pm 0.014}$ | $3.73^{\pm 0.34} \cdot 10^{-2}$ | $1.73^{\pm 0.02} \cdot 10^{-4}$ |
| DDTP-RHL | $1.06^{\pm 0.06} \cdot 10^{-6}$ | $0.306^{\pm 0.012}$ | $8.08^{\pm 0.61} \cdot 10^{-2}$ | $7.57^{\pm 0.62} \cdot 10^{-1}$ |
| DDTP-RHL (extra FB) | $9.93^{\pm 0.61} \cdot 10^{-7}$ | $0.308^{\pm 0.023}$ | $\mathbf{2.81^{\pm 0.25} \cdot 10^{-2}}$ | $3.48^{\pm 0.27} \cdot 10^{-1}$ |
| DDTP-RHL (rec. and extra FB) | $9.16^{\pm 4.26} \cdot 10^{-4}$ | $\mathbf{0.286^{\pm 0.011}}$ | $4.44^{\pm 0.44} \cdot 10^{-2}$ | $2.84^{\pm 0.11} \cdot 10^{-1}$ |
| DTPDRL | $8.95^{\pm 4.13} \cdot 10^{-7}$ | $0.350^{\pm 0.018}$ | $3.23^{\pm 0.76} \cdot 10^{-2}$ | $\mathbf{1.38^{\pm 0.07} \cdot 10^{-4}}$ |
| DDTP-control | $3.60^{\pm 2.06} \cdot 10^{-3}$ | $0.988^{\pm 0.055}$ | $5.99^{\pm 0.56} \cdot 10^{-2}$ | $6.58^{\pm 0.23} \cdot 10^{-1}$ |
| DTP | $4.46^{\pm 4.422} \cdot 10^{-3}$ | $1.282^{\pm 0.072}$ | $4.71^{\pm 0.35} \cdot 10^{-2}$ | $5.87^{\pm 0.12} \cdot 10^{-1}$ |
| DTP (pre-trained) | $2.84^{\pm 0.84} \cdot 10^{-6}$ | $1.275^{\pm 0.604}$ | $9.72^{\pm 1.66} \cdot 10^{-2}$ | $1.53^{\pm 0.03} \cdot 10^{-1}$ |
| DFA | $\mathbf{4.87^{\pm 3.57} \cdot 10^{-7}}$ | / | $3.92^{\pm 0.26} \cdot 10^{-2}$ | $2.44^{\pm 0.02} \cdot 10^{-4}$ |

**Training losses, CNNs.** Table S9 shows the training losses that were achieved after 100 epochs for the experiments of Table 3. We see that the training loss of DDTP-linear is close to the training loss of BP, while DFA has a slightly lower loss. As the test performance of DFA is worse compared to BP and DDTP-linear, lower minima do not seem to lead to better test performance for this dataset and these methods. Hence, the test performance likely benefits from implicit regularization and/or small learning rates.

Table S9: Training loss after 100 epochs on CIFAR10 with a small CNN for the hyperparameter configurations of Table 3 (optimized for best validation error). Mean $\pm$ SD for 10 random seeds. The best training loss (except BP) is displayed in bold.

| BP | DDTP-linear | DDTP-control | DFA |
|---|---|---|---|
| $1.15^{\pm 0.03} \cdot 10^{-1}$ | $1.19^{\pm 0.14} \cdot 10^{-1}$ | $2.95^{\pm 1.29} \cdot 10^{-1}$ | $\mathbf{7.95^{\pm 0.87} \cdot 10^{-2}}$ |

To investigate the optimization capabilities of the various methods, we selected hyperparameters tuned for minimizing the training loss and provide the results in Table S10. As can be seen, the training loss of DDTP-linear is an order of magnitude bigger compared to the training losses of DFA and BP. At first sight, this would suggest that the optimization capabilities of DDTP-linear are worse compared to DFA. However, we hypothesize that this worse performance can be explained by the learning rates: DDTP-linear selected learning rates at least an order of magnitude smaller compared to those of DFA. Most likely, DDTP-linear needs to select these small learning rates to make sure that the interplay between feedforward and feedback weight training remains stable. The smaller learning rates combined with the fact that the training loss keeps decreasing (it does not converge) during the whole training procedure for both DFA and DDTP-linear, can explain why DDTP-linear ends up with a bigger training loss. Figure S7 shows that the update directions of DDTP-linear are much better aligned with both the gradients and GNT updates compared to DFA, indicating that DDTP-linear has more efficient update directions for CNNs than DFA. The observation that DFA still succeeds in decreasing the training loss sufficiently while having update directions close to 90 degrees relative to the gradient direction in the convolutional layers (as seen in Figure S7), is remarkable and deserves further investigation in future work.

Table S10: Training loss after 300 epochs on CIFAR10 with a small CNN for a new hyperparameter configuration optimized for training loss. Mean $\pm$ SD for 10 random seeds. The best training loss (except BP) is displayed in bold.

| BP | DDTP-linear | DDTP-control | DFA |
|---|---|---|---|
| $4.07^{\pm0.41} \cdot 10^{-10}$ | $4.31^{\pm1.03} \cdot 10^{-6}$ | $3.71^{\pm0.71} \cdot 10^{-5}$ | $4.94^{\pm0.55} \cdot 10^{-7}$ |

### B.3  Experimental details for the toy experiments

Here, we provide the experimental details of the toy experiments in Fig. 2 and Fig. S3.

**Synthetic regression dataset.**  For both toy experiments, we used a synthetic student-teacher regression dataset. The dataset is generated by randomly initializing a nonlinear teaching network with 4 hidden layers of each 1000 neurons and an input and output layer matching dimensions of the student network. Random inputs are then fed through the teacher network to generate input-output pairs for the training and test set. The teacher network has ReLU activation functions and is initialized with random Gaussian weights, to ensure that the teacher dataset is nonlinear.

**Toy experiment 1.**  For the null-space component toy experiment of Fig. 2, we took a nonlinear student network with two hidden layers of 6 neurons, an input layer of 6 neurons and an output layer of 2 neurons. We trained this student network with DTP (pre-trained) and DDTP-linear on the synthetic dataset and computed the components of $\Delta W_2$ that lie in the nullspace of $\partial \bar{f}_{0,L}(\boldsymbol{h}_0)/\partial \Delta W_2$. Small steps in the direction of these null-space components don't result in a change of output and can thus be considered useless. Furthermore, these null-space components will likely interfere with the updates from other minibatches. We used a minibatch size of 1 for this toy experiment, to investigate the updates resulting from each batch sample separately, without averaging them over a bigger minibatch. We froze the forward weights of the network (but still computed what their updates would be), to investigate the forward parameter updates in an ideal regime, where the feedback weights are trained until convergence on their reconstruction losses, without needing to track changing forward weights.

**Toy experiment 2.**  For the output space toy experiment of Fig. S3, we took a nonlinear student network with two hidden layers of 4 neurons, an input layer of 4 neurons and an output layer of 2 neurons. We trained this student network with GNT and BP on the synthetic dataset and computed in which direction the resulting updates push the output activation of the network. To compute this output space direction, we updated the networks forward parameters with a small learning rate, computed the output activation before and after this update and normalized the difference vector between these two output activations to represent the direction in output space.

## C  Review of the Gauss-Newton method

The Gauss-Newton (GN) algorithm is an iterative optimization method that is used for non-linear regression problems, defined as follows:

$$\min_{\boldsymbol{\beta}} \quad \mathcal{L} = \frac{1}{2} \sum_{b=1}^{B} e^{(b)2} \tag{165}$$

$$e^{(b)} \triangleq y^{(b)} - l^{(b)}, \tag{166}$$

with $L$ the regression loss, $B$ the mini-batch size, $e^{(b)}$ the regression residual of the $b^{\text{th}}$ sample, $y^{(b)}$ the model output and $l^{(b)}$ the corresponding label. The one-dimensional output $y$ is a nonlinear function of the inputs $\boldsymbol{x}$, parameterized by $\boldsymbol{\beta}$. At the end of this section, the Gauss-Newton method will be extended for models with multiple outputs. The Gauss-Newton algorithm can be derived in two different ways: (i) via a linear Taylor expansion around the current parameter values $\boldsymbol{\beta}$ and (ii) via an approximation of Newton's method. Here, we discuss the first derivation, as this will give us the most insights in the inner working of target propagation.

## C.1  Derivation of the method

The goal of a Gauss-Newton iteration step is to find a parameter update $\Delta\boldsymbol{\beta}$ that leads to a lower regression loss:

$$\boldsymbol{\beta}^{(m+1)} \leftarrow \boldsymbol{\beta}^{(m)} + \Delta\boldsymbol{\beta}. \tag{167}$$

Ideally, we want to minimize the regression loss $L$ with respect to the parameters $\boldsymbol{\beta}$ by finding a local minimum:

$$0 \overset{!}{=} \frac{\partial\mathcal{L}}{\partial\boldsymbol{\beta}} = J^T \boldsymbol{e} \tag{168}$$

$$J \triangleq \frac{\partial\boldsymbol{y}}{\partial\boldsymbol{\beta}}, \tag{169}$$

with $\boldsymbol{e}$ a vector containing all the $B$ residuals $e^{(b)}$ and $\boldsymbol{y}$ a vector containing all the outputs. $\boldsymbol{y}$ and $\boldsymbol{e}$ can be approximated by a first order Taylor expansion around the current parameter values $\boldsymbol{\beta}^{(m)}$:

$$\boldsymbol{y}^{(m+1)} \approx \boldsymbol{y}^{(m)} + J\Delta\boldsymbol{\beta} \tag{170}$$

$$\boldsymbol{e}^{(m+1)} = \boldsymbol{y}^{(m+1)} - \boldsymbol{l} \approx \boldsymbol{e}^{(m)} + J\Delta\boldsymbol{\beta} \tag{171}$$

Now this approximation of $\boldsymbol{e}$ can be filled in equation (168), which results in

$$\frac{\partial\mathcal{L}}{\partial\boldsymbol{\beta}} \approx J^T\left(\boldsymbol{e}^{(m)} + J\Delta\boldsymbol{\beta}\right) = 0 \tag{172}$$

$$\Leftrightarrow \quad J^T J\Delta\boldsymbol{\beta} = -J^T \boldsymbol{e}^{(m)}. \tag{173}$$

$J^T J$ can be interpreted as an approximation of the loss Hessian matrix used in Newton's method and is often referred to as the *Gauss-Newton curvature matrix G*. If $J^T J$ is invertible, this leads to:

$$\Delta\boldsymbol{\beta} = -\left(J^T J\right)^{-1} J^T \boldsymbol{e}^{(m)} \tag{174}$$

$$\Delta\boldsymbol{\beta} = -J^\dagger \boldsymbol{e}^{(m)}, \tag{175}$$

With $J^\dagger$ the Moore-Penrose pseudo inverse of $J$. Note that if $J^T J$ is not invertible, $-J^\dagger \boldsymbol{e}^{(m)}$ leads to the solution $\Delta\boldsymbol{\beta}$ with the smallest norm. If $J$ is square and invertible, the pseudo inverse is equal to the real inverse, leading to the following expression:

$$\Delta\boldsymbol{\beta} = -J^{-1}\boldsymbol{e}^{(m)}. \tag{176}$$

Note the similarity between the above equations (173)-(176) and linear least squares: the design matrix $X$ is replaced by the Jacobian $J$ and the residuals and parameter increments are used instead of the output values and the parameters respectively. To get some intuition of this similarity, figure S8 illustrates a Gauss-Newton iteration for a toy problem with only one parameter $\beta$. In figure S8a, the current fitted nonlinear function together with the data samples is shown. Figure S8b shows the linear least squares problem that is solved in equation (173). The residuals $e_{(i)}$ are plotted on the vertical axis and the horizontal axis represents how sensitive these residuals are to a change of parameter $\beta$ (which is represented by $J$). The parameter update $\Delta\beta$ is then represented by the slope of the fitted line through the origin. In more dimensions, the same intuitive interpretation holds, only we have more 'horizontal' axes (one for each parameter sensitivity) and we fit a hyperplane through the origin instead of a line.

As mentioned earlier, the Gauss-Newton method can also be interpreted as an approximation to the Newton method, thereby making Gauss-Newton an approximate second-order optimization method.

## C.2  The Gauss-Newton method for multiple-output models

In the previous paragraphs, the Gauss-Newton method was derived for regression models with a one-dimensional output. This can easily be extended to regression models with multi-dimensional outputs, such as most feed-forward neural networks. The regression loss is now given by:

$$\mathcal{L} = \frac{1}{2}\sum_{b=1}^{B} \|e^{(b)}\|_2^2 \tag{177}$$

$$\boldsymbol{e}^{(b)} \triangleq \boldsymbol{y}^{(b)} - \boldsymbol{l}^{(b)}, \tag{178}$$

Figure S8: Gauss-Newton optimization step for a toy example. (a) The data points together with the fitted curve at the current iteration (b) a visualisation of the linear least squares regression problem that is solved in the Gauss-Newton iteration.

The Jacobian $J$ of the model outputs and samples with respect to the model parameters can be constructed by concatenating the Jacobians of the model outputs for each sample along the row dimension. The resulting Jacobian has size $Bn_{output} \times n_{parameters}$.

### C.3  Tikhonov damping and the Levenberg-Marquardt method

The linear system (173) can sometimes be poorly conditioned, leading to very large step sizes of the Gauss-Newton method, which pushes the model outside the region in which the linear model is a good approximation. The Levenberg-Marquardt (LM) method (Levenberg, 1944; Marquardt, 1963) mitigates this issue by adding Tikhonov damping to the curvature matrix $G = J^T J$:

$$\left(J^T J + \lambda I\right)\Delta\boldsymbol{\beta} = -J^T \boldsymbol{e}^{(m)}, \tag{179}$$

with $\lambda$ the damping parameter. $\lambda$ is typically updated during each training iteration by a heuristic based on trust regions. The added damping prevents the Levenberg-Marquardt method from taking too large steps and thereby greatly stabilizes the optimization process. Intuitively, the Levenberg-Marquardt method can be seen as an interpolation between Gauss-Newton optimization and gradient descent, as for $\lambda \to 0$ the LM method is equal to the GN method, and for $\lambda \to \infty$ the LM method is equal to gradient descent with a very small step size.

## D  Generalized-Gauss-Newton extension of TP

In this section, we discuss how the link between TP and GN optimization can be extended to other losses than the $L_2$ loss. Note that the minimum-norm interpretation of TP discussed in section 3.4 and A.4 holds for all loss functions, only the link to GN is specific to the $L_2$ loss. We start with a brief overview of the Generalized Gauss-Newton method (Schraudolph, 2002) and then provide a theoretical extension to the TP framework to incorporate generalized GN targets. For a more detailed discussion of generalized Gauss-Newton optimization in neural networks, we recommend the PhD thesis of Martens (2016).

### D.1  The generalized Gauss-Newton method

For understanding the generalized Gauss-Newton method, it is best to derive the GN method via Newton's method and then extend it to other loss functions. Newton's method updates the parameters in each iteration as follows:

$$\boldsymbol{\beta}^{(m+1)} \leftarrow \boldsymbol{\beta}^{(m)} - H^{-1}\boldsymbol{g}, \tag{180}$$

with $H$ and $\boldsymbol{g}$ the Hessian and gradient respectively of the loss function $\mathcal{L}$ with respect to parameters $\boldsymbol{\beta}$. For an $L_2$ loss function, the gradient is given by

$$\boldsymbol{g} \triangleq \frac{\partial \mathcal{L}}{\partial \boldsymbol{\beta}} = J^T \boldsymbol{e}_L, \tag{181}$$

with $J$ the Jacobian of your model outputs (of all minibatch samples) with respect to the parameters $\boldsymbol{\beta}$ and $\boldsymbol{e}_L$ the output errors (equal to the derivative of $\mathcal{L}$ with respect to the model outputs because we have and $L_2$ loss). Following equation 165, the elements of the Hessian $H$ are given by

$$H_{jk} = \sum_{b=1}^{B} \left( \frac{\partial e_L^{(b)}}{\partial \beta_j} \frac{\partial e_L^{(b)}}{\partial \beta_k} + e_L^{(b)} \frac{\partial^2 e_L^{(b)}}{\partial \beta_j \partial \beta_k} \right). \tag{182}$$

The Gauss-Newton algorithm approximates the Hessian by ignoring the second term in the above equation, as in many cases, the first term dominates the Hessian. This leads to the following approximation of the Hessian:

$$H \approx G \triangleq J^T J \tag{183}$$

This Gauss-Newton approximation of the Hessian is always a PSD matrix. The update from Newton's method with the approximated Hessian gives rise to the Gauss-Newton update:

$$\boldsymbol{\beta}^{(m+1)} \leftarrow \boldsymbol{\beta}^{(m)} - G^{-1} J^T \boldsymbol{e}_L^{(m)} = \boldsymbol{\beta}^{(m)} - J^\dagger \boldsymbol{e}_L^{(m)} \tag{184}$$

Schraudolph (2002) showed that a similar approximation of $H$ for any loss function $\mathcal{L}$ leads to

$$H \approx G = J^T H_L J \tag{185}$$

with $H_L$ the Hessian of $\mathcal{L}$ w.r.t. the model output. If $\mathcal{L}$ is convex, $H_L$ will be PSD and so will $G$. The Generalized Gauss-Newton (GGN) iteration is then given by:

$$\boldsymbol{\beta}^{(m+1)} \leftarrow \boldsymbol{\beta}^{(m)} - G^{-1} J^T \boldsymbol{e}_L^{(m)} = \boldsymbol{\beta}^{(m)} - \left( J^T H_L J \right)^{-1} J^T \boldsymbol{e}_L^{(m)}. \tag{186}$$

When Thikonov damping is added to $G$, this results in

$$\boldsymbol{\beta}^{(m+1)} \leftarrow \boldsymbol{\beta}^{(m)} - \left( J^T H_L J + \lambda I \right)^{-1} J^T \boldsymbol{e}_L^{(m)}. \tag{187}$$

## D.2 Propagating generalized Gauss-Newton targets

Following the derivation of GGN in section D.1, GGN targets can be defined by

$$\hat{\boldsymbol{h}}_i^{GGN} \triangleq \boldsymbol{h}_i - \left( J_{\bar{f}_{i,L}}^T H_L J_{\bar{f}_{i,L}} + \lambda I \right)^{-1} J_{\bar{f}_{i,L}}^T \boldsymbol{e}_L, \tag{188}$$

with $H_L$ the Hessian of $\mathcal{L}$ w.r.t. $\boldsymbol{h}_L$, $J_{\bar{f}_{i,L}} = \partial f_L(..(f_{i+1}(\boldsymbol{h}_i)))/\partial \boldsymbol{h}_i$ and $\boldsymbol{e}_L = (\partial \mathcal{L}/\partial \boldsymbol{h}_L)^T$. Hence, in order to propagate GGN targets, the following condition should hold for all minibatch samples $b$:

$$J_{\bar{g}_{L,i}}^{(b)} = \left( J_{\bar{f}_{i,L}}^{(b)T} H_L^{(b)} J_{\bar{f}_{i,L}}^{(b)} + \lambda I \right)^{-1} J_{\bar{f}_{i,L}}^{(b)T}. \tag{189}$$

From Theorem 4, we know that the DRL trains the feedback path $\bar{g}_{L,i}$ such that

$$J_{\bar{g}_{L,i}}^{(b)} \approx \left( J_{\bar{f}_{i,L}}^{(b)T} J_{\bar{f}_{i,L}}^{(b)} + \lambda I \right)^{-1} J_{\bar{f}_{i,L}}^{(b)T}. \tag{190}$$

With a small adjustment to the forward mapping $\bar{f}_{i,L}$, we can make sure that $H_L$ is also incorporated in $J_{\bar{g}_{L,i}}^{(b)}$. Consider the singular value decomposition (SVD) of $H_L$:

$$H_L = U_L \Sigma_L V_L^T \tag{191}$$

As $H_L$ is symmetric and PSD, $V_L = U_L$ and $H_L$ can be written as $H_L = K_L K_L^T$, with $K_L \triangleq U\Sigma^{0.5}$. Now let us define $\bar{f}_{i,L}^K$ as

$$\bar{f}_{i,L}^K(\boldsymbol{h}_i) \triangleq K_L^T \bar{f}_{i,L}(\boldsymbol{h}_i). \tag{192}$$

When we use $\bar{f}_{i,L}^K$ in the DRL, this results in:

$$\mathcal{L}_i^{\text{rec,diff,K}} = \frac{1}{\sigma^2} \sum_{b=1}^{B} \mathbb{E}_{\boldsymbol{\epsilon}_1 \sim \mathcal{N}(0,1)} \left[ \| \bar{g}_{L,i}^{\text{diff}} \left( \bar{f}_{i,L}^K(\boldsymbol{h}_i^{(b)} + \sigma\boldsymbol{\epsilon}_1), \boldsymbol{h}_L^{(b)}, \boldsymbol{h}_i^{(b)} \right) - (\boldsymbol{h}_i^{(b)} + \sigma\boldsymbol{\epsilon}_1) \|_2^2 \right]$$

$$+ \mathbb{E}_{\boldsymbol{\epsilon}_2 \sim \mathcal{N}(0,1)} \left[ \lambda \| \bar{g}_{L,i}^{\text{diff}}(\boldsymbol{h}_L^{(b)} + \sigma\boldsymbol{\epsilon}_2, \boldsymbol{h}_L^{(b)}, \boldsymbol{h}_i^{(b)}) - \boldsymbol{h}_i^{(b)} \|_2^2 \right], \tag{193}$$

Via a similar derivation as Theorem 4, it can be shown that the absolute minimum of $\lim_{\sigma \to 0} \mathcal{L}_i^{\text{rec,diff,K}}$ is attained when for each sample it holds that

$$J_{\bar{g}_L,i}^{(b)} = \left(J_{\bar{f}_{i,L}}^{(b)T} K_L^{(b)} K_L^{(b)T} J_{\bar{f}_{i,L}}^{(b)} + \lambda I\right)^{-1} J_{\bar{f}_{i,L}}^{(b)T} K_L^{(b)} \tag{194}$$

$$= \left(J_{\bar{f}_{i,L}}^{(b)T} H_L^{(b)} J_{\bar{f}_{i,L}}^{(b)} + \lambda I\right)^{-1} J_{\bar{f}_{i,L}}^{(b)T} K_L^{(b)} \tag{195}$$

If now the output target is defined as $\hat{\boldsymbol{h}}_L \triangleq \boldsymbol{h}_L - \hat{\eta} K_L^{-1} \boldsymbol{e}_L$, a first-order Taylor expansion of $\hat{\boldsymbol{h}}_i$ results in:

$$\hat{\boldsymbol{h}}_i \approx \boldsymbol{h}_i - \hat{\eta} J_{\bar{g}_L,i} K_L^{-1} \boldsymbol{e}_L. \tag{196}$$

We assume that $H_L$ is PD and hence $K_L$ is invertible. If the absolute minimum of $\mathcal{L}_i^{\text{rec,diff,K}}$ is attained and hence equation (195) holds, the network propagates approximate GGN targets:

$$\hat{\boldsymbol{h}}_i^{(b)} \approx \boldsymbol{h}_i^{(b)} - \hat{\eta} \left(J_{\bar{f}_{i,L}}^{(b)T} H_L^{(b)} J_{\bar{f}_{i,L}}^{(b)} + \lambda I\right)^{-1} J_{\bar{f}_{i,L}}^{(b)T} \boldsymbol{e}_L^{(b)}. \tag{197}$$

Note that $K^{-1} = \Sigma^{-0.5} U^T$ in which $\Sigma$ is a diagonal matrix, hence $K^{-1}$ is straight-forward attained from the SVD of $H_L$. When $\mathcal{L}$ is a sum of element-wise losses on each output neuron such as the cross-entropy loss and (weighted) $L_2$ loss, $H_L$ is a diagonal matrix and hence $K_L = H_L^{0.5}$, which removes the need for an SVD. When the cross-entropy loss is combined with the softmax output, this is not the case anymore and an SVD is needed.

In this section, we briefly summarized how the TP framework can be theoretically extended to propagate GGN targets. Future research can explore whether this GGN extension improves the performance of the TP variants and whether this GGN extension can be implemented or approximated in a more biologically plausible manner.

## Footnotes

[3]PyTorch implementation of all methods is available on `github.com/meulemansalex/theoretical_framework_for_target_propagation`