[Reviews · NeurIPS 2020]

Review 1

Summary and Contributions: The authors show how target propagation (TP) is a hybrid method that uses Gauss-Newton optimization to compute hidden layer targets and then backprop on the local losses. They show that for non-invertible networks, the typically used correction method of difference target propagation (DTP) does not propagate GN targets. They propose a novel difference reconstruction loss (DRL) to train the feedback parameters to propagate GN targets. Evaluating on MNIST, Fashion MNIST and CIFAR-10, their method and its variants perform marginally better than TP and DTP, though they do not still come near backprop.

Strengths: This is solid work connecting backprop, (D)TP and Gauss-Newton optimization. Based on their insights they also propose some improvements to DTP. This work is expected to be extremely useful to the computational neuroscience and neuromorphic communities looking for bio-plausible or local substitutes for backprop.

Weaknesses: The work is already quite extensive and more cannot be reasonably expected. Still, perhaps the authors can comment on these issues: 1) Following up on lines 245-246 'GNT on nonlinear networks does not converge to a true local minimum of the loss function' and the experimental gap between DRL+variants and backprop in section 4 / Table 1. 2) While Manchev and Spratling 2020 claim that their difference target prop for RNNs outperforms BPTT in (simpler) 4 of 5 tasks, why does DRL which is an improvement on DTP not come closer to backprop in these tasks? 3) Would a correction different from GN targets enable to come closer to backprop? 4) In lines 177-178, the authors write 'This damping interpolates between the pseudo-inverse and the transpose of Jf ̄ i,L , so for large λ, GN targets resemble gradient targets.' I would assume that the authors would optimize over lambda as a hyperparameter, but I did not find it in the list of hyperparams in Tables S1-S4. WHy not? In any case, it would be good to explore the dependence of performance on lambda explicitly in a (supplementary) plot and how this impacts performance expecially given that for 'large λ, GN targets resemble gradient targets'.

Correctness: Seems reasonable to me.

Clarity: Very clear and well-written.

Relation to Prior Work: Well discussed.

Reproducibility: Yes

Additional Feedback: Overall, this is a solid paper of timely relevance to the computational neuroscience / neuromorphic audience. Perhaps the authors can address the points raised in weaknesses either in the paper or in supplementary material. [Update: After reading the author response, other reviews and discussion, I find that the authors have further improved their paper. Thus, I maintain my high score.]


Review 2

Summary and Contributions: In this work, the authors present a theoretical analysis of target propagation, showing that it can be interpreted as a hybrid method -- it combines Gauss-Newton-like (GN) calculations of the targets for each layer, with gradient-based updates to the parameters. This analysis is supported with extensive mathematical proofs. The connection to GN targets is only valid under ideal conditions of invertible networks, leading the authors to propose a novel loss, DRL (and several accompanying architectures), which maintains fidelity to the GN targets in practice for non-invertible networks. The authors run several supervised learning experiments and show better performance than DTP (though still comparable).

Strengths: The link between target propagation and Gauss-Newton optimization is intriguing, and appears to be novel. Furthermore, the extensive theory developed for the behavior of target propagation is much needed in ML, which often relies too much on hacky solutions. Thus, this paper presents a useful contribution to the field, and may lead to further developments both in ML and biological learning.

Weaknesses: I did not find any major weaknesses in the content of the work.

Correctness: To the best of my knowledge, everything is correct.

Clarity: The clarify of writing could be improved. Here are some suggestions: (i) The paper is very theoretical in nature, which makes it difficult to understand at times; it would behoove the authors to ensure that the important parts are more clear for a general ML audience. Here are a few specific places which could be improved: - Lines 136-139: "First, the shallow auto-encoder..." -- This explanation is not very clear for me, Can the authors find a better way to say this? - Section 3.3: A schematic figure of DRL would be very helpful here. - Lines 148-155 are very dense; it might be more readable if at least the definition of g_{L,i}^diff is defined on its own line. (ii) The paper seems to overemphasize the performance advances gained by DRL over DTP, at least based upon the results. The reader may be left with the conclusion that DRL fixes the issues with scaling up DTP to larger tasks, though these types of tasks (Bartunov et al.) are not shown in the paper, and the performance gains are modest and still fall behind backprop. This is okay, but it merits a slightly more nuanced discussion. Perhaps the authors can address the differences between the theory and their simulations in the discussion (e.g., squared-error loss versus cross-entropy; Gauss-Newton Target being proved for contracting linear nets with minibatch of 1; using weight decay instead of regularization in the loss function). Could these be an explanation for why the method still performs worse than backprop? (iii) The authors briefly discuss the biological plausibility of DRL, but I would like slightly more information about the comparison to vanilla DTP. E.g., as far as I understand, DRL learns feedback weights by propagating signals from a particular layer all the way to the output (and vice versa), whereas vanilla DTP only involves neighboring layers. Does this change the biological plausibility?

Relation to Prior Work: Yes.

Reproducibility: Yes

Additional Feedback:


Review 3

Summary and Contributions: Post-rebuttal update. The authors address some of the plausibility concerns and provide some evidence for improved performance. I have therefore increased my score to reflect these responses. However, I would point out that the proposed method cannot be considered as a major advancements to biological plausibility for the following two reasons: 1. There are prior work where connections from downstream layers to early layers are present (and in a more realistic setting than the current submission), so this is not an advancement in what is possible. (e.g. [Nøkland and Eidnes 2019] is one example among many) 2. This statement ignores the other issues with biological plausibility such as the concurrent loops of noise. I would therefore suggest the authors greatly tone down the strength of this claim of advancement in biological plausibility and instead provide an honest critique of their own methods with both strengths AND weaknesses. The paper would greatly benefit from this. I would also point out that the experimental evidence provided in the author response looks very suspicios. DDTP-linear in this case achieves a loss 6 times (!) smaller than backprop and more than 2 orders of magnitude smaller than all other methods. At the same time the accuracy achieved by DDTP-linear is worse than backprop. Did the authors perform a fair experiment with an equal amount of time spent optimizing the hyperparameters of all methods? ======================================== This paper takes a deeper look at target propagation (TP) and difference target propagation (DTP) - algorithms previously introduced as biologically plausible backpropagation alternatives - with an eye on theory. The authors clarify TP as a hybrid gradient descent (GD) / Gauss-Newton (GN) algorithm. They further introduce a loss to elevate the DTP algorithm to the same hybrid status and propose new architectures to facilitate this new algorithm.

Strengths: The work adds valuable insight in terms of the relationship between TP and GN. And takes interesting steps to implement a DTP that also enjoys this relationship with GN. Furthermore, the idea of widening the reconstruction loop and its relationship with the fact that the pseudo-inverse is not factorizable is very interesting. The most remarkable empirical demonstration is the performance of DTPDRL in the frozen MNIST experiment. In this experiment were the weights of only the first layer are updated, including DRL improves the performance of DDT significantly. The improved performance of DDTP-linear and DDTP-RHL in this scenario is not surprising as they have direct connections to the first layer. However, the improved performance of DTPDRL is a testament to the power of widening the reconstruction loop.

Weaknesses: There are 4 major weaknesses of this work: 1. The difference reconstruction loss (DRL), while interesting, seems very challenging to implement biologically, as stated in the paper. The main difficulty arises from sending Gaussian noise from each layer to the final layer. The authors require the weights g_i to be updated by only the loop originating at the i'th level. How would a biological network distinguish between the target feedback coming from its own loop vs. the feedback coming from loops originating from earlier layers? This seems like an insurmountable issue with all implementations of DRL (DTPDRL, DDTPlinear, DDTP-RHL). 2. The variations DDTP-linear and DDTP-RHL, while interesting, do not seem to have any biological foundation. With biological plausibility being the primary raison d'être of TP and DTP, any discussions of modifications of architecture should ideally be accompanied with discussions of biological plausibility and realism. 3.The empirical experiments are not convincing with regards to the benefits of DRL (especially considering the issues of point #2) with the exception of MNIST (frozen). This is a good demonstration of DRL (see strengths discussion) but its practical value is questionable given that the experimental setup is not realistic. 4. The theoretical arguments of section 3.4, while again interesting, assume full GN updates. These arguments are strictly speaking only relevant for for networks with exact inverses built in or with DRL implemented and already a optimized to its minimum at every step. The practical relevance (and lessons learned) from these theoretical arguments should be explained for the general case. This is attempted in lines 214-221 and Fig 2. However, these conclusions and especially the large improvement in Fig 2 fail to translate into noticeably improved empirical performance and are therefore of questionable practical utility. ----- The above are weaknesses of this work and do not include the general criticisms of DTP itself (e.g. how realistic is it to backpropagate both the target h^\hat and the forward value h via the same function g). These criticisms apply to DTP as a whole and are not specific to the submission but do further render the relevance of the present work less significant.

Correctness: The theoretical claims seem correct. Regarding the empirical claims, I think that because the comparison is between very similar optimization algorithms, one should report the final values of the training loss and training accuracies and not just the test accuracy. This is because higher values of loss can sometimes have better test accuracy (this is especially true with x-entropy which is used here.) And better test accuracy (generalization) might come from an unintentional implicit regularization associated with an optimizer which might not be related to how well it actually optimizes that objective. (Angle plots such as Fig. 4 provide some insight but do not provide a complete picture.) In short, to be able to understand if improved test accuracy is because of better (deeper) minima or better generalization as a result of implicit regularization, both train/test accuracy as well as train loss should be considered.

Clarity: The paper is mostly well written but readability of the current content can be improved in the following ways: 1. Eq.(2) is different from what most reviews would call TP. Setting eta=1 and L=(h_L-e_L)^2 / 2, gives back the more familiar form of TP. It might be pedagogically useful to mention this relationship or mention why the form of eq. 2 is useful or at least that this is a generalization of just backpropagating the target itself. (Especially since the name target propagation as opposed to error propagation means no derivative backprop for most people, where as Eq.2 has a derivative of the loss in it.) 2. I find the statement in lines 117-119 is more confusing than necessary. It would be more easily understood that this is a hybrid method if one were to take a derivative of L = (h^hat - h)^2 with respect to a weight w to see: dL/dW =2 dh/dW dL/dh The hybrid method then replaces dL/dh with J_fbar^{-1} which is the GN step. In this way it is very clear that this is a hybrid step given by the chain rule \delta W ~ (dh/dW) * (\Delta h) (first term is GD second term is GN). 3. The taylor approximations mentioned in lines 102 and 124 are vague. It would be much clearer if it was explicitly mentioned that the approximation is of h^hat = g_i(h^hat_i+1) evaluated at h i.e. g_i(h_i + h^hat_i - h_i). Without this explanation, since there are many variables here and to the uninitiated reader it is not clear what is being expanded and where. I would also argue that the most important of the architectural choices should be in the main body (e.g. number of layers and width).

Relation to Prior Work: The references provided for the biologically plausible backprop are in no way comprehensive, but this is not necessary as this work focuses on theory and not on comparison with competing models.

Reproducibility: Yes

Additional Feedback: From the perspective of understanding TP and DTP, the submission does a good job of relating these algorithms to a hybrid GD/GN method. However, the contributions made to improve the performance of DTP (DRL etc.) are less convincing both on biological grounds and also on empirical grounds. In short, DRL and other proposed architectures, render any attempt at biological mappings more challenging without resulting in noticeable improvement in performance to justify the reduced realism. In light of the above, the submission can be rendered much stronger in one of the following ways: - Higher focus on TP and DTP with additional insights and less on the extensions. - A more in-depth exploration of DRL to show that it indeed leads to performance improvements as claimed in Sec. 3.4. - An attempt at justifying how DRL or the DDTP architectures can actually be implemented biologically.


Review 4

Summary and Contributions: The authors propose a novel Difference Reconstruction Loss (DRL) and a new neural network training algorithm Direct Difference Target Propagation (DDTP), which is a variant of Target Propagation (TP). Unlike Backpropogation (BP), TP propagates, and updates target activation from the output layer to the first hidden layer. The motivation behind DRL is that when the network is invertible, TP approximates Gauss-Newton optimization to compute the hidden layer targets (GN targets), but when the network is non-invertible, Different Target Propagation (DTP), a well-known variant of TP designed for dealing with non-invertible networks, losses its approximation to GN targets and performs inefficient updates. Hence, by including DRL in DTP, the training algorithm approximates GN targets again. In addition, they provide theories (*) for linear networks that GN targets will result in parameter updates that lie within 90 degrees of the corresponding loss gradient and the network converges to the global minimum with an infinitesimal small learning rate. Finally, Inspired by DRL, the authors propose DDTP, where the network has direct feedback mapping from the output to each hidden layer and can also be trained with DRL. In the experiments, they compare the performance of fully connected networks trained with baseline training algorithms (vanilla DTP and Direct Feedback Alignment, DFA, an algorithm proposed in 2016 that also has direct feedbacks from outputs), DDTP variants, DTP with DRL, and some control algorithms on MNIST, MNIST-frozen (all forward parameters except for those of the first hidden layer are frozen), Fashion-MNIST and CIFAR10. The proposed DDTP shows some improvements over DFA, and moderate improvements over DTP, and is close to the performance of models training by BP, especially on simpler tasks. They further examine the angles between the actual weight updates and (a) the loss gradient directions, (b) the GN target weight updates, and find DDTP and DTP with DRL lead to significantly smaller angles than baselines.

Strengths: Related to neural networks in a biological setting where the updates may not be computed via backpropagation. This is a necessary step if we would like to associate biological neurons with neurons used in machine learning. The improvements are consistent and the proposed algorithm shows close performance to BP. If TP type algorithms can achieve comparable results with BP, it would be interesting to see using them to exclude the drawbacks of BP (eg. vanishing or exploding gradients) or to provide new insights (the credit assignment point of view, which is not discussed in this paper). Well-organized motivations and flow for each proposed theory and method. The experiments also include a variety of algorithms, as well as control.

Weaknesses: Numerically unstable as described in Supplementary Materials Page 26 Numerical Errors. *The theories only consider cases where the batch size is 1, which is not that practical In experiments, weight decay is applied to all algorithms except for baseline DTP, which might be a confounding factor that reduces the usefulness of DRL or DDTP. Only applied to fully connected networks. Although theories for CNNs or ResNets might be more complicated, it would be better to see how the proposed algorithm fits into more practical settings. The assumption of having access to direct feedback mapping from the output to each hidden layer seems a little impractical in a biological sense. Since the environment does not interact with neurons directly.

Correctness: I did not check the proofs but since most of them are applying chain rules and relating to Gauss-Newton method, the theories would probably make sense. I did not re-run their code, but they provide hyperparameters and fixed seeds for all experiment settings used.

Clarity: Well written and clear.

Relation to Prior Work: DRL part seems novel and motivated by solving issues of previous algorithms. DDTP contains direct feedback from outputs, which is similar to DFA proposed in 2016, but the application to TP framework seems novel.

Reproducibility: Yes

Additional Feedback: RESPONSE TO REBUTTAL: The authors addressed one of my main concerns, which is whether the proposed framework generalizes beyond fully connected networks. They ran additional experiments using CNN and showed decent results. However, I am still having a hard time acknowledging the bio-plausibility since it is a strong assumption that you can have direct feedback from the output to all hidden neurons. I will update my score from 6 to 7 for the extension to CNN.

[Author Response · NeurIPS 2020]

We thank the reviewers for their constructive and valuable comments. The resulting revisions and additional experiments significantly strengthened the paper. In the following three sections we summarise and address all major concerns in detail. All other comments will be addressed as well, but not discussed here due to space limitations.

**Biological Plausibility.** Several reviewers raised concerns whether DDTP and DRL are still biologically plausible (rev. comments 2.iii, 3.1, 3.2, 4.5). (i) The purpose of our work was not to validate TP and its variants to be bio-plausible. Instead, we aimed to mathematically analyze the principles of TP/DTP optimization, to uncover strengths and weaknesses and to establish a theoretical framework that allows us and other researchers to address the latter. For example, we linked TP to GN and GD and showed that layer-wise DTP training of feedback paths leads to inefficient parameter updates. Next, we provided a theory-derived solution to address this (DRL). Even if such an improved TP variant turns out to be less bio-plausible we think this is still highly valuable information as it sets new grounds for future discussions in the field. (ii) Regarding the direct or skip feedback connections used in DDTP, we clarify that numerous anatomical studies of the mammalian neocortex consistently reported such direct feedback connections in the brain. In primate visual cortex, both V4 and area MT back-project to V1 (Ungerleider et al., Cereb. Cortex **18**, 2007; Rockland & Van Hoesen, Cereb. Cortex. **4**, 1994). We therefore argue in the revised paper that the flexibility to allow for direct feedback is a major advance in bio-plausibility, compared to methods that only allow strict layer-wise feedback. (iii) DRL requires coordinated noise level alteration to separate reconstruction loops in time which might be biologically questionable. Coordination in time has been used for several major bio-plausible learning methods (Akrout et al., NeurIPS 2019; Kunin et al., ICML 2020) and it is still an open question whether the brain could implement this. Currently, there are several promising paths towards overcoming this need for coordination in time. A first option would be to design a noisy estimator for the DRL derivative where all layers can be noisy simultaneously, similar to Lansdell et al. (ICLR 2020), making use of the correlation between the noise perturbation of layer $i$ and the relevant noise perturbation on the target originating from reconstruction loop $i$. This would directly address reviewer comment 3.1 on how layer $i$ can filter out the target perturbation that originates from its reconstruction loop. A second option would be to not learn the feedback weights explicitly through a reconstruction loss, but to use a dynamical control system for the inversion (Podlaski & Machens, arXiv, 2020) and adapt it such that it approximates GNT. While both options are interesting they require further examination and testing which would go beyond the scope of this work.

**New Experimental Results.** Based on suggestions by reviewer 3 we performed new experiments to benchmark the ability of the new TP variants to minimize the training loss. Table 1 shows the new performance results on Fashion-MNIST (other datasets will also be included in the paper) which reveal that the optimization performance of DDTP-linear is strikingly similar to BP while the DTP/DFA methods are inferior by at least one order of magnitude. Complementing the frozen-MNIST experiments in the paper, these new results show that DRL methods substantially improve optimization by feeding back more useful training signals deep into the network, as predicted by our theory. Furthermore, the new results indicate that DDTP-linear (for simple tasks) converges to fixed points of similar depth as BP, even though it does not converge to true local minima of the loss function (rev. comment 1.1).

Table 1: Training loss of last epoch for Fashion-MNIST (mean $\pm$ SD for n = 5 seeds).

| | |
|---|---|
| BP | $\cdot(6.46 \pm 0.25) \cdot 10^{-5}$ |
| **DDTP-linear** | $\mathbf{(1.03 \pm 0.15) \cdot 10^{-5}}$ |
| DTPDRL | $(1.36 \pm 0.48) \cdot 10^{-3}$ |
| DDTP-RHL | $(3.51 \pm 0.80) \cdot 10^{-3}$ |
| DDTP-control | $(3.88 \pm 2.63) \cdot 10^{-3}$ |
| DTP | $(4.07 \pm 0.42) \cdot 10^{-2}$ |
| DTP (pre-trained) | $(2.73 \pm 0.67) \cdot 10^{-2}$ |
| DFA | $(1.98 \pm 0.24) \cdot 10^{-2}$ |

**From Theory to Practice.** All reviewers suggested a more elaborate discussion on how our theoretical insights translate into a practical/experimental setting (rev. comments 1.1, 2.ii, 3.4, 4.2, 4.4). (i) We now discuss in greater detail how the propagated targets for DRL methods are not exactly equal to GNT because of the limited capacity of the feedback parameterization, limited training iterations for the feedback path and the approximation of $\lambda$ by weight decay and other approximations (see also lines 179-184; 672-721). We discuss as well Figure 2, showing that experimental methods still remove the inefficiencies of DTP, and Figures 4 and S3-S5, demonstrating that our methods well approximate GNT. However, for upstream layers, future studies are required for further improvement, e.g. by investigating better feedback parameterizations or by using dynamical inversion (Podlaski & Machens, arXiv, 2020). A better alignment between targets and GNT in upstream layers will likely improve the performance on more complex tasks such as CIFAR. (ii) We elaborate that $\lambda$ is approximated by weight decay and is negligible in practice due to the observed implicit damping (lines 1204-1232, rev. comment 1.4 and 4.3). (iii) We now discuss in the paper that although mini-batches of 1 are rarely used on GPUs, they are highly relevant for neuromorphic engineering and bio-plausible networks that use online learning. (iv) Finally, we detail that Theorem 4 applies to general forward mappings and that nothing prevents the GNT framework from being applied to CNNs and other feed-forward architectures. As a proof-of-concept, we now include a small CNN (Conv5x5x32; Maxpool3x3; Conv5x5x64; Maxpool3x3; FC512; FC10) on CIFAR10 with DDTP-linear and DFA with FC feedback. We achieved promising results: test error of $24.38 \pm 0.29\%$ (BP), $23.99 \pm 0.31\%$ (DDTP-linear) and $30.00 \pm 0.74\%$ (DFA), indicating that our theory also applies to CNNs. For comparing with DTP and DTPDRL, careful design of the feedback pathways is needed, which is outside of the scope of this theoretical work.

[Meta-Review · NeurIPS 2020]

Congrats on the acceptance— the reviewers provided extensive, constructive input, please do take it into account in preparing the final version of the paper.